# Learning with Expected Signatures: Theory and Applications

Lorenzo Lucchese [1]   Mikko S. Pakkanen [1]   Almut E. D. Veraart [1]

## Abstract

The expected signature maps a collection of data streams to a lower dimensional representation, with a remarkable property: the resulting feature tensor can fully characterize the data generating distribution. This "model-free" embedding has been successfully leveraged to build multiple domain-agnostic machine learning (ML) algorithms for time series and sequential data. The convergence results proved in this paper bridge the gap between the expected signature's empirical discrete-time estimator and its theoretical continuous-time value, allowing for a more complete probabilistic interpretation of expected signature-based ML methods. Moreover, when the data generating process is a martingale, we suggest a simple modification of the expected signature estimator with significantly lower mean squared error and empirically demonstrate how it can be effectively applied to improve predictive performance.

## 1. Introduction

The signature transform of a stream of data is an infinite but countable sequence of its "iterated integrals" summarizing the input in a top-down fashion, meaning the informational content of its terms decays factorially. Originally introduced by Chen (1954) and serving as a fundamental object of rough path analysis (Lyons et al., 2007), the signature

$$\mathbb{S} = \{S(\mathbb{X})_{[0,t]} \in T((\mathbb{R}^d)), \ t \in [0,T]\},$$

of a path $\mathbb{X} = \{\mathbf{X}_t, \ t \in [0,T]\} \in C([0,T], \mathbb{R}^d)$ is a lift (in the sense that it embeds $\mathbb{X}$) to the space of continuous functions over the tensor algebra $T((\mathbb{R}^d))$ possessing some nice algebraic and geometric properties. When the path is of bounded variation, the signature is defined as the sequence

[1]Department of Mathematics, Imperial College London, London, United Kingdom. Correspondence to: Lorenzo Lucchese <lorenzo.lucchese17@imperial.ac.uk, llucchese6@gmail.com>.

*Proceedings of the 42$^{nd}$ International Conference on Machine Learning*, Vancouver, Canada. PMLR 267, 2025. Copyright 2025 by the author(s).

of iterated integrals of $\mathbb{X}$, i.e. for $t \in [0,T]$, $k \geq 0$

$$S^k(\mathbb{X})_{[0,t]} = \int \cdots \int_{0 \leq s_1 \leq \ldots \leq s_k \leq t} \mathrm{d}\mathbf{X}_{s_1} \otimes \cdots \otimes \mathrm{d}\mathbf{X}_{s_k}. \quad (1)$$

In many practical applications the path $\mathbb{X}$ is taken to be the piecewise linear interpolation of a discrete-time stream of data, which is of bounded variation by construction. Signature-based machine learning (ML) approaches (Lyons & McLeod, 2024) thus often restrict the theoretical framework to paths in $\mathrm{BV}([0,T], \mathbb{R}^d)$. In this setting, two fundamental properties of the signature that make it a desirable non-parametric feature extraction method for sequential data are the characterization result of Hambly & Lyons (2005) and the universality approximation theorem of Levin et al. (2016). Moreover, when the path $\mathbb{X}$ is understood as a (realization of a) random process with distribution $\mathbb{P}$ over $\mathrm{BV}([0,T], \mathbb{R}^d)$, the shuffle property of the signature implies that all moments of the random variable $S(\mathbb{X})_{[0,T]}$ are determined by its expectation

$$\phi(T) := \mathbb{E}[S(\mathbb{X})_{[0,T]}] \in T((\mathbb{R}^d)).$$

A natural question, known as the Hamburger moment problem (Fawcett, 2003), is thus whether the expectation of the signature characterizes its law (and thus the law of the path). When imposing a probability distribution $\mathbb{P}$ on $\mathbb{X}$ the assumption of bounded variation paths becomes quite restrictive: Brownian motion, the basic building block of many stochastic models, has paths of infinite variation almost surely. Even if we observe a discrete-time stream of data, we often still would like to define the process $\mathbb{X}$ as a latent stochastic process of which we observe the linear interpolation over some partition $\pi$ of $[0,T]$, hereafter denoted by $\mathbb{X}^\pi$. We hence wish to make sense of the signature of a stochastic process $\mathbb{X}$ with paths of unbounded variation. For a given path $\mathbb{X} \in C([0,T], \mathbb{R}^d)$ of finite $p$-variation, once we "lift" the process to a $p$-rough path (Lyons et al., 2007, Definition 3.11) then the signature $\mathbb{S}$ of $\mathbb{X}$ is uniquely defined[1]. Without delving into the details of rough path theory, for our purposes it suffices to interpret the choice of lift as fixing a notion of integration with respect to $\mathbb{X}$: the higher order signatures terms are then understood as iterated integrals of the path $\mathbb{X}$ defined in this sense.

[1]This is the first fundamental theorem in the theory of rough paths (Lyons et al., 2007, Theorem 3.7).

Motivated by the fact that we can only ever observe the process $\mathbb{X}$ over a discrete partition $\pi$ of $[0, T]$ we restrict our attention to the class of stochastic processes whose lift (and hence signature) can be approximated by the lift (and hence signature) of the bounded variation path $\mathbb{X}^\pi$. Following the rough path literature we take such approximation in the $p$-variation metric to define the notion of *canonical geometric stochastic process*, cf. Definition 2.1. In Chevyrev & Lyons (2016); Chevyrev & Oberhauser (2018) the authors provide characterization results for the expected signature of canonical geometric stochastic processes, i.e. conditions under which the map $\mathbb{P} \mapsto \mathbb{E}[S(\mathbb{X})_{[0,T]}]$ is injective. Such characterizing property of the expected signature has found practical use in a wide range of applications, ranging from classic ML tasks (Lemercier et al., 2021; Triggiano & Romito, 2024; Schell & Oberhauser, 2023) to mathematical finance (Lyons et al., 2021; Futter et al., 2023).

The expected signature is thus a highly informative quantity and, consequently, methods for computing $\phi(T)$ have received considerable research interest. Such methods can be broadly categorized into two classes: those employing an analytical approach and those following a statistical one. Analytical methods aim to develop exact formulas for specific classes of models. A first step in this direction was taken in Ni (2012, Section 4) showing that the expected signature of an Itô diffusion satisfies an explicit partial differential equation (PDE). This result was subsequently generalized in Cuchiero et al. (2023) to the class of signature-SDEs and in Friz et al. (2022; 2024) to (discontinuous) semimartingales. On the other hand, the statistical approach aims to estimate $\phi(T)$ directly from observed data, preserving the model-free nature of the expected signature. For a given set of observations $\mathbb{X}^{1,\pi}, \ldots, \mathbb{X}^{N,\pi}$ one can form the estimator

$$\hat{\phi}^{N,\pi}(T) := \frac{1}{N} \sum_{n=1}^{N} S(\mathbb{X}^{n,\pi})_{[0,T]},$$

as illustrated in Figure 1, and study its in-fill $|\pi| \to 0$ and large-sample $N \to \infty$ asymptotics. This line of work includes the explicit results of Ni (2012, Section 3.2) for Brownian motion and of Passeggeri (2020) for fractional Brownian motion with Hurst parameter $H > 1/2$ as well as the preliminary results in Friz & Victoir (2010) for more general semimartingales. Additionally, Schell & Oberhauser (2023, Section 8) develops asymptotic results for processes of bounded variation. In this work we provide a unifying set of general conditions under which the expected signature estimator $\hat{\phi}^{N,\pi}(T)$ displays important asymptotic statistical properties, namely consistency and asymptotic normality. Our results allow for irregular[2] observation partitions $\pi$ – possibly varying across samples – and for dependency

across the samples $\mathbb{X}^{1,\pi}, \ldots, \mathbb{X}^{n,\pi}$. The first main contribution of this paper is thus to bridge the gap between the empirical expected signature estimator and the expected signature of a latent continuous-time stochastic process, unlocking a more general probabilistic interpretation of several ML algorithms and effectively moving beyond the expected signature as a simple feature extraction method. This naturally leads to the second theoretical contribution: by starting from the continuous-time setting we devise a modification of the expected signature estimator with significantly better finite sample properties when the latent data generating process is a martingale. The superior performance of this modified estimator is empirically verified through various experiments with expected signature-based ML algorithms from the literature.

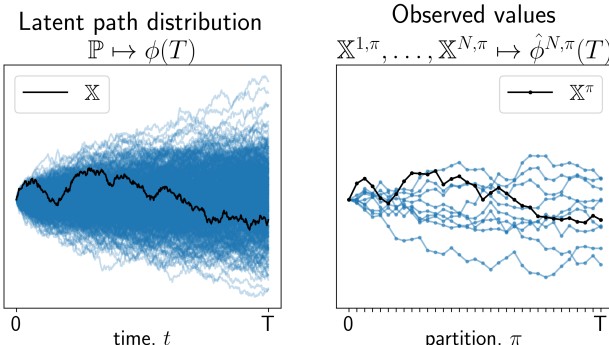

Latent path distribution
$\mathbb{P} \mapsto \phi(T)$

Observed values
$\mathbb{X}^{1,\pi}, \ldots, \mathbb{X}^{N,\pi} \mapsto \hat{\phi}^{N,\pi}(T)$

*Figure 1.* Estimating the expected signature estimation from a finite collection of discretely-observed paths.

## 2. Theory

Let $\mathbb{X} = \{\mathbf{X}_t, \ t \in [0, T]\}$ denote a $d$-dimensional stochastic process over the probability space $(\Omega, \mathcal{F}, \mathbb{P})$.

**Definition 2.1.** We say $\mathbb{X}$ is a canonical geometric stochastic process of rough order $p$ if there exists a sequence of partitions $\rho$ with $|\rho| \to 0$ such that the limit in the $p$-variation metric of the canonically lifted linearly interpolated process $\mathbb{X}^\rho$ exists in probability. Convergence in probability implies almost sure convergence (along a subsequence) and hence we can almost surely define the lift of $\mathbb{X}$ as such limit.

*Remark* 2.2. The definition of lift suggests this might depend on the choice of the sequence of partitions $\rho$. In any case, for a wide range of stochastic processes there exist *canonical* lifts that satisfy our definition of canonical geometric rough path. These include:

- Semimartingales: For $p \in (2, 3)$ any semimartingale can be lifted to a geometric $p$-rough path by defining the lift via Stratonovich integration; the signature of $\mathbb{X}$ then coincides with iterated Stratonovich integrals. For any sequence of partitions $\rho$ the lifts of the linear

---

[2]Clearly, for the estimation problem to be well-posed, the sequence of partitions needs to be signature defining in the sense of Definition 2.5.

interpolations converge in $p$-variation metric to the Stratonovich lift (Friz & Victoir, 2010, Chapter 14) and hence $\mathbb{X}$ is a canonical geometric stochastic process in the sense of Definition 2.1.

- Gaussian processes: Many Gaussian processes admit canonical lifts to geometric $p$-rough paths (Friz & Victoir (2010, Theorem 15.34, Definition 15.35) and Coutin & Qian (2002)) with the existence criterion for such canonical lifts easily stated in terms of the covariance function. The definition of the lift implicitly requires $\rho$ to be any sequence such that $\mathbb{X}^\rho$ converges uniformly to $\mathbb{X}$ almost surely. For example, fractional Brownian motion with Hurst parameter $H > 1/4$ can be lifted to a geometric $p$-rough path with $p > 1/H$ by choosing $\rho$ to be the sequence of dyadic partitions.

In what follows, we will assume the canonical geometric stochastic process $\mathbb{X}$ has a *canonical* lift (i.e. a canonical sequence of partitions $\rho$ along which the lift is defined) and unambiguously refer to it as *the* lift of $\mathbb{X}$.

Let $\rho$ denote a partition of $[0, T]$ with mesh $|\rho|$ and $\mathbb{X}^\rho = \{\mathbf{X}_t^\rho, \ t \in [0, T]\}$ the linear approximation of $\mathbb{X}$ over $\rho$, i.e.

$$\mathbf{X}_t^\rho = \mathbf{X}_u + \frac{t - u}{v - u} \mathbf{X}_{u,v}, \quad t \in [u, v] \in \rho,$$

with $\mathbf{X}_{u,v} = \mathbf{X}_v - \mathbf{X}_u$. The signature of the bounded variation path $\mathbb{X}^\rho$ up to time $t \in [0, T]$ is defined by Equation (1) through classic Riemann-Stieltjes integration and can thus be computed by

$$S(\mathbb{X}^\rho)_{[0,t]} = \bigotimes_{[u,v] \in \rho_{[0,t]}} \exp_\otimes \mathbf{X}_{u,v}^\rho, \quad (2)$$

where $\rho_{[0,t]}$ denotes the restriction of $\rho$ to $[0, t]$. The canonical lift of $\mathbb{X}^\rho$ to a (geometric) $p$-rough path

$$\left(1, S^1(\mathbb{X}^\rho)_{[0,t]}, \ldots, S^{\lfloor p \rfloor}(\mathbb{X}^\rho)_{[0,t]}\right) \in T^{\lfloor p \rfloor}\left((\mathbb{R}^d)\right), \quad (3)$$

for $t \in [0, T]$. Definition 2.1 requires that there exists a sequence of partitions $\rho$ for which this sequence of geometric $p$-rough paths converges in probability in the $p$-variation metric. A key result from rough path theory is that a geometric $p$-rough path has a full signature. Fixing the lift of $\mathbb{X}$ via Definition 2.1, we thus have a uniquely specified signature for $\mathbb{X}$.

**Definition 2.3.** The signature of a canonical geometric stochastic process $\mathbb{X}$,

$$\mathbb{S} = \{S(\mathbb{X})_{[0,t]} \in T((\mathbb{R}^d)), \ t \in [0, T]\},$$

is defined pathwise (on a set of full measure) as the unique extension of the lift of $\mathbb{X}$ to a multiplicative functional of arbitrary order in the sense of (Lyons et al., 2007, Theorem 3.7). The elements of the signature are the rough iterated integrals of $\mathbb{X}$.

*Remark* 2.4. Taking $\rho$ to be a sequence such that Definition 2.1 holds, by continuity of the extension map (Lyons et al., 2007, Theorem 3.10), it immediately follows that the signature of $\mathbb{X}^\rho$ (truncated at level $K \geq \lfloor p \rfloor$) converges in probability to the signature of $\mathbb{X}$ (up to level $K$) in the $p$-variation topology. In particular, this implies that, for any finite collection of words $\mathbf{I}$,

$$S^{\mathbf{I}}(\mathbb{X}^\rho)_{[0,t]} \xrightarrow{\mathbb{P}} S^{\mathbf{I}}(\mathbb{X})_{[0,t]}. \quad (4)$$

Similar arguments imply that, when convergence to the lift along $\rho$ holds almost surely in the $p$-variation metric, then also the higher order signature terms converge almost surely, and, in particular, (4) holds in the almost sure limit.

In the following sections, we will be estimating the expected signature at fixed time horizon $T > 0$. To develop the properties of these estimators, it will be thus sufficient to work with pointwise limits like (4) without having to deal with the stronger pathwise $p$-variation convergence used to define canonical geometric stochastic processes. This mode of convergence will thus be sufficient to consider a sequence of partitions as *signature-defining*.

**Definition 2.5.** Let $\mathbb{X} = \{\mathbf{X}_t, \ t \in [0, T]\}$ be a canonical geometric stochastic process, we say that a sequence of partitions $\pi$ of the interval $[s, t] \subseteq [0, T]$ with $|\pi| \to 0$ is signature-defining if for any collection of words $\mathbf{I}$,

$$S^{\mathbf{I}}(\mathbb{X}^\pi)_{[s,t]} \xrightarrow{\mathbb{P}} S^{\mathbf{I}}(\mathbb{X})_{[s,t]}, \quad |\pi| \to 0. \quad (5)$$

### 2.1. Expected Signature Estimation

In this section, we assume we have access to $N$ copies of $\mathbb{X}$ discretely observed over possibly different partitions of the interval $[0, T]$, i.e. each $\mathbb{X}^{n, \pi_{N,n}}$ is an observation over $\pi_{N,n}$ of a continuous-time latent process $\mathbb{X}^n$, for $n = 1, \ldots, N$. We will focus on two observational schemes:

(ind) Repeatedly observe $\mathbb{X}$ through $N$ independent experiments, in which case the "underlying" signatures $S(\mathbb{X}^n)_{[0,T]}$, for $n = 1, \ldots, N$, are independent and identically distributed.

(chop) Chop-up (and shift in time) a single observation of the process $\{\mathbf{X}_t, \ t \geq 0\}$ over a partition

$$\Pi(N) := \pi_{N,1} \cup \cdots \cup ((N - 1)T + \pi_{N,N}),$$

of $[0, NT]$. In this setting, we assume that the latent sequence $\{\mathbb{X}^n, \ n \geq 1\}$ taking values in $C([0, T]; \mathbb{R}^d)$ is stationary, i.e. for $k \in \mathbb{N}$, $n_1, \ldots, n_k \in \mathbb{N}$ and $n \geq 0$,

$$(\mathbb{X}^{n_1}, \ldots, \mathbb{X}^{n_k}) \overset{\mathcal{L}}{=} (\mathbb{X}^{n_1 + n}, \ldots, \mathbb{X}^{n_k + n}), \quad (6)$$

and hence the signatures $S(\mathbb{X}^n)_{[0,T]}$ form a stationary sequence. This assumption ensures the task of estimating $\phi_{\mathbf{I}}(T)$ is well-posed. Note this condition is slightly stronger than necessary but weaker than requiring $\{\mathbf{X}_t,\ t \geq 0\}$ to be stationary, cf. Proposition 2.13.

The first observational framework can be recast in the second by appropriately pasting the $\mathbb{X}^n$'s into a single process $\{\mathbf{X}_t,\ t \geq 0\}$. Going forward we hence focus on the second setting and refer to the large sample asymptotics $N \to \infty$ as long-span asymptotics. For any finite collection of words $\mathbf{I}$, we thus consider the estimator

$$\hat{\phi}_{\mathbf{I}}^{\Pi(N)}(T) := \frac{1}{N}\sum_{n=1}^{N} S^{\mathbf{I}}(\mathbb{X}^{n,\pi_{N,n}})_{[0,T]}. \qquad (7)$$

We will be interested in the double asymptotics where, as the number of signature evaluations $N$ increases, the granularity of the discretized paths from which such signatures are computed also increases, i.e.

$$|\Pi(N)| := \max_{1 \leq n \leq N} |\pi_{N,n}| \to 0, \quad N \to \infty.$$

We can decompose

$$\hat{\phi}_{\mathbf{I}}^{\Pi(N)}(T) - \phi_{\mathbf{I}}(T)$$
$$= \frac{1}{N}\sum_{n=1}^{N}\left(S^{\mathbf{I}}(\mathbb{X}^{n,\pi_{N,n}})_{[0,T]} - S^{\mathbf{I}}(\mathbb{X}^n)_{[0,T]}\right)$$
$$+ \frac{1}{N}\sum_{n=1}^{N} S^{\mathbf{I}}(\mathbb{X}^n)_{[0,T]} - \mathbb{E}\left[S^{\mathbf{I}}(\mathbb{X})_{[0,T]}\right]. \qquad (8)$$

Under suitable conditions, we shall prove $\hat{\phi}_{\mathbf{I}}^{\Pi(N)}(T)$ is consistent and asymptotically normal for $\phi_{\mathbf{I}}(T)$ by showing

1. each summand in the first term converges to zero in $L^m$ in the in-fill asymptotics $|\pi_{N,n}| \to 0$;

2. the second term, when inflated by $\sqrt{N}$, converges in distribution to a normal random variable in the large sample asymptotics $N \to \infty$.

### 2.1.1. IN-FILL ASYMPTOTICS

The convergence in probability (5) is not sufficient to show consistency of the expected signature estimator. In this section, we thus explore continuity conditions on the process $\mathbb{X}$ ensuring the convergence holds in a stronger $L^m$ sense.

Let $\{\mathcal{F}_{s,t},\ [s,t] \subseteq [0,T]\}$ be a family of sigma-algebras such that, for $[u,v] \subseteq [s,t] \subseteq [0,T]$, $\mathcal{F}_{u,v} \subseteq \mathcal{F}_{s,t}$ and, for $[s,t] \subseteq [0,T]$, $\mathbf{X}_{s,u}$ is $\mathcal{F}_{s,t}$-measurable for all $u \in [s,t]$.

The following continuity assumptions will be used to state the in-fill asymptotics.

**Assumption 2.6.** For all $0 \leq s < u < t \leq T$,

(A$\alpha$) $\|\mathbf{X}_{s,t}\|_{L^p} \lesssim |t-s|^\alpha$.

(A$\beta$) $\|\mathbb{E}_{\mathcal{F}_{0,s}\vee\mathcal{F}_{t,T}}[\mathbf{X}_{s,u}\otimes\mathbf{X}_{u,t}]\|_{L^{p/2}} \lesssim |t-s|^\beta$.

(A$\gamma$) $\|\mathbb{E}_{\mathcal{F}_{0,s}\vee\mathcal{F}_{t,T}}[\mathbf{X}_{s,u}\otimes\mathbf{X}_{u,t}^{\otimes 2}]\|_{L^{p/3}} \lesssim |t-s|^\gamma$,
$\|\mathbb{E}_{\mathcal{F}_{0,s}\vee\mathcal{F}_{t,T}}[\mathbf{X}_{s,u}^{\otimes 2}\otimes\mathbf{X}_{u,t}]\|_{L^{p/3}} \lesssim |t-s|^\gamma$.

(A$\delta$) $\|\mathbb{E}_{\mathcal{F}_{0,s}}[\mathbf{X}_{s,t}]\|_{L^p} \lesssim |t-s|^\delta$.

*Remark* 2.7. By the contraction property of the conditional expectation, the strongest form of (A$\beta$), (A$\gamma$) and (A$\delta$) is obtained by setting $\mathcal{F}_{s,t} = \sigma(\mathbf{X}_{s,u},\ u \in [s,t])$.

**Theorem 2.8.** *Let* $k = \max_{I\in\mathbf{I}}|I|$ *and, for* $m \geq 2$, *set* $p = mk$. *Assume* $\mathbb{X}$ *is a canonical geometric stochastic process that satisfies one of the following:*

*(i)* (A$\alpha$) *for* $\alpha > 1/2$;

*(ii)* (A$\alpha$), (A$\delta$) *for* $\alpha = 1/2, \delta \geq 1$;

*(iii)* (A$\alpha$), (A$\beta$) *for* $\alpha \in (1/3, 1/2], \beta > 1$;

*(iv)* (A$\alpha$), (A$\beta$), (A$\gamma$) *for* $\alpha \in (1/4, 1/3], \beta > 1, \gamma > 1$;

*with*

$$\epsilon = \begin{cases} 2\alpha - 1, & \text{if (i)}, \\ (2\alpha - 1/2) \wedge (\alpha + \delta - 1), & \text{if (ii)}, \\ 3\alpha \wedge \beta - 1, & \text{if (iii)}, \\ 4\alpha \wedge \beta \wedge \gamma - 1, & \text{if (iv)}, \end{cases} \qquad (9)$$

*and consider a signature-defining, cf. Definition 2.5, sequence of refining partitions* $\{\pi_n,\ n \geq 1\}$ *of the interval* $[0,T]$ *such that*

$$\sum_{n\geq 1}|\pi_n|^\epsilon < \infty,$$

*then the stronger convergence holds*

$$S^{\mathbf{I}}(\mathbb{X}^{\pi_n})_{[0,T]} \xrightarrow{L^m} S^{\mathbf{I}}(\mathbb{X})_{[0,T]}, \quad n \to \infty, \qquad (10)$$

*with rate* $\mathcal{O}(\sum_{n'\geq n}|\pi_{n'}|^\epsilon)$.

*Proof.* See Appendix B.1. $\qquad\qquad\square$

*Remark* 2.9. Note that, if $\{\pi_n,\ n \geq 1\}$ is a sequence of dyadic partitions with $|\pi_n| = 2^{-n}T$, then

$$\sum_{n\geq 1}|\pi_n|^\epsilon = \sum_{n\geq 1}2^{-n\epsilon}T^\epsilon = \frac{T^\epsilon}{1 - 2^{-\epsilon}} < \infty,$$

and the rate of convergence is $\mathcal{O}(2^{-n\epsilon})$.

### 2.1.2. LONG-SPAN ASYMPTOTICS

**Theorem 2.10.** *Fix $T > 0$ and let $\{\mathbf{X}_t,\ t \geq 0\}$ be a stochastic process such that $\mathbb{X}^1 = \{\mathbf{X}_t,\ t \in [0,T]\}$ satisfies the assumptions of Theorem 2.8 with $m > 2$. Assume $\{\mathbb{X}^n,\ n \geq 1\}$ is stationary and ergodic and the sequence of partitions $\{\Pi(N),\ N \geq 1\}$ is such tha,t for each $n \geq 1$, $\pi_{\cdot,n} = \{\pi_{N,n},\ N \geq n\}$ is a signature-defining sequence of refining partitions, and*

$$\sum_{N' \geq N} |\Pi(N')|^\epsilon \to 0, \quad N \to \infty. \tag{11}$$

*Then the expected signature estimator (7) is*

1. *consistent, i.e. $\hat{\phi}_{\mathbf{I}}^{\Pi(N)}(T) \xrightarrow{L^2} \phi_{\mathbf{I}}(T)$ as $N \to \infty$.*

*If, moreover, $\{\mathbb{X}^n,\ n \geq 1\}$ is strongly mixing with mixing coefficient $\{\alpha(n),\ n \geq 1\}$ such that, for $\zeta = m - 2 > 0$,*

$$\sum_{n \geq 1} \alpha(n)^{\zeta/(2+\zeta)} < \infty, \tag{12}$$

*and*

$$\sqrt{N} \sum_{N' \geq N} |\Pi(N')|^\epsilon \to 0, \quad N \to \infty, \tag{13}$$

*where $\epsilon$ is given in Equation (9), then the estimator is also*

2. *asymptotically normal, i.e.*

$$\sqrt{N}\left(\hat{\phi}_{\mathbf{I}}^{\Pi(N)}(T) - \phi_{\mathbf{I}}(T)\right) \xrightarrow{\mathcal{L}} \mathcal{N}(0, \Sigma_{\mathbf{I}}), \quad N \to \infty,$$

*as long as $\Sigma_{\mathbf{I}}$ is strictly positive definite, where*

$$\begin{aligned}\Sigma_{\mathbf{I}} =& \mathrm{Var}\left(S^{\mathbf{I}}(\mathbb{X}^1)_{[0,T]}\right) \\ &+ 2 \sum_{n \geq 2} \mathrm{Cov}\left(S^{\mathbf{I}}(\mathbb{X}^1)_{[0,T]}, S^{\mathbf{I}}(\mathbb{X}^n)_{[0,T]}\right).\end{aligned}$$

*Proof.* See Appendix B.2. □

*Remark* 2.11. If $\{\Pi(N),\ N \geq 1\}$ is a sequence of expanding dyadic refinements, i.e. for each $n \geq 1$, $\pi_{\cdot,n}$ is a sequence of dyadic partitions with $|\pi_{N,n}| = 2^{-N}T$, $N \geq n$, as in Remark 2.9, then $|\Pi(N)| = 2^{-N}T$ and, hence,

$$\sqrt{N} \sum_{N' \geq N} |\Pi(N')|^\epsilon = \mathcal{O}(\sqrt{N}2^{-\epsilon N}) \to 0, \quad N \to \infty.$$

**Corollary 2.12.** *Assume the conditions of Theorem 2.10 hold with Theorem 2.8.(ii) satisfied for some $m > 4$ and for any $T > 0$. Assume furthermore we can characterize the rate of convergence of Theorem 2.10.1 as $\rho(N) \sim N^{-\upsilon}$ for some $\upsilon \in (0,1)$. Then the kernel estimator*

$$\hat{\Sigma}_{\mathbf{I}}^{\Pi(N)} = \sum_{|n| \leq h_N} \hat{\Sigma}_{\mathbf{I}}^{n,\Pi(N)},$$

*with $h_N = N^{\upsilon/2}$, non-overlapping cross-covariances*

$$\hat{\Sigma}_{\mathbf{I}}^{n,\Pi(N)} = \frac{1}{M} \sum_{m=1}^{M} [S^{\mathbf{I}}(\mathbb{X}^{\pi_{N,(n+1)m-n}})_{[0,T]} - \hat{\phi}_{\mathbf{I}}^{\Pi(N)}(T)]$$

$$\times [S^{\mathbf{I}}(\mathbb{X}^{\pi_{N,(n+1)m}})_{[0,T]} - \hat{\phi}_{\mathbf{I}}^{\Pi(N)}(T)]^{\mathrm{T}},$$

*for $M = \lfloor N/(n+1) \rfloor$ and*

$$\hat{\Sigma}_{\mathbf{I}}^{-n,\Pi(N)} := \left(\hat{\Sigma}_{\mathbf{I}}^{n,\Pi(N)}\right)^{\mathrm{T}}, \quad n = 1, \ldots, N-1,$$

*is consistent for $\Sigma_{\mathbf{I}}$, i.e. $\Sigma_{\mathbf{I}}^{\Pi(N)} \xrightarrow{L^2} \Sigma_{\mathbf{I}}$ as $N \to \infty$, and hence the CLT result of Theorem 2.10 can be made feasible.*

*Proof.* See Appendix B.3. □

Requiring $\{\mathbb{X}^n,\ n \geq 1\}$ to be stationary and ergodic or strongly mixing are high-level conditions. The following results give stronger but easier-to-interpret conditions.

**Proposition 2.13.** *Fix $T > 0$ and let $\{\mathbf{X}_t,\ t \geq 0\}$ be a stochastic process. Then[3]*

$\{\mathbf{X}_t,\ t \geq 0\}$ *is stationary*
$\implies \{\mathbf{X}_t,\ t \geq 0\}$ *has jointly stationary increments*
$\implies \{\mathbb{X}^n,\ n \geq 1\}$ *is stationary.*

*If any of the above holds, and $\mathbb{X}^1$ is a canonical geometric stochastic process, then, for any collection of words $\mathbf{I}$,*

$\{\mathbf{X}_t, t \geq 0\}$ *is strongly mixing*
$\implies \{\mathbb{X}^n,\ n \geq 1\}$ *is strongly mixing.*

*Proof.* See Appendix B.4. □

One might expect a similar statement to hold for ergodicity, but Remark B.6 shows that

$\{\mathbf{X}_t, t \geq 0\}$ is ergodic $\not\implies \{\mathbb{X}^n,\ n \geq 1\}$ is ergodic.

Strong mixing implies ergodicity and hence the second part of Proposition 2.13 yields a sufficient condition (as far as $\{\mathbb{X}^n,\ n \geq 1\}$ is concerned) for both the consistency and asymptotic normality results of Theorem 2.10. Strong mixing is a somewhat restrictive assumption and hence one might wish to find a set of interpretable conditions weaker than strong mixing ensuring at least consistency of the estimator. The following theorem gives such a condition when $\{\mathbf{X}_t,\ t \geq 0\}$ is a Gaussian process.

---

[3]We say $\{\mathbf{X}_t,\ t \geq 0\}$ has jointly stationary increments if for all $n \in \mathbb{N}$, $0 \leq s_i \leq t_i$ with $i = 1, \ldots, n$, and $t \geq 0$,

$$(\mathbf{X}_{s_1,t_1}, \ldots, \mathbf{X}_{s_n,t_n}) \stackrel{\mathcal{L}}{=} (\mathbf{X}_{t+s_1,t+t_1}, \ldots, \mathbf{X}_{t+s_n,t+t_n}). \tag{14}$$

**Theorem 2.14.** *Fix $T > 0$ and let $\{\mathbf{X}_t, \ t \geq 0\}$ be a Gaussian process such that $\mathbb{X} = \{\mathbf{X}_t, \ t \in [0,T]\}$ is a canonical geometric stochastic process satisfying[4] $(A\alpha)$ with $\alpha \geq 1/2$ and $p = 2$. Assume the sequence of dyadic partitions of $[0,T]$ is signature-defining for $\mathbb{X}$ and for each $N \geq 1$ let $\pi_{N,n}$ be the dyadic partition the interval $[0,T]$ with mesh $|\pi_{N,n}| = 2^{-N}T$.*

*Suppose $\{\mathbf{X}_t, \ t \geq 0\}$ has constant mean and time-homogeneous increment covariance, i.e. $\forall u,v,s,t,r \geq 0$*

$$\mathrm{Cov}\left(\mathbf{X}_{u,v}, \mathbf{X}_{s,t}\right) = \mathrm{Cov}\left(\mathbf{X}_{u+r,v+r}, \mathbf{X}_{s+r,t+r}\right),$$

*satisfying, for some decreasing $\theta : \mathbb{R}_+ \to \mathbb{R}_+$ with $\theta(t) \to 0$, $t \to \infty$ and $\int_0^T \theta(t)dt < \infty$ and $m \in \mathbb{N}$,*

$(A\theta)$ $\|\mathrm{Cov}\left(\mathbf{X}_{u,v}, \mathbf{X}_{s,t}\right)\| \lesssim \theta(|s-v|)|v-u||t-s|,$

*for all $0 \leq u \leq v < s \leq t$ with $|s-v| \geq \frac{m}{2}(|t-s|+|v-u|)$. Then the expected signature estimator (7) is consistent, i.e. $\hat{\phi}_{\mathbf{I}}^{\Pi(N)}(T) \xrightarrow{\mathbb{P}} \phi_{\mathbf{I}}(T)$ as $N \to \infty$.*

*Proof.* See Appendix B.5. $\square$

### 2.2. Variance Reduction via Martingale Correction

In Section 2.1 we developed the necessary theory to establish the asymptotic properties of the estimator (7) for the statistic $\phi_I(T) = \mathbb{E}[S^I(\mathbb{X})_{[0,T]}]$, for any word $I = (i_1, \ldots, i_k)$. This section aims to find an alternative estimator with better finite sample properties when the process $\mathbb{X} = \{\mathbf{X}_t, \ t \in [0,T]\}$ is a martingale. We restrict ourselves to the independent observation setting, with the same partition across samples, i.e. $\pi_{N,n} = \pi$ for $n = 1, \ldots, N$. We will hence be considering the estimator

$$\hat{\phi}_I^{N,\pi}(T) := \frac{1}{N} \sum_{n=1}^N S^I(\mathbb{X}^{n,\pi})_{[0,T]}, \qquad (15)$$

where the $\mathbb{X}^{n,\pi}$ are i.i.d. piecewise linear observations of $\mathbb{X}$ over the partition[5] $\pi$. We introduce the control-variate modification of the estimator (15),

$$\hat{\phi}_I^{N,\pi,c}(T) := \frac{1}{N} \sum_{n=1}^N \left( S^I(\mathbb{X}^{n,\pi})_{[0,T]} - cS_c^I(\mathbb{X}^{n,\pi})_{[0,T]} \right), (16)$$

where, setting $I_{-1} := (i_1, \ldots, i_{k-1})$,

$$S_c^I(\mathbb{X}^\pi)_{[0,T]} := \sum_{[u,v] \in \pi} S^{I_{-1}}(\mathbb{X}^\pi)_{[0,u]} X_{u,v}^{(i_k)}.$$

---

[4]When $\alpha = 1/2$, assume furthermore $\mathbb{X}$ satisfies $(A\delta)$ with $\delta \geq 1$ and $p = 2k$ where $k = \max_{I \in \mathbf{I}} |I|$.

[5]Note that by Friz & Victoir (2010, Chapter 14) any sequence of partitions with vanishing mesh size is signature-defining.

The correction term $S_c^I(\mathbb{X}^\pi)_{[0,T]}$ is inspired by considering the continuous-time signature

$$S^I(\mathbb{X})_{[0,T]} = \int_0^T S^{I_{-1}}(\mathbb{X})_{[0,s]} \circ \mathrm{d}X_s^{(i_k)},$$

where the integral is defined in the Stratonovich sense. To preserve the estimator's unbiasedness while reducing the variance we aim to find a mean-zero control variate $S_c^I(\mathbb{X})_{[0,T]}$ that is highly correlated with $S^I(\mathbb{X})_{[0,T]}$. A natural candidate is

$$S_c^I(\mathbb{X})_{[0,T]} = \int_0^T S^{I_{-1}}(\mathbb{X})_{[0,s]} \, \mathrm{d}X_s^{(i_k)},$$

where the outermost integral is now interpreted in the Itô sense. If $\mathbb{X}$ is a square-integrable martingale satisfying the conditions of Jacod & Shiryaev (1987, Theorem I.4.40), $\{S_c^I(\mathbb{X})_{[0,t]}, \ t \in [0,T]\}$ is also a square-integrable martingale with $\mathbb{E}[S_c^I(\mathbb{X})_{[0,T]}] = 0$. Going back to the discretized setting, we note that, when $\mathbb{X}$ is a martingale, the discretized correction term $S_c^I(\mathbb{X}^\pi)_{[0,T]}$ is also mean-zero and, hence, the control variate estimator $\hat{\phi}_I^{N,\pi,c}(T)$ has the same bias as $\hat{\phi}_I^{N,\pi}(T)$, but, when picking the optimal[6]

$$c = c_\pi^* := \frac{\mathrm{Cov}(S^I(\mathbb{X}^\pi)_{[0,T]}, S_c^I(\mathbb{X}^\pi)_{[0,T]})}{\mathrm{Var}(S_c^I(\mathbb{X}^\pi)_{[0,T]})},$$

it has reduced variance

$$\mathrm{Var}(\hat{\phi}_I^{N,\pi,c_\pi^*}(T)) = (1 - \rho_{I,\pi}^2)\mathrm{Var}(\hat{\phi}_I^{N,\pi}(T)),$$

where $\quad \rho_{I,\pi} := \mathrm{Corr}(S^I(\mathbb{X}^\pi)_{[0,T]}, S_c^I(\mathbb{X}^\pi)_{[0,T]}).$

In practice, to estimate $c_\pi^*$, the most straightforward approach would be to use the sample variance and covariance. In this case the estimator for $c_\pi^*$ is the slope of the simple linear regression of $\{S^I(\mathbb{X}^{n,\pi})_{[0,T]}, \ n = 1, \ldots, N\}$ against $\{S_c^I(\mathbb{X}^{n,\pi})_{[0,T]}, \ n = 1, \ldots, N\}$ or, exploiting the mean zero property of the control,

$$\hat{c}_\pi^* = \frac{\sum_{n=1}^N S^I(\mathbb{X}^{n,\pi})_{[0,T]} S_c^I(\mathbb{X}^{n,\pi})_{[0,T]}}{\sum_{n=1}^N S_c^I(\mathbb{X}^{n,\pi})_{[0,T]}^2}.$$

In Appendix C.2 we propose an alternative estimator for $c_\pi^*$ derived using the properties of the signature.

*Remark* 2.15. This variance reduction technique is not limited to processes $\mathbb{X}$ that are *full* martingales but can also be applied to *partial* martingales, i.e. $\mathbb{X}$ such that only a subset of the components is a martingale. In this case, we can use the control variate expected signature estimator for any word $I = (i_1, \ldots, i_k)$ such that $\mathbb{X}^{(i_k)}$ is a martingale.

---

[6]We assume throughout $\mathrm{Var}(S_c^I(\mathbb{X}^\pi)_{[0,T]}) \in (0, \infty)$.

Even when the data generating process $\mathbb{X}$ is not a martingale, the variance reduction achieved by the corrected estimator (16) may outweigh the bias it introduces, leading to better performance – in terms of mean squared error (MSE) – than the classic estimator (15). In cases where the underlying process cannot be assumed to be a martingale we thus suggest to treat the martingale correction as a data transformation applicable in the learning pipeline (a model hyper-parameter in a similar spirit to the add-time or the lead-lag transform in the signature context) whose usefulness may be empirically ascertained via cross-validation.

## 3. Applications

### 3.1. Examples

We now consider a few concrete examples of continuous-time stochastic processes satisfying the assumptions of Theorem 2.10 and Theorem 2.14. Note that BM, CAR and Heston are semimartingales and hence, by Remark 2.2, they are canonical geometric stochastic processes such that any sequence of partitions with vanishing mesh size is signature defining. fBm is instead an example of a process that is not a semimartingale but is a canonical geometric stochastic process with dyadic signature-defining sequence of partitions (Remark 2.2). Taking $\{\Pi(N),\ N \geq 1\}$ to be a sequence of expanding dyadic partitions thus ensures the observational assumptions of Theorem 2.10 and Theorem 2.14 are satisfied by all four processes, cf. Remark 2.11.

**BM**  A standard Brownian motion $\{\mathbf{B}_t,\ t \geq 0\}$. It can be easily checked it satisfies (A$\alpha$) and (A$\delta$), for any $\alpha \geq 1/2, \delta \geq 1$ and $p \geq 2$. Moreover, $\{\mathbf{B}_t,\ t \geq 0\}$ has stationary and independent increments and, hence, the (ind) and (chop) sampling schemes are equivalent: in both cases we can apply[7] Theorem 2.10 to deduce consistency and asymptotic normality of the expected signature estimator.

**fBm**  A fractional Brownian motion $\{\mathbf{B}_t^H,\ t \geq 0\}$ with Hurst parameter $H > 1/2$. $\mathbb{B}^H$ satisfies (A$\alpha$) with $\alpha = H$ (Appendix E.2.2) and, hence, Assumption 2.6 is fulfilled. Under (ind) sampling, $\{\mathbb{B}^{H,n}, n \geq 1\}$ is trivially stationary and strong mixing and, hence, we can apply Theorem 2.10. When instead paths are obtained under (chop) we can apply[8] Theorem 2.14, cf. Example E.2.2, to deduce consistency.

---

[7]Brownian motion is a Gaussian process with constant mean function and time-homogeneous covariance of the increments trivially satisfying (A$\theta$) with $\theta \equiv 0$ and $m = 0$, it thus also falls under the scope of Theorem 2.14.

[8]The increments of fractional Brownian motion are not strongly mixing (Mandelbrot & Van Ness, 1968) and, hence, we cannot apply the second part of Theorem 2.10 to deduce asymptotic normality.

**CAR**  A bidimensional Continuous-time Autoregressive (CAR) process $\{\mathbf{Y}_t,\ t \geq 0\}$ of order $p = 2$ driven by a standard Brownian motion with drift $\mathbf{A} = (A_1, A_2) \in (\mathbb{R}^{2\times 2})^2$. The CAR process is defined as the first $d = 2$ entries of its $pd = 4$-dimensional state space representation $\{\mathbf{X}_t,\ t \geq 0\}$: an Ornstein-Uhlenbeck process with drift and diffusion

$$A_{\mathbf{A}} = \begin{pmatrix} 0_{2\times 2} & -I_{2\times 2} \\ A_2 & A_1 \end{pmatrix}, \quad \Sigma = \begin{pmatrix} 0_{2\times 2} & 0_{2\times 2} \\ 0_{2\times 2} & I_{2\times 2} \end{pmatrix},$$

(Lucchese et al., 2023; Marquardt & Stelzer, 2007). We can apply the first set of conditions in Appendix D.1.2 to deduce that $\{\mathbf{X}_t,\ t \geq 0\}$ (and hence $\{\mathbf{Y}_t,\ t \geq 0\}$) satisfies (A$\alpha$) and (A$\delta$), for $\alpha = 1/2$, $\delta = 1$ and any $p \geq 2$. Under (ind) sampling we can hence apply Theorem 2.10. Moreover, when $A_{\mathbf{A}}$ has positive real parts of all eigenvalues and the process is started in its stationary distribution, $\{\mathbf{X}_t,\ t \geq 0\}$ and $\{\mathbf{Y}_t,\ t \geq 0\}$ are stationary, ergodic and strongly mixing with strong mixing coefficient $\alpha(t) = \mathcal{O}(e^{-at})$, for some $a > 0$ (Marquardt & Stelzer, 2007). We can hence apply Proposition 2.13 to deduce that $\{\mathbb{Y}^n,\ n \geq 1\}$ is stationary and strongly mixing with strong mixing coefficient $\alpha(n) = \mathcal{O}(e^{-anT})$, i.e. satisfying Equation (12), for (any) $\zeta > 0$. Under (chop) sampling we can thus apply[9] the consistency and asymptotic normality results of Theorem 2.10.

**Heston**  The joint price-variance dynamics of a Heston model under the risk-neutral measure $\mathbb{Q}$ with zero interest rate and no dividends, i.e. $\{(S_t, V_t),\ t \geq 0\}$ such that

$$\mathrm{d}S_t = \sqrt{V_t}S_t \mathrm{d}W_t^S,$$
$$\mathrm{d}V_t = \kappa(\theta - V_t)\mathrm{d}t + \xi\sqrt{V_t}\mathrm{d}W_t^V,$$

where $\{W_t^S,\ t \geq 0\}$ and $\{W_t^V,\ t \geq 0\}$ are standard Brownian motions with correlation $\langle W^S, W^V \rangle_t = \rho t$. Under the Feller condition $2\kappa\theta > \xi^2$, the variance process is strictly positive (and so is $\{S_t,\ t \geq 0\}$). The Heston model is thus an Itô diffusion with Lipschitz drift $f : \mathbb{R}_+ \times \mathbb{R}_+ \mapsto \mathbb{R}^2$ and $1/2$-Hölder continuous diffusion $\sigma : \mathbb{R}_+ \times \mathbb{R}_+ \mapsto \mathbb{R}^{2\times 2}$. We can thus apply the third case of Appendix D.1.2 to prove that $\{(S_t, V_t),\ t \geq 0\}$ satisfies (A$\alpha$) and (A$\delta$) with $\alpha = 1/2$, $\delta = 1$ and any $p > 2$ for deterministic initial conditions $S_0 = s_0$ and $V_0 = v_0$. When paths are sampled under (ind) we can hence apply Theorem 2.10 to deduce consistency and asymptotic normality of the expected signature estimator.

### 3.2. Experiments

Quite a wide range of learning algorithms has been developed leveraging the properties of the expected signature. The theory for such algorithms is usually developed under

---

[9]The CAR process is a Gaussian process satisfying (A$\theta$), cf. Appendix E.2.1, and hence also falls under the scope of Theorem 2.14.

the assumption of bounded variation paths for the input process $\mathbb{X}$, assumed to be piecewise linear. The results in Section 2.1 give the theoretical foundation for their probabilistic interpretation when the underlying process $\mathbb{X}$ is an, arguably more realistic, continuous-time stochastic process such as the ones discussed in Section 3.1. In this section we review a few algorithms from the literature, showcasing the practical relevance of the asymptotic results of Section 2.1 and the potential improvements achieved by the martingale correction introduced in Section 2.2. Code and examples demonstrating the integration of the martingale correction into machine learning algorithms, along with the simulation results from the previous section, are available at https://github.com/lorenzolucchese/esig. The code is designed to be compatible with Python-based ML pipelines, supporting both numpy arrays and torch tensors.

### 3.2.1. TIME SERIES CLASSIFICATION

The first model we consider, introduced in Triggiano & Romito (2024), falls under the general task of time series classification, mapping an input path $\mathbf{x} \in \mathbb{R}^{d \times M_1}$ to a class label $c \in \mathcal{C}$. The input stream is interpreted as a discrete-time realization of a Gaussian process, whose conditional mean and covariance are learned parametrically. The expected signature of the latent Gaussian process, used as input to a classification layer, is estimated by super-sampling the process. Theorem 2.14 ensures this approach consistently estimates the expected signature of the latent continuous-time Gaussian process, a fundamental step for the probabilistic interpretation of the algorithm.

We replicate the synthetic data experiments of Triggiano & Romito (2024) on the (FBM), (OU) and (Bidim) datasets. The performance on the out-of-sample testing datasets of the Gaussian Process augmented Expected Signature (GPES) classifier with and without martingale correction is reported in Table 1. The output of the GPES model is by construction stochastic and, hence, we repeat the evaluation of the model with 10 different seeds. In Table 1 we report the mean accuracy and standard error of the model with and without martingale correction (MC), as well as the results of an independent samples $t$-test between their accuracies. The martingale correction significantly improves the performance of the GPES model, a remarkable result considering that most processes in the three datasets are not martingales.

### 3.2.2. PRICING PATH-DEPENDENT DERIVATIVES

The next application we consider is a purely financial one. The objective is to price (and hedge) path-dependent derivatives by decomposing them into a set of atomic Arrow-Debreu-like securities. Let $\mathbb{X} = \{\mathbf{X}_t, \ t \in [0, T]\}$ be a price process, i.e. a semimartingale over some probability space. In Lyons et al. (2021, Proposition 4.5) the authors use the

| | Predictive Accuracy [%] | | |
| | FBM | OU | Bidim |
| --- | --- | --- | --- |
| GPES | 95.62 (0.18) | 62.20 (0.70) | 79.33 (0.46) |
| GPES-MC | 95.26 (0.70) | 88.26 (0.31) | 88.97 (0.44) |
| $t$-stat | 1.49 | $-101.92$ | $-45.52$ |
| $p$-value | 0.15 | 0.00 | 0.00 |

*Table 1.* Synthetic data experiments of Triggiano & Romito (2024): GPES model without and with martingale correction (MC).

universality of the signature to show that a large class of path-depend payoffs $F$ can be arbitrarily well approximated by a linear payoff on the signature, i.e.

$$\text{price}(F) = \mathbb{E}^{\mathbb{Q}}[Z_T F] \approx \langle f, Z_T \mathbb{E}^{\mathbb{Q}}[S(\hat{\mathbb{X}}^{\text{LL}})_{[0,T]}] \rangle,$$

for a set of linear coefficients $f \in T((\mathbb{R}^4)^*)$ where $\mathbb{Q}$ is a pricing measure for $\mathbb{X}$, $Z_T$ a deterministic discount factor over $[0, T]$ and $\hat{\mathbb{X}}^{\text{LL}}$ denotes the add-time lead-lag transform of $\mathbb{X}$. In Appendix F.2.2 we also discuss the corresponding hedging problem.

Given a pricing model $\mathbb{Q}$ for $\mathbb{X}$, we can hence price $F$ via Monte Carlo simulations. This provides a classic setting for applying the martingale correction described in Section 2.2 since, under $\mathbb{Q}$, the (discounted) price process $\mathbb{X}$ is a martingale. In Figure 2, we compare the finite sample properties of the expected signature estimator with and without martingale correction when the price process is assumed to follow a Brownian motion (BM); in the context of option pricing, this is known as the Bachelier model. Similarly, in Figure 3, we plot the densities of the two estimators under the Heston dynamics[10] (Heston). Both figures suggest the martingale correction (blue) materially improves the classic estimator (red), and hence more accurate pricing is achieved by the modified estimator introduced in Section 2.2.

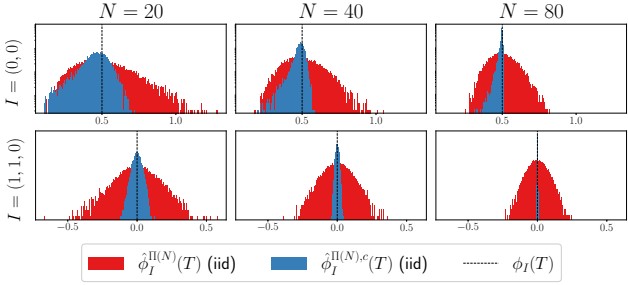

*Figure 2.* Distributions of expected signature estimators for BM. The $y$-axis is in log-scale.

---

[10]In both simulations we fix $T = 1$ and consider $\pi$ to be uniform with mesh $|\pi| = 2^{-\lfloor N/10 \rfloor + 1}$. This choice ensures the sequences of partitions are signature-defining for both processes and satisfy the conditions necessary for consistency and asymptotic normality, cf. Remark 2.11.

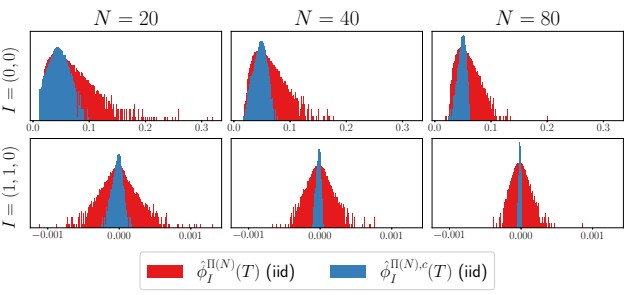

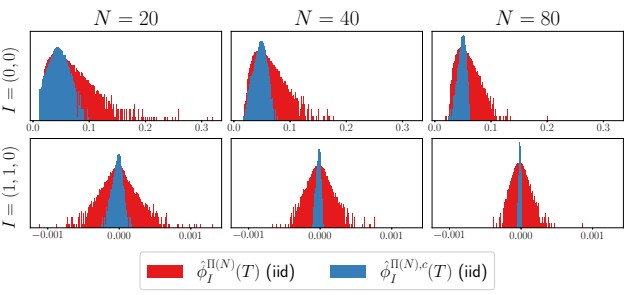

*Figure 3.* Distributions of expected signature estimators for the Heston process with parameters $s_0 = 1, v_0 = 0.1, \theta = 0.1, \kappa = 0.6, \xi = 0.2$ and $\rho = -0.15$. The $y$-axis is in log-scale.

### 3.2.3. DISTRIBUTIONAL REGRESSION FOR STREAMS

Introduced in Lemercier et al. (2021), the Signature of the pathwise Expected Signature (SES) model aims to learn a map from a collection of paths, understood as an empirical measure on path space, to a scalar value, a task known as distributional regression. Under appropriate conditions, the authors show that linear functionals on the signature of the pathwise expected signature are universal for weakly continuous functions (Lemercier et al., 2021, Theorem 3.2).

We repeat two of the synthetic data experiments conducted in Lemercier et al. (2021), analyzing the performance of the SES model without and with martingale correction (MC). We report the average out-of-sample mean-squared error (MSE) and its standard deviation in Table 2 and Table 3, as well as the $t$-statistic and $p$-value of a pairwise t-test between the MSEs of the two models. While the results do not yield statistical significance there still seems to be a mild benefit in using the martingale correction, especially considering that the processes of both experiments are not martingales[11].

|  | Predictive MSE [$\times 10^{-2}$] | |
|  | $r_1 = 0.35 \times \sqrt[3]{V/N}$ | $r_2 = 0.65 \times \sqrt[3]{V/N}$ |
|---|---|---|
| SES | 1.27 (0.23) | 0.09 (0.03) |
| SES-MC | 1.31 (0.45) | 0.07 (0.02) |
| $t$-stat | $-0.29$ | 1.41 |
| $p$-value | 0.79 | 0.23 |

*Table 2.* Ideal gas experiment of Lemercier et al. (2021): SES model without and with martingale correction (MC).

| | Predictive MSE [$\times 10^{-3}$] | | |
| | $N = 20$ | $N = 50$ | $N = 100$ |
|---|---|---|---|
| SES | 1.49 (0.39) | 0.33 (0.13) | 0.20 (0.08) |
| SES-MC | 1.26 (0.48) | 0.31 (0.09) | 0.19 (0.05) |
| $t$-stat | 0.87 | 0.63 | 0.29 |
| $p$-value | 0.43 | 0.56 | 0.79 |

*Table 3.* Rough volatility experiment of Lemercier et al. (2021): SES model without and with martingale correction (MC).

## 4. Conclusions

In this paper, we established new estimation results for the expected signature, a model-free embedding for collections of data streams. Our consistency and asymptotic normality results bridge the gap between the theoretically "optimal" continuous-time expected signature and the empirical discrete-time estimator that can be computed from data. Moreover, we introduced a simple modification of such an estimator with significantly better finite sample properties under the assumption of martingale observations. Our empirical results suggest that the modified estimator might improve the performance of models employing expected signature computations even when the underlying data generating process is not necessarily a martingale.

## Acknowledgements

This research has been supported by the EPSRC Centre for Doctoral Training in Mathematics of Random Systems: Analysis, Modelling and Simulation (EP/S023925/1). The authors would like to thank Nicola Muca Cirone and Will Turner for helpful discussions on the topic, as well as the three anonymous reviewers for their insightful comments.

## Impact Statement

This paper presents work whose goal is to advance the field of Machine Learning. There are many potential societal consequences of our work, none which we feel must be specifically highlighted here.

---

[11]In the first experiment, when the particle radii are large and collisions are more frequent, one could argue the motion of the gas particles to be amenable to that of pollen grains in water, the original experiment which led to the "discovery" of Brownian motion by Scottish botanist Robert Brown in 1827.

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

# Contents of the Appendix

# A. Informal Glossary

This informal glossary provides accessible explanations of selected technical terms and notational conventions used in this paper, aimed at readers with little or no background in rough path theory. These intuitive definitions are intended to aid the understanding of the theoretical framework presented in Section 2, particularly Definition 2.1. However, they remain closely tied to more technical definitions – such as those of multiplicative functionals, rough paths, and geometric rough paths – which require a more rigorous exposition of rough path theory. For a concise introduction to rough paths, we refer the reader to Lyons et al. (2007), and for a treatment in the stochastic setting, to Friz & Victoir (2010).

$p$-**variation**     The $p$-variation of a path is a measure of its regularity. For the purpose of our discussion it suffices to note that paths that have finite $p$-variation for low $p$ are more regular. A bounded variation (BV) path is a path with finite 1-variation (also known as total variation). This regularity ensures there exists a well-defined notion of integral against this path (e.g. a piecewise linear paths or continuously differentiable path) and, hence, we can easily define its signature as in Equation (2). Many interesting stochastic processes (e.g. those driven by Brownian motion) have infinite 1-variation (i.e. are not BV) but have finite $p$-variation for all $p > 2$ and, hence, defining their signature requires rough path theory.

**Convergence in** $p$-**variation**     Convergence in the $p$-variation metric/topology is a pathwise mode of convergence (i.e. over all points $t \in [0, T]$ simultaneously) that is (much) stronger than the pointwise (i.e. at fixed $t \in [0, T]$) convergence required to state and prove our results. See, for example, Remark 2.4.

**Spaces of paths**     We denote by $C([0, T], \mathbb{R}^d)$, resp. BV$([0, T], \mathbb{R}^d)$, the space of $\mathbb{R}^d$-valued continuous, resp. bounded variation, paths over the interval $[0, T]$.

**Mesh of a partition**     For a partition $\pi = \{0 = t_0 < t_1 < \ldots < T\}$ of $[0, T]$, we define its mesh as $|\pi| = \max_{[s,t] \in \pi} |t - s|$ where the maximum is taken over all sub-intervals of the partition.

**Shuffle property**     The shuffle property of the signature is an algebraic property stating that the product of two signature terms is a linear combination of higher-order signature terms. More precisely, the product of the signature terms corresponding to words $I$ and $J$ is the sum of all signature terms indexed by words $K$ of length $|I| + |J|$ obtained by interleaving $I$ and $J$. In the context of the discussion on page 1 this means that all moments of the signature can be written as linear combinations of higher order expected signature terms.

**Signature indexing**     A word $I = (i_1, \ldots, i_n)$ with $i_1, \ldots, i_n \in \{1, \ldots, d\}$ is a multi-index used to denote an entry of the signature, i.e. a real-valued number. The length of the word, i.e. $|I| = n$, denotes the signature level, i.e. an $n$-dimensional tensor, to which such entry belongs. For example $S^I(\mathbb{X})_{[0,T]}$, where $I = (1, 2)$, denotes the $(1, 2)$-entry of the second level of the signature (a matrix), while $I = (2, 1, 1)$ denotes the $(2, 1, 1)$-entry of the third level of the signature (a three-dimensional tensor).

**Stochastic processes**     A continuous stochastic process $\mathbb{X} = \{\mathbf{X}_t, \ t \in [0, T]\}$ over a probability space $(\Omega, \mathcal{F}, \mathbb{P})$ is such that, for each $\omega \in \Omega$, the realization $\mathbb{X}(\omega) = \{\mathbf{X}_t(\omega), \ t \in [0, T]\} \in C([0, T], \mathbb{R}^d)$. If one takes $\Omega = C([0, T], \mathbb{R}^d)$ and $\mathbb{P}$ a probability measure over this path space then each $\omega \in \Omega$ denotes a possible path realization of $\mathbb{X}$. We thus say a property holds pathwise or almost surely if the set of $\omega \in \Omega$ for which that property holds has probability one.

**Canonical geometric stochastic process**     We define a canonical geometric stochastic process as a continuous stochastic process whose "higher order structure" can be approximated by the iterated integrals of its piecewise-linear interpolations (in probability in the $p$-variation metric). Its signature is then defined as the limit of the signatures of its piecewise-linear interpolations, i.e. the iterated integrals given in Equation (3). For clarity, we emphasize that *canonicity* here refers to the aforementioned construction of the signature, not to the underlying probability space on which the process is defined.

# B. Proofs of Section 2

### B.1. Proof of Theorem 2.8

> *Sketch of proof. The main idea of the proof is to show the sequence of discretized signatures $\{S^k(\mathbb{X}^{\pi_n})_{[0,T]}, \ n \geq 1\}$ is Cauchy in $L^m$. Since $L^m$ is a Banach space this implies the sequence converges in $L^m$. By uniqueness of limits, we can*

*deduce this limit is the same as its $\mathbb{P}$-limit, i.e. $S^k(\mathbb{X})_{[0,T]}$. To show the sequence is Cauchy in $L^m$ we proceed inductively on the signature level $k' \in \{1, \ldots, k\}$ under the progressively weaker norm $L^{mk/k'}$. The main ingredient of the inductive step is a manipulation of the discrete-time signature (2), ensuring*

$$S^{k'}(\mathbb{X}^{\pi_{n+1}})_{[\tau_0, \tau_1]} - S^{k'}(\mathbb{X}^{\pi_n})_{[\tau_0, \tau_1]}, \quad [\tau_0, \tau_1] \subseteq [0, T],$$

*can be written as a sum over time intervals $\pi_{n+1, [\tau_0, \tau_1]}$. The inductive assumption is then verified by bounding this summation using Lemma B.1 when a simple Minkowski bound is too weak. We use two different manipulations of the discrete-time signature under assumptions (i), (iii) or (iv) and under assumption (ii): In the former case we use the classic representation given in (2), while in the latter we rely on the "causal" representation of Lemma B.3. For clarity of exposition we thus divide the proof of Theorem 2.8 into two parts.*

We first establish a couple of useful lemmas which will be used repeatedly in the proof of this in-fill asymptotic results. The first is a basic result which is also applied in the proof of the stochastic sewing lemma (Lê, 2020). In the following, let $E$ denote a Banach space.

**Lemma B.1.** *Let $\{Z_n, \ n = 1, \ldots, N\}$ be a finite sequence of $E$-valued random variables in $L^m$ with $m \in [2, \infty)$ and let $\{\mathcal{G}_n, \ n = 1, \ldots, N\}$ be a filtration such that, for each $n \in \{1, \ldots, N\}$, the variables $Z_1, \ldots, Z_{n-1}$ are $\mathcal{G}_n$-measurable. Then*

$$\left\| \sum_{n=1}^{N} Z_n \right\|_{L^m} \leq \sum_{n=1}^{N} \|\mathbb{E}_{\mathcal{G}_n}[Z_n]\|_{L^m} + 2C_m \left( \sum_{n=1}^{N} \|Z_n\|_{L^m}^2 \right)^{1/2}.$$

*Proof.*

$$
\begin{aligned}
\left\| \sum_{n=1}^{N} Z_n \right\|_{L^m} &\overset{(i)}{\leq} \left\| \sum_{n=1}^{N} \mathbb{E}_{\mathcal{G}_n}[Z_n] \right\|_{L^m} + \left\| \sum_{n=1}^{N} (Z_n - \mathbb{E}_{\mathcal{G}_n}[Z_n]) \right\|_{L^m} \\
&\overset{(ii)}{\leq} \sum_{n=1}^{N} \|\mathbb{E}_{\mathcal{G}_n}[Z_n]\|_{L^m} + C_m \left\| \sum_{n=1}^{N} \|Z_n - \mathbb{E}_{\mathcal{G}_n}[Z_n]\|^2 \right\|_{L^{m/2}}^{1/2} \\
&\overset{(iii)}{\leq} \sum_{n=1}^{N} \|\mathbb{E}_{\mathcal{G}_n}[Z_n]\|_{L^m} + C_m \left( \sum_{n=1}^{N} \|Z_n - \mathbb{E}_{\mathcal{G}_n}[Z_n]\|_{L^m}^2 \right)^{1/2} \\
&\overset{(iv)}{\leq} \sum_{n=1}^{N} \|\mathbb{E}_{\mathcal{G}_n}[Z_n]\|_{L^m} + C_m \left( \sum_{n=1}^{N} (\|Z_n\|_{L^m} + \|\mathbb{E}_{\mathcal{G}_n}[Z_n]\|_{L^m})^2 \right)^{1/2} \\
&\overset{(v)}{\leq} \sum_{n=1}^{N} \|\mathbb{E}_{\mathcal{G}_n}[Z_n]\|_{L^m} + 2C_m \left( \sum_{n=1}^{N} \|Z_n\|_{L^m}^2 \right)^{1/2},
\end{aligned}
$$

by using in $(i)$ the triangle inequality, in $(ii)$ triangle inequality and the Burkholder-Davis-Gundy (BDG) inequality (Burkholder et al., 1972) applied to the martingale $\{M_n, \ n = 1, \ldots, N\}$ with $M_n = \sum_{i=1}^{n} (Z_i - \mathbb{E}_{\mathcal{G}_i}[Z_i])$, in $(iii)$ and $(iv)$ the triangle inequality and in $(v)$ the contraction property of conditional expectation. $\square$

**Lemma B.2.** *Let $p, p_1, \ldots, p_l \in (0, \infty) \cup \{+\infty\}$ be such that $p_1^{-1} + \ldots + p_l^{-1} = p^{-1}$, then, for any set of tensors $\mathbf{A_1} \in L^{p_1}((\mathbb{R}^d)^{\otimes k_1}), \ldots, \mathbf{A_l} \in L^{p_l}((\mathbb{R}^d)^{\otimes k_l})$,*

$$\|\mathbf{A}_1 \otimes \cdots \otimes \mathbf{A}_l\|_{L^p} \lesssim d^l \|\mathbf{A}_1\|_{L^{p_1}} \cdots \|\mathbf{A}_l\|_{L^{p_l}}.$$

*Proof.*

$$
\begin{aligned}
\|\mathbf{A}_1 \otimes \cdots \otimes \mathbf{A}_l\|_{L^p} &\leq \sum_{(w_1, \ldots, w_l) \in \mathcal{W}_{k_1 + \ldots + k_l}} \|A_1^{w_1} \cdots A_l^{w_l}\|_{L^p} \\
&\overset{(*)}{\leq} \sum_{(w_1, \ldots, w_l) \in \mathcal{W}_{k_1 + \ldots + k_l}} \|A_1^{w_1}\|_{L^{p_1}} \cdots \|A_l^{w_l}\|_{L^{p_l}}
\end{aligned}
$$

$$\leq d^{k_1 + \ldots + k_l} \|\mathbf{A}_1\|_{L^{p_1}} \cdots \|\mathbf{A}_l\|_{L^{p_l}},$$

where $\mathcal{W}_k = \{1, \ldots, d\}^k$ denotes the set of words of length $k$ and, in $(*)$, we applied the classical Hölder inequality. $\square$

We also prove a useful lemma that allows us to write the $k$-th level signature of a piecewise linear path as a "causal" sum of lower order signature terms, i.e. preserving time order. This will allow us to derive an in-fill result with assumptions on the regularity of $\mathbb{E}_{\mathcal{F}_{0,s}}[\mathbf{X}_{s,t}]$, a more natural object than $\mathbb{E}_{\mathcal{F}_{0,s} \vee \mathcal{F}_{t,T}}[\mathbf{X}_{s,u} \otimes \mathbf{X}_{u,t}]$, when $\alpha = 1/2$, i.e. Theorem 2.8 under *(ii)*.

**Lemma B.3.** *Let $\pi$ be a partition of $[0, T]$ and let $\tau \in \pi$. Then, for $k \geq 0$, we can write*

$$S^{k+1}(\mathbb{X}^\pi)_{[0,\tau]} = \sum_{i=0}^{k} \frac{1}{(1+i)!} \sum_{[u,v] \in \pi_{[0,\tau]}} S^{k-i}(\mathbb{X}^\pi)_{[0,u]} \otimes \mathbf{X}_{u,v}^{\otimes(i+1)}.$$

*Proof.* Note that for $k \geq 0$,

$$S^{k+1}(\mathbb{X}^\pi)_{[0,\tau]} = \sum_{[u,v] \in \pi_{[0,\tau]}} \left[ S^{k+1}(\mathbb{X}^\pi)_{[0,v]} - S^{k+1}(\mathbb{X}^\pi)_{[0,u]} \right]$$

$$\overset{(*)}{=} \sum_{[u,v] \in \pi_{[0,\tau]}} \left[ \sum_{i=0}^{k+1} S^{k+1-i}(\mathbb{X}^\pi)_{[0,u]} \otimes \frac{\mathbf{X}_{u,v}^{\otimes i}}{i!} - S^{k+1}(\mathbb{X}^\pi)_{[0,u]} \right]$$

$$= \sum_{[u,v] \in \pi_{[0,\tau]}} \sum_{i=1}^{k+1} S^{k+1-i}(\mathbb{X}^\pi)_{[0,u]} \otimes \frac{\mathbf{X}_{u,v}^{\otimes i}}{i!}$$

$$= \sum_{i=0}^{k} \frac{1}{(1+i)!} \sum_{[u,v] \in \pi_{[0,\tau]}} S^{k-i}(\mathbb{X}^\pi)_{[0,u]} \otimes \mathbf{X}_{u,v}^{\otimes(1+i)},$$

where, in $(*)$, we use Chen's relation and $S(\mathbb{X}^\pi)_{[u,v]} = \exp_\otimes \mathbf{X}_{u,v}$ since $\mathbb{X}^\pi$ is linear over $[u, v] \in \pi$. $\square$

### B.1.1. PROOF OF THEOREM 2.8 UNDER *(i)*, *(iii)* OR *(iv)*

Denote by $\{\pi_n, \ n \geq 1\}$ the signature-defining sequence of refining partitions of the interval $[0, T]$. Without loss of generality, we can consider $\{\pi_n, \ n \geq 1\}$ to be such that $\pi_{n+1}$ is obtained from $\pi_n$ by adding at most one refinement in each sub-interval, i.e., for each $[s, t] \in \pi_n$, either $[s, t] \in \pi_{n+1}$ or $[s, u], [u, t] \in \pi_{n+1}$, for $u \in (s, t)$. If not, one can consider a super-sequence satisfying this property and then pass to the original subsequence.

In the following, for any $n \geq 1$ and $[s, t] \in \pi_n$, denote by $\pi_{n,[s,t]}$ the restriction of $\pi_n$ to $[s, t]$ and, abusing notation slightly, $S(\mathbb{X}^{\pi_n})_{[s,t]} = S(\mathbb{X}^{\pi_{n,[s,t]}})_{[s,t]}$.

Let $[\tau_0, \tau_1] \in \pi_N$, for $N \geq 1$, and note that, for any $k \geq 2$ and $n \geq N$, we can write

$$S^k(\mathbb{X}^{\pi_{n+1}})_{[\tau_0, \tau_1]} - S^k(\mathbb{X}^{\pi_n})_{[\tau_0, \tau_1]} = \sum_{[s,t] \in \pi_{n,[\tau_0, \tau_1]}} \left[ S^k(\mathbb{X}^{\pi_{n,t}})_{[\tau_0, \tau_1]} - S^k(\mathbb{X}^{\pi_{n,s}})_{[\tau_0, \tau_1]} \right], \tag{17}$$

where the partitions $\pi_{n,s}$ are defined as $\pi_{n,s} = \pi_{n+1,[0,s]} \cup \pi_{n,[s,T]}$, i.e., for each $[s, t] \in \pi_n$, the partitions $\pi_{n,s}$ and $\pi_{n,t}$ differ by at most one point $u \in (s, t)$. Using Chen's relation and the definition of the tensor product, we can write for each $[s, t] \in \pi_n$ with refinement $u \in (s, t)$,

$$S^k(\mathbb{X}^{\pi_{n,t}})_{[\tau_0, \tau_1]} - S^k(\mathbb{X}^{\pi_{n,s}})_{[\tau_0, \tau_1]}$$
$$= \sum_{\substack{i_1, i_2, i_3 \geq 0 \\ i_1 + i_2 + i_3 = k}} S^{i_1}(\mathbb{X}^{\pi_{n+1}})_{[\tau_0, s]} \otimes \left[ S^{i_2}(\mathbb{X}^{\pi_{n+1}})_{[s,t]} - S^{i_2}(\mathbb{X}^{\pi_n})_{[s,t]} \right] \otimes S^{i_3}(\mathbb{X}^{\pi_n})_{[t, \tau_1]}. \tag{18}$$

Note that, for $i_2 \in \{0, 1\}$,

$$S^{i_2}(\mathbb{X}^{\pi_{n+1}})_{[s,t]} - S^{i_2}(\mathbb{X}^{\pi_n})_{[s,t]} = 0,$$

and applying again Chen's relation when $i_2 \geq 2$ yields

$$S^{i_2}(\mathbb{X}^{\pi_{n+1}})_{[s,t]} = \sum_{j=0}^{i_2} S^j(\mathbb{X}^{\pi_{n+1}})_{[s,u]} \otimes S^{i_2-j}(\mathbb{X}^{\pi_{n+1}})_{[u,t]} = \frac{1}{i_2!} \sum_{j=0}^{i_2} \binom{i_2}{j} \mathbf{X}_{s,u}^{\otimes j} \otimes \mathbf{X}_{u,t}^{\otimes(i_2-j)},$$

where we used the fact that, if $\mathbb{Y}$ is linear over $[s,t]$, then $S(\mathbb{Y})_{[s,t]} = \exp_\otimes \mathbf{Y}_{s,t}$, which also implies

$$S^{i_2}(\mathbb{X}^{\pi_n})_{[s,t]} = \frac{\mathbf{X}_{s,t}^{\otimes i_2}}{i_2!} = \frac{(\mathbf{X}_{s,u} + \mathbf{X}_{u,t})^{\otimes i_2}}{i_2!} = \frac{1}{i_2!} \sum_{\mathcal{I} \in \{0,1\}^{i_2}} \bigotimes_{i \in \mathcal{I}} \left( \mathbf{X}_{s,u}^{\otimes i} \otimes \mathbf{X}_{u,t}^{\otimes(1-i)} \right),$$

denoting by $\mathcal{I} \in \{0,1\}^{i_2}$ a binary number of length $i_2$ with $|\mathcal{I}| = \sum_{i \in \mathcal{I}} i$ and recalling that $\mathbf{x}^{\otimes 0} = 1, \mathbf{x}^{\otimes 1} = \mathbf{x}$, for any $\mathbf{x} \in \mathbb{R}^d$. We hence have that

$$S^{i_2}(\mathbb{X}^{\pi_{n+1}})_{[s,t]} - S^{i_2}(\mathbb{X}^{\pi_n})_{[s,t]} = \sum_{\mathcal{I} \in \{0,1\}^{i_2}} C_{\mathcal{I}} \bigotimes_{i \in \mathcal{I}} \left( \mathbf{X}_{s,u}^{\otimes i} \otimes \mathbf{X}_{u,t}^{\otimes(1-i)} \right),$$

where for $\mathcal{I} \in \{0,1\}^{i_2}$,

$$C_{\mathcal{I}} = \begin{cases} \frac{1}{i_2!} \left[ \binom{i_2}{|\mathcal{I}|} - 1 \right], & \text{if } \mathcal{I} = (1, \ldots, 1, 0, \ldots, 0), \\ -\frac{1}{i_2!}, & \text{otherwise.} \end{cases}$$

Plugging this into Equation (17) via (18) and noting that $C_{\mathcal{I}} = 0$ for $\mathcal{I} \in \{(0, \ldots, 0), (1, \ldots, 1)\}$, we can write, for any $N \geq 1, [\tau_0, \tau_1] \in \pi_N, n \geq N$ and $k \geq 2$,

$$S^k(\mathbb{X}^{\pi_{n+1}})_{[\tau_0,\tau_1]} - S^k(\mathbb{X}^{\pi_n})_{[\tau_0,\tau_1]}$$
$$= \sum_{\substack{[s,t] \in \pi_{n,[\tau_0,\tau_1]} \\ u \in (s,t)}} \sum_{\substack{i_1,i_3 \geq 0, i_2 \geq 2 \\ i_1+i_2+i_3=k}} \sum_{\substack{\mathcal{I} \in \{0,1\}^{i_2} \\ \mathcal{I} \neq (0,\ldots,0),(1,\ldots,1)}} C_{\mathcal{I}}\, S^{i_1}(\mathbb{X}^{\pi_{n+1}})_{[\tau_0,s]} \otimes \bigotimes_{i \in \mathcal{I}} \left( \mathbf{X}_{s,u}^{\otimes i} \otimes \mathbf{X}_{u,t}^{\otimes(1-i)} \right) \otimes S^{i_3}(\mathbb{X}^{\pi_n})_{[t,\tau_1]}.$$

We now proceed inductively to show that, for any $i \in \{1, \ldots, k\}$ and any $[\tau_0, \tau_1] \in \pi_N$ with $N \geq 1$, the sequence $\{S^i(\mathbb{X}^{\pi_n})_{[\tau_0,\tau_1]}, n \geq N\}$ converges in $L^{mk/i}$ with rate $\mathcal{O}(\sum_{n' \geq n} |\pi_{n'}|^\epsilon)$ and

$$\sup_{N \geq 1} \sup_{[\tau_0,\tau_1] \in \pi_N} \|S^i(\mathbb{X}^{\pi_N})_{[\tau_0,\tau_1]}\|_{L^{mk/i}} < \infty. \tag{19}$$

$k' = 1$. Note that for $[\tau_0, \tau_1] \in \pi_N$ with $N \geq 1$ one has $S^1(\mathbb{X}^{\pi_n})_{[\tau_0,\tau_1]} = \mathbf{X}_{\tau_0,\tau_1}$, for all $n \geq N$, and

$$\|\mathbf{X}_{\tau_0,\tau_1}\|_{L^{mk}} \lesssim |\tau_1 - \tau_0|^\alpha \leq T^\alpha < \infty,$$

by Assumption (A$\alpha$). Hence $S^1(\mathbb{X})_{[0,T]} = \mathbf{X}_{0,T} \in L^{mk}$ and the statement holds trivially.

Assume the inductive hypothesis holds for all $i \in \{1, \ldots, k'\}$ with $k' \in \{1, \ldots, k-1\}$. Then, for each $[\tau_0, \tau_1] \in \pi_N$ with $N \geq 1$ and $n \geq N$, let

$$\left\| S^{k'+1}(\mathbb{X}^{\pi_{n+1}})_{[\tau_0,\tau_1]} - S^{k'+1}(\mathbb{X}^{\pi_n})_{[\tau_0,\tau_1]} \right\|_{L^{mk/(k'+1)}}$$
$$\leq \sum_{\substack{i_1,i_3 \geq 0, i_2 \geq 2 \\ i_1+i_2+i_3=k'+1}} \sum_{\substack{\mathcal{I} \in \{0,1\}^{i_2} \\ \mathcal{I} \neq (0,\ldots,0),(1,\ldots,1)}} |C_{\mathcal{I}}| \left\| \sum_{\substack{[s,t] \in \pi_{n,[\tau_0,\tau_1]} \\ u \in (s,t)}} Z_{[s,t]}^{\mathcal{I}} \right\|_{L^{mk/(k'+1)}}, \tag{20}$$

where, for each $[\tau_0, \tau_1] \in \pi_N, \pi_n$ with $n \geq N, i_1, i_3 \geq 0, i_2 \geq 2$ with $i_1 + i_2 + i_3 = k' + 1$ and $\mathcal{I} \in \{0,1\}^{i_2}$, we define

$$Z_{[s,t]}^{\mathcal{I}} := S^{i_1}(\mathbb{X}^{\pi_{n+1}})_{[\tau_0,s]} \otimes \bigotimes_{i \in \mathcal{I}} \left( \mathbf{X}_{s,u}^{\otimes i} \otimes \mathbf{X}_{u,t}^{\otimes(1-i)} \right) \otimes S^{i_3}(\mathbb{X}^{\pi_n})_{[t,\tau_1]}, \quad [s,t] \in \pi_{n,[\tau_0,\tau_1]} \text{ with } u \in (s,t),$$

keeping only the dependence on $\mathcal{I}$ for notational convenience. Note that, by applying Lemma B.2, the inductive hypothesis and Assumption (A$\alpha$),

$$\left\| Z_{[s,t]}^{\mathcal{I}} \right\|_{L^{mk/(k'+1)}} \lesssim \left\| S^{i_1}(\mathbb{X}^{\pi_{n+1}})_{[\tau_0,s]} \right\|_{L^{mk/i_1}} \|\mathbf{X}_{s,u}\|_{L^{mk}}^{|\mathcal{I}|} \|\mathbf{X}_{u,t}\|_{L^{mk}}^{i_2-|\mathcal{I}|} \left\| S^{i_3}(\mathbb{X}^{\pi_n})_{[t,\tau_1]} \right\|_{L^{mk/i_3}} \lesssim |t-s|^{i_2\alpha},$$

and, hence, each $Z_{[s,t]}^{\mathcal{I}} \in L^{mk/(k'+1)}$. Moreover, by a simple application of the triangle inequality,

$$\left\| \sum_{\substack{[s,t]\in\pi_{n,[\tau_0,\tau_1]} \\ u\in(s,t)}} Z_{[s,t]}^{\mathcal{I}} \right\|_{L^{mk/(k'+1)}} \lesssim \sum_{\substack{[s,t]\in\pi_{n,[\tau_0,\tau_1]} \\ u\in(s,t)}} |t-s|^{i_2\alpha}. \tag{21}$$

**Assumption** $(i)$   Hence, if $\alpha > 1/2$, we have for each $[\tau_0,\tau_1] \in \pi_N$, $\pi_n$ with $n \geq N$, $i_1,i_3 \geq 0, i_2 \geq 2$ with $i_1 + i_2 + i_3 = k'+1$ and $\mathcal{I} \in \{0,1\}^{i_2}$,

$$\left\| \sum_{\substack{[s,t]\in\pi_{n,[\tau_0,\tau_1]} \\ u\in(s,t)}} Z_{[s,t]}^{\mathcal{I}} \right\|_{L^{mk/(k'+1)}} \lesssim |\pi_n|^{2\alpha-1}|\tau_1-\tau_0|. \tag{22}$$

**Assumption** $(iii)$   If $\alpha \in (1/3,1/2]$ note that if $\mathcal{I}$ is such that $i_2 \geq 3$, then

$$\left\| \sum_{\substack{[s,t]\in\pi_{n,[\tau_0,\tau_1]} \\ u\in(s,t)}} Z_{[s,t]}^{\mathcal{I}} \right\|_{L^{mk/(k'+1)}} \lesssim |\pi_n|^{3\alpha-1}|\tau_1-\tau_0|, \tag{23}$$

but if $i_2 = 2$ then the bound (21) is not strong enough. We can instead apply Lemma B.1 with the filtration $\{\mathcal{G}_{[s,t]}, [s,t] \in \pi_n\}$ defined by

$$\mathcal{G}_{[s,t]} := \mathcal{F}_s \vee \sigma(\mathbf{X}_{v,w}, [v,w] \in \pi_{n,[t,\tau]}),$$

by noting that each $Z_{[v,w]}^{\mathcal{I}}$ with $w \leq s$ is $\mathcal{G}_{[s,t]}$-measurable and $mk/(k'+1) \geq 2$ for all $k'+1 \leq k$. This implies

$$\left\| \sum_{\substack{[s,t]\in\pi_{n,[\tau_0,\tau_1]} \\ u\in(s,t)}} Z_{[s,t]}^{\mathcal{I}} \right\|_{L^{mk/(k'+1)}} \leq \sum_{\substack{[s,t]\in\pi_{n,[\tau_0,\tau_1]} \\ u\in(s,t)}} \left\| \mathbb{E}_{\mathcal{G}_{[s,t]}}[Z_{[s,t]}^{\mathcal{I}}] \right\|_{L^{mk/(k'+1)}} + \left( \sum_{\substack{[s,t]\in\pi_{n,[\tau_0,\tau_1]} \\ u\in(s,t)}} \|Z_{[s,t]}^{\mathcal{I}}\|_{L^{mk/(k'+1)}}^2 \right)^{1/2}$$

$$\leq \sum_{\substack{[s,t]\in\pi_{n,[\tau_0,\tau_1]} \\ u\in(s,t)}} |t-s|^\beta + \left( \sum_{\substack{[s,t]\in\pi_{n,[\tau_0,\tau_1]} \\ u\in(s,t)}} |t-s|^{4\alpha} \right)^{1/2}$$

$$\leq |\pi_n|^{(\beta-1)\wedge(2\alpha-1/2)} \left( |\tau_1-\tau_0| + |\tau_1-\tau_0|^{1/2} \right), \tag{24}$$

where we used the fact that for $\mathcal{I} \in \{(0,1),(1,0)\}$,

$$\left\| \mathbb{E}_{\mathcal{G}_{[s,t]}}[Z_{[s,t]}^{\mathcal{I}}] \right\|_{L^{mk/(k'+1)}}$$

$$\stackrel{(i)}{=} \left\| S^{i_1}(\mathbb{X}^{\pi_{n+1}})_{[\tau_0,s]} \otimes \mathbb{E}_{\mathcal{G}_{[s,t]}} \left[ \bigotimes_{i\in\mathcal{I}} \left( \mathbf{X}_{s,u}^{\otimes i} \otimes \mathbf{X}_{u,t}^{\otimes(1-i)} \right) \right] \otimes S^{i_3}(\mathbb{X}^{\pi_n})_{[t,\tau_1]} \right\|_{L^{mk/(k'+1)}}$$

$$\stackrel{(ii)}{\leq} \left\| S^{i_1}(\mathbb{X}^{\pi_{n+1}})_{[\tau_0,s]} \right\|_{L^{mk/i_1}} \left\| \mathbb{E}_{\mathcal{G}_{[s,t]}} \left[ \bigotimes_{i\in\mathcal{I}} \left( \mathbf{X}_{s,u}^{\otimes i} \otimes \mathbf{X}_{u,t}^{\otimes(1-i)} \right) \right] \right\|_{L^{mk/2}} \left\| S^{i_3}(\mathbb{X}^{\pi_n})_{[t,\tau_1]} \right\|_{L^{mk/i_3}}$$

$$\stackrel{(iii)}{\lesssim} \left\| \mathbb{E}_{\mathcal{G}_{[s,t]}}\left[ \mathbf{X}_{s,u} \otimes \mathbf{X}_{u,t} \right] \right\|_{L^{mk/2}}$$

$$\stackrel{(iv)}{\lesssim} \left\| \mathbb{E}_{\mathcal{F}_{0,s}\vee\mathcal{F}_{t,T}}\left[ \mathbf{X}_{s,u} \otimes \mathbf{X}_{u,t} \right] \right\|_{L^{mk/2}}$$

$$\overset{(v)}{\lesssim} |t-s|^\beta,$$

by using in $(i)$ measurability of $S^{i_1}(\mathbb{X}^{\pi_{n+1}})_{[\tau_0,s]}$ and $S^{i_3}(\mathbb{X}^{\pi_n})_{[t,\tau_1]}$ with respect to $\mathcal{G}_{[s,t]}$, in $(ii)$ Hölder inequality for tensors Lemma B.2, in $(iii)$ the inductive assumption (19) and the fact that $\|\mathbf{A} \otimes \mathbf{B}\| = \|\mathbf{B} \otimes \mathbf{A}\|$ for any $\mathbf{A}, \mathbf{B} \in \mathbb{R}^d$, in $(iv)$ the tower property and the contractive property of conditional expectation applied to $\mathcal{G}_{[s,t]} \subseteq \mathcal{F}_{0,s} \vee \mathcal{F}_{t,T}$ and in $(v)$ Assumption (A$\beta$). Combining bound (24) when $i_2 = 2$ and bound (23) when $i_2 \geq 3$ with $\alpha \in (1/3, 1/2)$, it follows that for each $[\tau_0, \tau_1] \in \pi_N$, $\pi_n$ with $n \geq N$, $i_1, i_3 \geq 0$, $i_2 \geq 2$ with $i_1 + i_2 + i_3 = k' + 1$ and $\mathcal{I} \in \{0,1\}^{i_2}$,

$$\left\| \sum_{\substack{[s,t] \in \pi_{n,[\tau_0,\tau_1]} \\ u \in (s,t)}} Z^{\mathcal{I}}_{[s,t]} \right\|_{L^{mk/(k'+1)}} \lesssim |\pi_n|^{(\beta-1) \wedge (3\alpha-1)} |\tau_1 - \tau_0|. \tag{25}$$

**Assumption** $(iv)$  A similar reasoning can be applied when $\alpha \in (1/4, 1/3)$ (and $k \geq 3$) by considering the cases $i_2 \geq 4$, $i_2 = 3$ and $i_2 = 2$ separately. The case $i_2 \geq 4$ follows directly from (21), the case $i_2 = 2$ follows from (24) and the case $i_2 = 3$ can be shown in the same way as $i_2 = 2$ with the only difference being that we require Assumption (A$\gamma$) to show that, for $\mathcal{I} \in \{(0,0,1), (0,1,0), (1,0,0)\}$,

$$\left\| \mathbb{E}_{\mathcal{G}_{[s,t]}}[Z^{\mathcal{I}}_{[s,t]}] \right\|_{L^{mk/(k'+1)}} \lesssim \left\| \mathbb{E}_{\mathcal{F}_{0,s} \vee \mathcal{F}_{t,T}} \left[ \mathbf{X}_{s,u} \otimes \mathbf{X}^{\otimes 2}_{u,t} \right] \right\|_{L^{mk/3}} \lesssim |t-s|^\gamma,$$

and, for $\mathcal{I} \in \{(0,1,1), (1,0,1), (1,1,0)\}$,

$$\left\| \mathbb{E}_{\mathcal{G}_{[s,t]}}[Z^{\mathcal{I}}_{[s,t]}] \right\|_{L^{mk/(k'+1)}} \lesssim \left\| \mathbb{E}_{\mathcal{F}_{0,s} \vee \mathcal{F}_{t,T}} \left[ \mathbf{X}^{\otimes 2}_{s,u} \otimes \mathbf{X}_{u,t} \right] \right\|_{L^{mk/3}} \lesssim |t-s|^\gamma,$$

so that applying again Lemma B.1,

$$\left\| \sum_{\substack{[s,t] \in \pi_{n,[\tau_0,\tau_1]} \\ u \in (s,t)}} Z^{\mathcal{I}}_{[s,t]} \right\|_{L^{mk/(k'+1)}} \leq \sum_{\substack{[s,t] \in \pi_{n,[\tau_0,\tau_1]} \\ u \in (s,t)}} \left\| \mathbb{E}_{\mathcal{G}_{[s,t]}}[Z^{\mathcal{I}}_{[s,t]}] \right\|_{L^{mk/(k'+1)}} + \left( \sum_{\substack{[s,t] \in \pi_{n,[\tau_0,\tau_1]} \\ u \in (s,t)}} \| Z^{\mathcal{I}}_{[s,t]} \|^2_{L^{mk/(k'+1)}} \right)^{1/2}$$

$$\leq \sum_{\substack{[s,t] \in \pi_{n,[\tau_0,\tau_1]} \\ u \in (s,t)}} |t-s|^\gamma + \left( \sum_{\substack{[s,t] \in \pi_{n,[\tau_0,\tau_1]} \\ u \in (s,t)}} |t-s|^{6\alpha} \right)^{1/2}$$

$$\leq |\pi_n|^{(\gamma-1) \wedge (3\alpha-1/2)} \left( |\tau_1 - \tau_0| + |\tau_1 - \tau_0|^{1/2} \right). \tag{26}$$

Combining the cases $i_2 = 2$, $i_2 = 3$ and $i_2 \geq 4$ when $\alpha \in (1/4, 1/3)$ yields, for each $[\tau_0, \tau_1] \in \pi_N$, $\pi_n$ with $n \geq N$, $i_1, i_3 \geq 0$, $i_2 \geq 2$ with $i_1 + i_2 + i_3 = k' + 1$ and $\mathcal{I} \in \{0,1\}^{i_2}$,

$$\left\| \sum_{\substack{[s,t] \in \pi_{n,[\tau_0,\tau_1]} \\ u \in (s,t)}} Z^{\mathcal{I}}_{[s,t]} \right\|_{L^{mk/(k'+1)}} \lesssim |\pi_n|^{(\beta-1) \wedge (\gamma-1) \wedge (4\alpha-1)} |\tau_1 - \tau_0|. \tag{27}$$

Defining $\epsilon$ as in Equation (9), we can plug bounds (22), (25) and (27) into Equation (20) to deduce that

$$\left\| S^{k'+1}(\mathbb{X}^{\pi_{n+1}})_{[\tau_0,\tau_1]} - S^{k'+1}(\mathbb{X}^{\pi_n})_{[\tau_0,\tau_1]} \right\|_{L^{mk/(k'+1)}} \lesssim |\tau_1 - \tau_0| |\pi_n|^\epsilon,$$

and, hence, under the assumption that $\sum_{n \geq 1} |\pi_n|^\epsilon < \infty$, for any $[\tau_0, \tau_1] \in \pi_N$ with $N \geq 1$, the sequence $\{S^{k'+1}(\mathbb{X}^{\pi_n})_{[\tau_0,\tau_1]}, \quad n \geq N\}$ is Cauchy in $L^{mk/(k'+1)}$. Since $L^{mk/(k'+1)}$ is a Banach space the sequence converges in $L^{mk/(k'+1)}$ to $S^{k'+1}(\mathbb{X})_{[\tau_0,\tau_1]} \in L^{mk/(k'+1)}$ (by uniqueness of limits) with rate

$$\left\| S^{k'+1}(\mathbb{X})_{[\tau_0,\tau_1]} - S^{k'+1}(\mathbb{X}^{\pi_n})_{[\tau_0,\tau_1]} \right\|_{L^{mk/(k'+1)}}$$

$$\leq \sum_{n'\geq n} \left\| S^{k'+1}(\mathbb{X}^{\pi_{n'+1}})_{[\tau_0,\tau_1]} - S^{k'+1}(\mathbb{X}^{\pi_{n'}})_{[\tau_0,\tau_1]} \right\|_{L^{mk/(k'+1)}} \lesssim |\tau_1 - \tau_0| \sum_{n'\geq n} |\pi_{n'}|^\epsilon.$$

And, to complete the inductive step for $k'+1$, note that for all $N\geq 1$ and $[\tau_0,\tau_1]\in\pi_N$,

$$\left\| S^{k'+1}(\mathbb{X}^{\pi_N})_{[\tau_0,\tau_1]} \right\|_{L^{mk/(k'+1)}} \lesssim \left\| S^{k'+1}(\mathbb{X})_{[\tau_0,\tau_1]} \right\|_{L^{mk/(k'+1)}} + |\tau_1 - \tau_0| \sum_{n'\geq N} |\pi_{n'}|^\epsilon$$

$$\lesssim \left\| S^{k'+1}(\mathbb{X})_{[\tau_0,\tau_1]} \right\|_{L^{mk/(k'+1)}} + T \sum_{n'\geq 1} |\pi_{n'}|^\epsilon.$$

$\square$

### B.1.2. PROOF OF THEOREM 2.8 UNDER *(ii)*

In what follows we shall simplify notation and denote $\mathbb{E}_{\mathcal{F}_{0,t}}$ by $\mathbb{E}_t$.

Denote by $\{\pi_n,\ n\geq 1\}$ the signature-defining sequence of refining partitions of the interval $[0,T]$. Without loss of generality, we can consider $\{\pi_n,\ n\geq 1\}$ to be such that $\pi_{n+1}$ is obtained from $\pi_n$ by adding at most one refinement in each sub-interval, i.e. for each $[s,t]\in\pi_n$ either $[s,t]\in\pi_{n+1}$ or $[s,u],[u,t]\in\pi_{n+1}$ for $u\in(s,t)$. If not, one can consider a super-sequence satisfying this property and then pass to the original subsequence.

We start by showing inductively that, for any $i\in\{1,\dots,k\}$,

$$\sup_{n\geq 1}\sup_{\tau\in\pi_n} \|S^i(\mathbb{X}^{\pi_n})_{[0,\tau]}\|_{L^{mk/i}} < \infty. \tag{28}$$

Note that the case $i=1$ is trivial since, for any $n\geq 1$ and $\tau\in\pi_n$, by (A$\alpha$)

$$\|S^1(\mathbb{X}^{\pi_n})_{[0,\tau]}\|_{L^{mk}} = \|\mathbf{X}_{0,\tau}\|_{L^{mk}} \lesssim \tau^\alpha \lesssim T^\alpha.$$

Next, for the inductive step, assume that (28) holds for all $i\in\{1,\dots k'\}$ with $k'\leq k-1$. Then, by using Lemma B.3, we can bound for any $n\geq 1$ and $\tau\in\pi_n$,

$$\|S^{k'+1}(\mathbb{X}^{\pi_n})_{[0,\tau]}\|_{L^{mk/(k'+1)}}$$

$$\overset{(i)}{\leq} \sum_{i=0}^{k'} \frac{1}{(1+i)!} \left\| \sum_{[u,v]\in\pi_{n,[0,\tau]}} S^{k'-i}(\mathbb{X}^{\pi_n})_{[0,u]} \otimes \mathbf{X}_{u,v}^{\otimes(i+1)} \right\|_{L^{mk/(k'+1)}}$$

$$\overset{(ii)}{\leq} \sum_{[u,v]\in\pi_{n,[0,\tau]}} \left\| S^{k'}(\mathbb{X}^{\pi_n})_{[0,u]} \otimes \mathbb{E}_u[\mathbf{X}_{u,v}] \right\|_{L^{mk/(k'+1)}}$$

$$\quad + \left( \sum_{[u,v]\in\pi_{n,[0,\tau]}} \left\| S^{k'}(\mathbb{X}^{\pi_n})_{[0,u]} \otimes \mathbf{X}_{u,v} \right\|_{L^{mk/(k'+1)}}^2 \right)^{1/2}$$

$$\quad + \sum_{i=1}^{k'} \frac{1}{(1+i)!} \sum_{[u,v]\in\pi_{n,[0,\tau]}} \left\| S^{k'-i}(\mathbb{X}^{\pi_n})_{[0,u]} \otimes \mathbf{X}_{u,v}^{\otimes(i+1)} \right\|_{L^{mk/(k'+1)}}$$

$$\overset{(iii)}{\lesssim} \sum_{[u,v]\in\pi_{n,[0,\tau]}} \|\mathbb{E}_u[\mathbf{X}_{u,v}]\|_{L^{mk}} + \left( \sum_{[u,v]\in\pi_{n,[0,\tau]}} \|\mathbf{X}_{u,v}\|_{L^p}^2 \right)^{1/2} + \sum_{i=1}^{k'} \sum_{[u,v]\in\pi_{n,[0,\tau]}} \|\mathbf{X}_{u,v}\|_{L^{mk}}^{i+1}$$

$$\overset{(iv)}{\lesssim} \sum_{[u,v]\in\pi_{n,[0,\tau]}} |v-u|^\delta + \left( \sum_{[u,v]\in\pi_{n,[0,\tau]}} |v-u|^{2\alpha} \right)^{1/2} + \sum_{i=1}^{k'} \sum_{[u,v]\in\pi_{n,[0,\tau]}} |v-u|^{(i+1)\alpha}$$

$$\overset{(v)}{\lesssim} \tau + \sqrt{\tau} + \tau \lesssim T + \sqrt{T},$$

where in $(i)$ we applied the triangle inequality, in $(ii)$ we bounded the $i = 0$ term by applying Lemma B.1 to the sequence of random variables

$$Z_{[u,v]} := S^{k'}(\mathbb{X}^{\pi_n})_{[0,u]} \otimes \mathbf{X}_{u,v} \in L^{mk/(k'+1)},$$

with filtration $\{\mathcal{F}_u, [u,v] \in \pi_{n,[0,\tau]}\}$ and we bounded the $i = 1, \dots, k'$ terms by applying the triangle inequality, in $(iii)$ we applied the Hölder inequality given in Lemma B.2 and the inductive hypothesis Equation (28) for all signature levels up to $k'$, in $(iv)$ we used Assumptions (A$\alpha$) and (A$\delta$).

Proceeding again by induction, we will show the conclusion of the theorem holds by proving the stronger statement: For each $i \in \{1, \dots, k\}$, for all $N \geq 1$, $\tau \in \pi_N$ and $n \geq N$,

$$\|S^i(\mathbb{X}^{\pi_{n+1}})_{[0,\tau]} - S^i(\mathbb{X}^{\pi_n})_{[0,\tau]}\|_{L^{mk/i}} \lesssim |\pi_n|^\epsilon. \tag{29}$$

The case $k' = 1$ is again trivial since, for all $N \geq 1$, $\tau \in \pi_N$ and $n \geq N$, $S^1(\mathbb{X}^{\pi_{n+1}})_{[0,\tau]} = \mathbf{X}_{0,\tau}$, and hence

$$\|S^1(\mathbb{X}^{\pi_{n+1}})_{[0,\tau]} - S^1(\mathbb{X}^{\pi_n})_{[0,\tau]}\|_{L^{mk}} = 0.$$

For the inductive step, assume Equation (29) holds for all $i \in \{1, \dots, k'\}$ with $k' \leq k$. Fix $N \geq 1$, $\tau \in \pi_N$ and $n \geq N$, then we can write the telescoping sum

$$S^{k'+1}(\mathbb{X}^{\pi_{n+1}})_{[0,\tau]} - S^{k'+1}(\mathbb{X}^{\pi_n})_{[0,\tau]} = \sum_{[s,t]\in\pi_{n,[0,\tau]}} \left[S^{k'+1}(\mathbb{X}^{\pi_{n,t}})_{[0,\tau]} - S^{k'+1}(\mathbb{X}^{\pi_{n,s}})_{[0,\tau]}\right], \tag{30}$$

where the partitions $\pi_{n,s}$ are defined as $\pi_{n,s} = \pi_{n+1,[0,s]} \cup \pi_{n,[s,T]}$, i.e. for each $[s,t] \in \pi_n$, the partitions $\pi_{n,s}$ and $\pi_{n,t}$ differ by at most one point $u \in (s,t)$. Note that, for each $[s,t] \in \pi_n$ with refinement $u \in (s,t)$, we can apply Lemma B.3 to write

$$S^{k'+1}(\mathbb{X}^{\pi_{n,t}})_{[0,\tau]} - S^{k'+1}(\mathbb{X}^{\pi_{n,s}})_{[0,\tau]}$$

$$= \sum_{i=0}^{k'} \frac{1}{(1+i)!}\left\{ S^{k'-i}(\mathbb{X}^{\pi_{n,t}})_{[0,s]} \otimes \mathbf{X}_{s,u}^{\otimes(1+i)} + S^{k'-i}(\mathbb{X}^{\pi_{n,t}})_{[0,u]} \otimes \mathbf{X}_{u,t}^{\otimes(1+i)} - S^{k'-i}(\mathbb{X}^{\pi_{n,s}})_{[0,s]} \otimes \mathbf{X}_{s,t}^{\otimes(1+i)}\right.$$

$$\left. + \sum_{[v,w]\in\pi_{n,[t,\tau]}} \left[S^{k'-i}(\mathbb{X}^{\pi_{n,t}})_{[0,v]} - S^{k'-i}(\mathbb{X}^{\pi_{n,s}})_{[0,v]}\right] \otimes \mathbf{X}_{v,w}^{\otimes(1+i)}\right\}$$

$$= \sum_{i=0}^{k'} \frac{1}{(1+i)!}\left\{ S^{k'-i}(\mathbb{X}^{\pi_{n+1}})_{[0,s]} \otimes \left[\mathbf{X}_{s,u}^{\otimes(1+i)} - \mathbf{X}_{s,t}^{\otimes(1+i)}\right] + \sum_{j=0}^{k'-i} S^{k'-i-j}(\mathbb{X}^{\pi_{n+1}})_{[0,s]} \otimes \frac{\mathbf{X}_{s,u}^{\otimes j}}{j!} \otimes \mathbf{X}_{u,t}^{\otimes(1+i)}\right.$$

$$\left. + \sum_{[v,w]\in\pi_{n,[t,\tau]}} \left[S^{k'-i}(\mathbb{X}^{\pi_{n,t}})_{[0,v]} - S^{k'-i}(\mathbb{X}^{\pi_{n,s}})_{[0,v]}\right] \otimes \mathbf{X}_{v,w}^{\otimes(1+i)}\right\}$$

$$= \sum_{i=0}^{k'} \frac{1}{(1+i)!}\left\{ S^{k'-i}(\mathbb{X}^{\pi_{n+1}})_{[0,s]} \otimes \left[\mathbf{X}_{s,u}^{\otimes(1+i)} + \mathbf{X}_{u,t}^{\otimes(1+i)} - \mathbf{X}_{s,t}^{\otimes(1+i)}\right]\right.$$

$$+ \sum_{j=0}^{k'-i-1} \frac{1}{(1+j)!} S^{k'-i-j-1}(\mathbb{X}^{\pi_{n+1}})_{[0,s]} \otimes \mathbf{X}_{s,u}^{\otimes(1+j)} \otimes \mathbf{X}_{u,t}^{\otimes(1+i)}$$

$$\left. + \sum_{[v,w]\in\pi_{n,[t,\tau]}} \left[S^{k'-i}(\mathbb{X}^{\pi_{n,t}})_{[0,v]} - S^{k'-i}(\mathbb{X}^{\pi_{n,s}})_{[0,v]}\right] \otimes \mathbf{X}_{v,w}^{\otimes(1+i)}\right\}$$

$$= \sum_{i=0}^{k'} \frac{1}{(1+i)!}\left\{ - S^{k'-i}(\mathbb{X}^{\pi_{n+1}})_{[0,s]} \otimes \sum_{\substack{\mathcal{I}\in\{0,1\}^{1+i} \\ \mathcal{I}\neq(0,\dots,0),(1,\dots,1)}} \bigotimes_{l\in\mathcal{I}} \left(\mathbf{X}_{s,u}^{\otimes l} \otimes \mathbf{X}_{u,t}^{\otimes(1-l)}\right)\right.$$

$$+ \sum_{j=0}^{k'-i-1} \frac{1}{(1+j)!} S^{k'-i-j-1}(\mathbb{X}^{\pi_{n+1}})_{[0,s]} \otimes \mathbf{X}_{s,u}^{\otimes(1+j)} \otimes \mathbf{X}_{u,t}^{\otimes(1+i)}$$

$$+ \sum_{[v,w]\in\pi_{n,[t,\tau]}} \left[ S^{k'-i}(\mathbb{X}^{\pi_{n,t}})_{[0,v]} - S^{k'-i}(\mathbb{X}^{\pi_{n,s}})_{[0,v]} \right] \otimes \mathbf{X}_{v,w}^{\otimes(1+i)} \Bigg\},$$

by noting that, for all $[v,w] \in \pi_n$ with $w \le s$, $S(\mathbb{X}^{\pi_{n,t}})_{[0,v]} = S(\mathbb{X}^{\pi_{n,s}})_{[0,v]} = S(\mathbb{X}^{\pi_{n+1}})_{[0,v]}$ and when applying Chen's relation to $S(\mathbb{X}^{\pi_{n,t}})_{[0,u]}$, setting $S(\mathbb{X}^{\pi_{n,t}})_{[s,u]} = \exp_\otimes \mathbf{X}_{s,u}$ since $\mathbb{X}^{\pi_{n,t}}$ is linear over $[s,u] \in \pi_{n,t}$. Plugging this expression into Equation (30) and exchanging the orders of the summations we obtain

$$S^{k'+1}(\mathbb{X}^{\pi_{n+1}})_{[0,\tau]} - S^{k'+1}(\mathbb{X}^{\pi_n})_{[0,\tau]}$$

$$= \sum_{i=0}^{k'} \frac{1}{(1+i)!} \Bigg\{ - \sum_{\substack{\mathcal{I}\in\{0,1\}^{1+i} \\ \mathcal{I}\ne(0,\dots,0),(1,\dots,1)}} \sum_{\substack{[s,t]\in\pi_{n,[0,\tau]} \\ u\in(s,t)}} S^{k'-i}(\mathbb{X}^{\pi_{n+1}})_{[0,s]} \otimes \bigotimes_{l\in\mathcal{I}} \left( \mathbf{X}_{s,u}^{\otimes l} \otimes \mathbf{X}_{u,t}^{\otimes(1-l)} \right)$$

$$+ \sum_{j=0}^{k'-i-1} \frac{1}{(1+j)!} \sum_{\substack{[s,t]\in\pi_{n,[0,\tau]} \\ u\in(s,t)}} S^{k'-i-j-1}(\mathbb{X}^{\pi_{n+1}})_{[0,s]} \otimes \mathbf{X}_{s,u}^{\otimes(1+j)} \otimes \mathbf{X}_{u,t}^{\otimes(1+i)}$$

$$+ \sum_{[v,w]\in\pi_{n,[0,\tau]}} \left( \sum_{[s,t]\in\pi_{n,[0,v]}} \left[ S^{k'-i}(\mathbb{X}^{\pi_{n,t}})_{[0,v]} - S^{k'-i}(\mathbb{X}^{\pi_{n,s}})_{[0,v]} \right] \right) \otimes \mathbf{X}_{v,w}^{\otimes(1+i)} \Bigg\}$$

$$= \sum_{i=0}^{k'} \frac{1}{(1+i)!} \Bigg\{ - \sum_{\substack{\mathcal{I}\in\{0,1\}^{1+i} \\ \mathcal{I}\ne(0,\dots,0),(1,\dots,1)}} \sum_{\substack{[s,t]\in\pi_{n,[0,\tau]} \\ u\in(s,t)}} S^{k'-i}(\mathbb{X}^{\pi_{n+1}})_{[0,s]} \otimes \bigotimes_{l\in\mathcal{I}} \left( \mathbf{X}_{s,u}^{\otimes l} \otimes \mathbf{X}_{u,t}^{\otimes(1-l)} \right)$$

$$+ \sum_{j=0}^{k'-i-1} \frac{1}{(1+j)!} \sum_{\substack{[s,t]\in\pi_{n,[0,\tau]} \\ u\in(s,t)}} S^{k'-i-j-1}(\mathbb{X}^{\pi_{n+1}})_{[0,s]} \otimes \mathbf{X}_{s,u}^{\otimes(1+j)} \otimes \mathbf{X}_{u,t}^{\otimes(1+i)}$$

$$+ \sum_{[v,w]\in\pi_{n,[0,\tau]}} \left[ S^{k'-i}(\mathbb{X}^{\pi_{n+1}})_{[0,v]} - S^{k'-i}(\mathbb{X}^{\pi_n})_{[0,v]} \right] \otimes \mathbf{X}_{v,w}^{\otimes(1+i)} \Bigg\}$$

$$= \sum_{i=0}^{k'} \frac{1}{(1+i)!} \Bigg\{ \sum_{\substack{\mathcal{I}\in\{0,1\}^{1+i} \\ \mathcal{I}\ne(0,\dots,0),(1,\dots,1)}} \sum_{\substack{[s,t]\in\pi_{n,[0,\tau]} \\ u\in(s,t)}} Z_{[s,t]}^{1,\mathcal{I}} + \sum_{j=0}^{k'-i-1} \frac{1}{(1+j)!} \sum_{\substack{[s,t]\in\pi_{n,[0,\tau]} \\ u\in(s,t)}} Z_{[s,t]}^{2,i,j} + \sum_{[v,w]\in\pi_{n,[0,\tau]}} Z_{[v,w]}^{3,i} \Bigg\}.$$

We thus proceed to bound each of the summation terms over $\pi_{n,[0,\tau]}$ using Lemma B.1 and Assumptions (A$\alpha$) and (A$\delta$).

Let $\mathcal{I} \in \{0,1\}^{1+i}$ with $\mathcal{I} \ne (0,\dots,0),(1,\dots,1)$ with $i\in\{1,\dots,k'\}$. Note that for all $[s,t]\in\pi_{n,[0,\tau]}$

$$\|Z_{[s,t]}^{1,\mathcal{I}}\|_{L^{mk/(k'+1)}} \le \left\|S^{k'-i}(\mathbb{X}^{\pi_{n+1}})_{[0,s]}\right\|_{L^{mk/(k'-i)}} \|\mathbf{X}_{s,u}\|_{L^{mk}}^{|\mathcal{I}|} \|\mathbf{X}_{u,t}\|_{L^{mk}}^{1+i-|\mathcal{I}|} \lesssim |t-s|^{(1+i)\alpha}, \tag{31}$$

by applying Lemma B.2, the uniform bound (28) and Assumption (A$\alpha$), hence $Z_{[s,t]}^{1,\mathcal{I}} \in L^{mk/(k'+1)}$. When $i=1$ we can thus apply Lemma B.1 to the sequence $\{Z_{[s,t]}^{1,\mathcal{I}},\ [s,t]\in\pi_{n,[0,\tau]},\ u\in(s,t)\}$ with filtration $\{\mathcal{F}_u,\ [s,t]\in\pi_{n,[0,\tau]},\ u\in(s,t)\}$ to bound

$$\left\| \sum_{\substack{[s,t]\in\pi_{n,[0,\tau]} \\ u\in(s,t)}} Z_{[s,t]}^{1,\mathcal{I}} \right\|_{L^{mk/(k'+1)}} \le \sum_{\substack{[s,t]\in\pi_{n,[0,\tau]} \\ u\in(s,t)}} \|\mathbb{E}_u[Z_{[s,t]}^{1,\mathcal{I}}]\|_{L^{mk/(k'+1)}} + \left( \sum_{\substack{[s,t]\in\pi_{n,[0,\tau]} \\ u\in(s,t)}} \|Z_{[s,t]}^{1,\mathcal{I}}\|_{L^{mk/(k'+1)}}^2 \right)^{1/2}$$

$$\lesssim \sum_{\substack{[s,t]\in\pi_{n,[0,\tau]} \\ u\in(s,t)}} |t-s|^{\alpha+\delta} + \left( \sum_{\substack{[s,t]\in\pi_{n,[0,\tau]} \\ u\in(s,t)}} |t-s|^{4\alpha} \right)^{1/2}$$

$$\lesssim |\pi_n|^{\alpha+\delta-1}\tau + |\pi_n|^{2\alpha-1/2}\sqrt{\tau} \lesssim |\pi_n|^{\epsilon}(\tau+\sqrt{\tau}), \tag{32}$$

where we used Equation (31) with $i = 1$ and

$$\|\mathbb{E}_u[Z_{[s,t]}^{1,\mathcal{I}}]\|_{L^{mk/(k'+1)}} = \left\|S^{k'-i}(\mathbb{X}^{\pi_{n+1}})_{[0,s]} \otimes \mathbf{X}_{s,u} \otimes \mathbb{E}_u[\mathbf{X}_{u,t}]\right\|_{L^{mk/(k'+1)}}$$
$$\leq \left\|S^{k'-1}(\mathbb{X}^{\pi_{n+1}})_{[0,s]}\right\|_{L^{mk/(k'-1)}} \|\mathbf{X}_{s,u}\|_{L^{mk}} \|\mathbb{E}_u[\mathbf{X}_{u,t}]\|_{L^{mk}} \lesssim |t-s|^{\alpha+\delta},$$

by applying Lemma B.2, the uniform bound (28) and Assumptions (A$\alpha$) and (A$\delta$). When $i \geq 2$ we can directly apply the triangle inequality and Equation (31) to bound

$$\left\|\sum_{\substack{[s,t]\in\pi_{n,[0,\tau]} \\ u\in(s,t)}} Z_{[s,t]}^{1,\mathcal{I}}\right\|_{L^{mk/(k'+1)}} \leq \sum_{\substack{[s,t]\in\pi_{n,[0,\tau]} \\ u\in(s,t)}} \|Z_{[s,t]}^{1,\mathcal{I}}\|_{L^{mk/(k'+1)}}$$
$$\lesssim \sum_{\substack{[s,t]\in\pi_{n,[0,\tau]} \\ u\in(s,t)}} |t-s|^{(1+i)\alpha} \lesssim |\pi_n|^{3\alpha-1}\tau \lesssim |\pi_n|^\epsilon \tau. \tag{33}$$

Next, let $i \in \{0, \ldots, k'\}$ and $j \in \{0, \ldots, k'-i-1\}$. We can proceed exactly as for $Z_{[s,t]}^{1,\mathcal{I}}$ (applying Lemma B.1 when $i = j = 0$ or the triangle inequality when $i + j \geq 1$) to show that under Assumptions (A$\alpha$) and (A$\delta$),

$$\left\|\sum_{\substack{[s,t]\in\pi_{n,[0,\tau]} \\ u\in(s,t)}} Z_{[s,t]}^{2,i,j}\right\|_{L^{mk/(k'+1)}} \lesssim |\pi_n|^\epsilon(\tau + \sqrt{\tau}). \tag{34}$$

Finally, let $i \in \{0, \ldots, k'\}$. We proceed in a similar way as for $Z_{[s,t]}^{1,\mathcal{I}}$ and $Z_{[s,t]}^{2,i,j}$ but using the inductive hypothesis (29) instead of the bound (28). Note that for all $[s,t] \in \pi_{n,[0,\tau]}$

$$\|Z_{[s,t]}^{3,i}\|_{L^{mk/(k'+1)}} \leq \left\|S^{k'-i}(\mathbb{X}^{\pi_{n+1}})_{[0,s]} - S^{k'-i}(\mathbb{X}^{\pi_n})_{[0,s]}\right\|_{L^{mk/(k'-i)}} \|\mathbf{X}_{s,t}\|_{L^{mk}}^{1+i} \lesssim |\pi_n|^\epsilon |t-s|^{(1+i)\alpha}, \tag{35}$$

by applying Lemma B.2, the inductive hypothesis (29) and Assumption (A$\alpha$), hence $Z_{[s,t]}^{3,i} \in L^{mk/(k'+1)}$. When $i = 0$ we can hence apply Lemma B.1 to the sequence $\{Z_{[s,t]}^{3,i}, [s,t] \in \pi_{n,[0,\tau]}\}$ with filtration $\{\mathcal{F}_s, [s,t] \in \pi_{n,[0,\tau]}\}$ to bound

$$\left\|\sum_{[s,t]\in\pi_{n,[0,\tau]}} Z_{[s,t]}^{3,i}\right\|_{L^{mk/(k'+1)}} \leq \sum_{[s,t]\in\pi_{n,[0,\tau]}} \|\mathbb{E}_s[Z_{[s,t]}^{3,i}]\|_{L^{mk/(k'+1)}} + \left(\sum_{[s,t]\in\pi_{n,[0,\tau]}} \|Z_{[s,t]}^{3,i}\|_{L^{mk/(k'+1)}}^2\right)^{1/2}$$
$$\lesssim \sum_{[s,t]\in\pi_{n,[0,\tau]}} |\pi_n|^\epsilon |t-s|^\delta + \left(\sum_{[s,t]\in\pi_{n,[0,\tau]}} |\pi_n|^{2\epsilon} |t-s|^{2\alpha}\right)^{1/2}$$
$$\lesssim |\pi_n|^\epsilon(\tau + \sqrt{\tau}), \tag{36}$$

where we used Equation (35) with $i = 0$ and

$$\|\mathbb{E}_s[Z_{[s,t]}^{3,i}]\|_{L^{mk/(k'+1)}} = \left\|\left(S^{k'}(\mathbb{X}^{\pi_{n+1}})_{[0,s]} - S^{k'}(\mathbb{X}^{\pi_n})_{[0,s]}\right) \otimes \mathbb{E}_s[\mathbf{X}_{s,t}]\right\|_{L^{mk/(k'+1)}}$$
$$\leq \left\|S^{k'}(\mathbb{X}^{\pi_{n+1}})_{[0,s]} - S^{k'}(\mathbb{X}^{\pi_n})_{[0,s]}\right\|_{L^{mk/k'}} \|\mathbb{E}_s[\mathbf{X}_{s,t}]\|_{L^{mk}} \lesssim |\pi_n|^\epsilon |t-s|^\delta,$$

by applying Lemma B.2, the inductive hypothesis (29) and Assumption (A$\delta$). When $i \geq 1$, we can instead directly apply the triangle inequality and Equation (35) to bound

$$\left\|\sum_{[s,t]\in\pi_{n,[0,\tau]}} Z_{[s,t]}^{3,i}\right\|_{L^{mk/(k'+1)}} \leq \sum_{[s,t]\in\pi_{n,[0,\tau]}} \|Z_{[s,t]}^{3,i}\|_{L^{mk/(k'+1)}} \lesssim \sum_{[s,t]\in\pi_{n,[0,\tau]}} |\pi_n|^\epsilon |t-s|^{(1+i)\alpha} \lesssim |\pi_n|^\epsilon \tau. \tag{37}$$

Combining bounds (32), (33), (34), (36) and (37) yields

$$\left\| S^{k'+1}(\mathbb{X}^{\pi_{n+1}})_{[0,\tau]} - S^{k'+1}(\mathbb{X}^{\pi_n})_{[0,\tau]} \right\|_{L^{mk/(k'+1)}}$$

$$\leq \sum_{i=0}^{k'} \frac{1}{(1+i)!} \Bigg\{ \sum_{\substack{\mathcal{I} \in \{0,1\}^{1+i} \\ \mathcal{I} \neq (0,\ldots,0),(1,\ldots,1)}} \left\| \sum_{\substack{[s,t] \in \pi_{n,[0,\tau]} \\ u \in (s,t)}} Z_{[s,t]}^{1,\mathcal{I}} \right\|_{L^{mk/(k'+1)}}$$

$$+ \sum_{j=0}^{k'-i-1} \frac{1}{(1+j)!} \left\| \sum_{\substack{[s,t] \in \pi_{n,[0,\tau]} \\ u \in (s,t)}} Z_{[s,t]}^{2,i,j} \right\|_{L^{mk/(k'+1)}}$$

$$+ \left\| \sum_{[s,t] \in \pi_{n,[0,\tau]}} Z_{[s,t]}^{3,i} \right\|_{L^{mk/(k'+1)}} \Bigg\}$$

$$\lesssim |\pi_n|^\epsilon (\tau + \sqrt{\tau}) \lesssim |\pi_n|^\epsilon (T + \sqrt{T}),$$

which proves (29) with $i = k' + 1$, completing the inductive step.

Setting $i = k$, $N = 1$ and $\tau = T$ in (29) yields

$$\|S^k(\mathbb{X}^{\pi_{n+1}})_{[0,T]} - S^k(\mathbb{X}^{\pi_n})_{[0,T]}\|_{L^m} \lesssim |\pi_n|^\epsilon, \quad n \geq 1,$$

and hence, assuming $\sum_{n \geq 1} |\pi_n|^\epsilon < \infty$, the sequence $\{S^k(\mathbb{X}^{\pi_n})_{[0,T]}, n \geq\}$ is Cauchy in $L^m$. Since $L^m$ is a Banach space, the sequence converges in $L^m$ to $S^k(\mathbb{X})_{[0,T]} \in L^m$ (by uniqueness of limits) with rate

$$\|S^k(\mathbb{X}^{\pi_n})_{[0,T]} - S^k(\mathbb{X})_{[0,T]}\|_{L^m} \leq \sum_{n' \geq n} \|S^k(\mathbb{X}^{\pi_{n'+1}})_{[0,T]} - S^k(\mathbb{X}^{\pi_{n'}})_{[0,T]}\|_{L^m} \lesssim \sum_{n' \geq n} |\pi_n|^\epsilon.$$

$\square$

*Remark* B.4. When $\{\pi_n, n \geq 1\}$ is a sequence of refining partitions with $\sum_{n \geq 1} |\pi_n|^\epsilon < \infty$, the proof actually yields the following (stronger) result

$$\sup_{\tau \in \pi_n} \|S^k(\mathbb{X}^{\pi_n})_{[0,\tau]} - S^k(\mathbb{X})_{[0,\tau]}\|_{L^m} \lesssim \sum_{n' \geq n} |\pi_{n'}|^\epsilon \to 0, \quad n \to \infty.$$

## B.2. Proof of Theorem 2.10

***Sketch of proof.*** *The proof of this result relies on decomposition (8). We can combine Theorem 2.8 with assumption (11) to show the first term in the decomposition vanishes in $L^m$, for $m > 2$. To show the full consistency result it thus suffices to show the second term in (8) also vanishes in $L^2$, which follows by Birkhoff's ergodic theorem under the stated assumptions. Similarly, for the asymptotic normality result, we combine Theorem 2.8 with assumption (13) to show the first term in the decomposition vanishes in $L^m$ when inflated by $\sqrt{N}$. The asymptotic normality of the second term can then be obtained by a simple application of a central limit theorem (CLT) for dependent random variables.*

Under the assumption that $\{\mathbb{X}^n, n \geq 1\}$ is stationary and $\mathbb{X}^1$ satisfies the assumptions of Theorem 2.8 with $m > 2$, we have that, for each $n \geq 1$,

$$S^{\mathbf{I}}(\mathbb{X}^{n,\pi_{N,n}})_{[0,T]} \xrightarrow{L^m} S^{\mathbf{I}}(\mathbb{X})_{[0,T]}, \quad N \to \infty,$$

with rate $\mathcal{O}(\sum_{N' \geq N} |\pi_{N',n}|^\epsilon)$. And hence

$$\left\| \frac{1}{N} \sum_{n=1}^N \left( S^{\mathbf{I}}(\mathbb{X}^{n,\pi_{N,n}})_{[0,T]} - S^{\mathbf{I}}(\mathbb{X}^n)_{[0,T]} \right) \right\|_{L^m} \lesssim \max_{1 \leq n \leq N} \sum_{N' \geq N} |\pi_{N',n}|^\epsilon \leq \sum_{N' \geq N} |\Pi(N')|^\epsilon,$$

since $|\Pi(N')| = \max_{1 \leq n \leq N'} |\pi_{N',n}|$. Under assumption (11), we have thus established the first term in decomposition (8) vanishes in $L^m$ as $N \to \infty$ and therefore can focus on showing the second term also vanishes. Note that, under the stronger

assumption (13), a similar reasoning can be applied when "blowing up" decomposition (8) by $\sqrt{N}$, and hence it suffices to show the second term, when rescaled by $\sqrt{N}$, converges to a Gaussian random variable to establish the asymptotic normality result.

We first prove the following somewhat technical result. In what follows we abuse notation slightly and write $S^{\mathbf{I}}(\cdot)_{[0,T]}$ to denote $\mathcal{S}^{\mathbf{I}}(\cdot)$.

**Proposition B.5.** *Let* $\mathbb{X} = \{\mathbf{X}_t,\ t \in [0,T]\}$ *denote a canonical geometric stochastic process defined on the probability space* $(\Omega_{[0,T]} = C([0,T];\mathbb{R}^d), \mathcal{B}_{[0,T]}, \mathbb{P}_{[0,T]})$ *by* $\mathbf{X}_t = \omega_{[0,T]}(t)$ *for* $t \in [0,T]$ *and* $\omega_{[0,T]} \in \Omega_{[0,T]}$. *Then for any collection of words* $\mathbf{I}$ *there exists a measurable map* $\mathcal{S}^{\mathbf{I}} : C([0,T];\mathbb{R}^d) \to \mathbb{R}^{|\mathbf{I}|}$ *such that*

$$\mathcal{S}^{\mathbf{I}}(\omega_{[0,T]}) = S^{\mathbf{I}}(\mathbb{X})_{[0,T]}, \quad \mathbb{P}_{[0,T]} - \text{a.s.}$$

*Proof.* By Remark 2.4 every canonical geometric stochastic process has at least one signature-defining sequence of partitions over any $[s,t] \subseteq [0,T]$ given by the sequence $\rho$ in Definition 2.1. Moreover, by passing to a subsequence if necessary, this also guarantees the existence of a sequence of partitions $\rho_*$ along which (5) holds almost surely. Hence, there exists a Borel set $\Omega'_{[0,T]} \in \mathcal{B}_{[0,T]}$ and a sequence of partitions $\rho^*$ with vanishing mesh such that for all $\omega_{[0,T]} \in \Omega'_{[0,T]}$,

$$S^{\mathbf{I}}\left(\omega_{[0,T]}^{\rho_*}\right)_{[0,T]} \to S^{\mathbf{I}}\left(\omega_{[0,T]}\right)_{[0,T]}, \quad |\rho_*| \to 0,$$

and $\mathbb{P}_{[0,T]}(\Omega'_{[0,T]}) = 1$. For every partition $\rho_*$ the map

$$\omega_{[0,T]} \in \Omega'_{[0,T]} \mapsto S^{\mathbf{I}}\left(\omega_{[0,T]}^{\rho_*}\right)_{[0,T]} \in \mathbb{R}^{|\mathbf{I}|},$$

is $\Omega'_{[0,T]} \cap \mathcal{B}_{[0,T]}$-measurable (by measurability of the sums and products of coordinate maps appearing in the discretized signature) and hence also

$$\omega_{[0,T]} \in \Omega'_{[0,T]} \mapsto S^{\mathbf{I}}\left(\omega_{[0,T]}\right)_{[0,T]} \in \mathbb{R}^{|\mathbf{I}|},$$

is $\Omega'_{[0,T]} \cap \mathcal{B}_{[0,T]}$-measurable (by measurability of the pointwise limit of measurable functions). We can hence extend $S^{\mathbf{I}} : \Omega'_{[0,T]} \to \mathbb{R}^{|\mathbf{I}|}$ to a measurable map on the whole of $\Omega_{[0,T]} = C([0,T];\mathbb{R}^d)$. $\qquad\square$

### B.2.1. PROOF OF THEOREM 2.10, CONSISTENCY

The consistency result then follows by a simple application of Birkhoff's ergodic theorem (Kallenberg, 2021, Theorem 25.6). Let $(\Omega = C([0,\infty);\mathbb{R}^d), \mathcal{F} = \mathcal{B}_{[0,\infty)}, \mathbb{P})$ denote the canonical space on which $\{\mathbf{X}_t,\ t \geq 0\}$ is defined, i.e. $\mathbb{P}$ is the law of $\{\mathbf{X}_t,\ t \geq 0\}$. Consider $\{\mathbb{X}^n,\ n \geq 1\}$ as the sequence of $C([0,T];\mathbb{R}^d)$-valued random variables on the space[12] $(C([0,T];\mathbb{R}^d)^\infty, \mathcal{B}_{[0,T]}^\infty, \mathbb{P}_{[0,T]}^\infty)$, where the probability measure $\mathbb{P}_{[0,T]}^\infty$ is obtained by pushing forward $\mathbb{P}$ by the measurable mapping

$$\omega \in \Omega \mapsto \{\omega_{[0,T]}^n,\ n \geq 1\} \in C([0,T];\mathbb{R}^d)^\infty,$$

where

$$\omega_{[0,T]}^n := \{\omega((n-1)T + t) - \omega((n-1)T),\ t \in [0,T]\}.$$

If $\{\mathbb{X}^n,\ n \geq 1\}$ is stationary and $\mathbb{X} = \mathbb{X}^1$ is a canonical geometric stochastic process, then each $\mathbb{X}^n$ is also a canonical geometric stochastic process on $(C([0,T];\mathbb{R}^d), \mathcal{B}_{[0,T]}, \mathbb{P}_* \mathbb{X}^n)$ with signature given by the same measurable mapping $\mathcal{S}^{\mathbf{I}} : C([0,T];\mathbb{R}^d) \to \mathbb{R}^{|\mathbf{I}|}$ given in Proposition B.5. We can then apply Birkhoff's ergodic theorem (Kallenberg, 2021, Theorem 25.6) to conclude

$$\frac{1}{N}\sum_{n=1}^{N} S^{\mathbf{I}}(\mathbb{X}^n)_{[0,T]} \overset{\mathbb{P}-\text{a.s.}}{\to} \mathbb{E}[S^{\mathbf{I}}(\mathbb{X})_{[0,T]}], \quad N \to \infty.$$

$\qquad\square$

---

[12]Recall that if $(E, \mathcal{B}(E))$ is a Borel measurable space and we equip $E^\infty$, i.e. the space of sequences with values in $E$, with the product topology then

$$\mathcal{B}(E^\infty) = \otimes_{n \geq 1} \mathcal{B}(E) := \sigma\left(\cup_{n \geq 1}(\mathcal{B}(E)^n \times E^\infty)\right),$$

as long as $E$ is second-countable.

B.2.2. PROOF OF THEOREM 2.10, ASYMPTOTIC NORMALITY

We apply the dependent central limit theorem (CLT) given in Ibragimov (1962, Theorem 1.7) and extended to the multivariate setting via the Cramér-Wold theorem. To apply this result we require $\{S^{\mathbf{I}}(\mathbb{X}^n)_{[0,T]}, \; n \geq 1\}$ to be stationary with $S^{\mathbf{I}}(\mathbb{X}^n)_{[0,T]} \in L^{2+\zeta}$ and strongly mixing with strong mixing coefficient $\alpha(n), \; n \in \mathbb{N}$ satisfying

$$\sum_{n \geq 1} \alpha(n)^{\zeta/(2+\zeta)} < \infty,$$

so that

$$\Sigma_{\mathbf{I}} = \mathrm{Var}\left(S^{\mathbf{I}}(\mathbb{X}^1)_{[0,T]}\right) + 2\sum_{n \geq 2} \mathrm{Cov}\left(S^{\mathbf{I}}(\mathbb{X}^1)_{[0,T]}, S^{\mathbf{I}}(\mathbb{X}^n)_{[0,T]}\right) < \infty,$$

and, if $\Sigma_{\mathbf{I}}$ is strictly positive definite,

$$\sqrt{N}\left(\frac{1}{N}\sum_{n=1}^N S^{\mathbf{I}}(\mathbb{X}^n)_{[0,T]} - \mathbb{E}[S^{\mathbf{I}}(\mathbb{X})_{[0,T]}]\right) \xrightarrow{\mathcal{L}} \mathcal{N}(0, \Sigma_{\mathbf{I}}), \quad N \to \infty.$$

Note that $S^{\mathbf{I}}(\mathbb{X}^n)_{[0,T]} \in L^{2+\zeta}$ for $\zeta > 0$ is immediately obtained by applying Theorem 2.8 with $m > 2$. By measurability of $S^{\mathbf{I}} : C([0,T]; \mathbb{R}) \to \mathbb{R}$ (Proposition B.5), $\sigma(S^{\mathbf{I}}(\mathbb{X}^n)) \subseteq \sigma(\mathbb{X}^n)$, for all $n \geq 1$, which implies $\{S^{\mathbf{I}}(\mathbb{X}^n)_{[0,T]}, \; n \geq 1\}$ is also strongly mixing with strong mixing coefficient at most $\alpha(n), \; n \in \mathbb{N}$. The assumptions of Theorem 2.10 are thus sufficient to deduce asymptotic normality of the expected signature estimator. $\quad\square$

## B.3. Proof of Corollary 2.12

*Sketch of proof.* In order to show that the CLT result of Theorem 2.10 can be made feasible we need to prove the kernel estimator $\hat{\Sigma}_{\mathbf{I}}^{\Pi(N)}$ is consistent for the long-run covariance matrix $\Sigma_{\mathbf{I}}$. To do so, we first show that the kernelized estimator is consistent for $\Sigma_{\mathbf{I}}$ if, for each $n \geq 0$, the cross-covariance estimator term $\hat{\Sigma}_{\mathbf{I}}^{n,\Pi(N)}$ is consistent for $\Sigma_{\mathbf{I}}^n$ (with a "fast enough" convergence rate). To show consistency of each cross-covariance term, we introduce an auxiliary process $\mathbb{Y}^n$ obtained by appropriately "stitching" the processes $\mathbb{X}^1, \ldots, \mathbb{X}^{1+n}$ together. This choice of $\mathbb{Y}^n$, along with the shuffle property of the expected signature, implies that each term in $\Sigma_{\mathbf{I}}^n$ can be expressed as a combination of terms from the expected signature of $\mathbb{Y}^n$. The rest of the proof is thus devoted to showing that, under the assumptions of this Corollary, the process $\mathbb{Y}^n$ satisfies the conditions of Theorem 2.10.1, ensuring we can consistently estimate its signature terms and, in turn, the entries of $\Sigma_{\mathbf{I}}^n$.

To make the CLT feasible one requires a consistent estimator for the long-run covariance of the sequence of random variables $\{S^{\mathbf{I}}(\mathbb{X}^n)_{[0,T]}, \; n \geq 1\}$,

$$\Sigma_{\mathbf{I}} = \Sigma_{\mathbf{I}}^0 + 2\sum_{n \geq 1} \Sigma_{\mathbf{I}}^n, \quad \text{where} \quad \Sigma_{\mathbf{I}}^n = \mathrm{Cov}\left(S^{\mathbf{I}}(\mathbb{X}^1)_{[0,T]}, S^{\mathbf{I}}(\mathbb{X}^{1+n})_{[0,T]}\right), \; n \geq 0.$$

We consider the *non-overlapping* sample (cross-)covariances of the sequence $\{S^{\mathbf{I}}(\mathbb{X}^{n,\pi_{N,n}})_{[0,T]}, \; n = 1, \ldots, N\}$, i.e. for $|n| \leq N - 1$,

$$\hat{\Sigma}_{\mathbf{I}}^{n,\Pi(N)} = \frac{1}{\lfloor N/(n+1) \rfloor} \sum_{m=1}^{\lfloor N/(n+1) \rfloor} \left(S^{\mathbf{I}}(\mathbb{X}^{\pi_{N,(n+1)m-n}})_{[0,T]} - \hat{\phi}_{\mathbf{I}}^{\Pi(N)}(T)\right)\left(S^{\mathbf{I}}(\mathbb{X}^{\pi_{N,(n+1)m}})_{[0,T]} - \hat{\phi}_{\mathbf{I}}^{\Pi(N)}(T)\right)^{\mathrm{T}},$$

as estimators for $\Sigma_{\mathbf{I}}^n$. Note that, for a fixed observation partition $\Pi(N)$, we are only able to estimate $\Sigma_{\mathbf{I}}^n$ up to $n = N - 1$ with the quality of the estimator decreasing as $n$ increases[13]. A natural choice would thus be to put less weight on $\hat{\Sigma}_{\mathbf{I}}^{n,\Pi(N)}$

---

[13]In this context, one might exploit the available data more efficiently by considering the full sample cross-covariances of the sequence $\{S^{\mathbf{I}}(\mathbb{X}^{n,\pi_{N,n}})_{[0,T]}, \; n = 1, \ldots, N\}$, i.e. for $|n| \leq N - 1$,

$$\frac{1}{N-n} \sum_{m=1}^{N-n} \left(S^{\mathbf{I}}(\mathbb{X}^{\pi_{N,m}})_{[0,T]} - \hat{\phi}_{\mathbf{I}}^{\Pi(N)}(T)\right)\left(S^{\mathbf{I}}(\mathbb{X}^{\pi_{N,m+n}})_{[0,T]} - \hat{\phi}_{\mathbf{I}}^{\Pi(N)}(T)\right)^{\mathrm{T}},$$

with $n$ large than on $\hat{\Sigma}_{\mathbf{I}}^{n,\Pi(N)}$ with small $n$. To do so, one can consider the kernel estimator

$$\hat{\Sigma}_{\mathbf{I}}^{\Pi(N)} = \sum_{n=-(N-1)}^{N-1} k\left(\frac{n}{h_N}\right) \hat{\Sigma}_{\mathbf{I}}^{n,\Pi(N)}, \quad \hat{\Sigma}_{\mathbf{I}}^{-n,\Pi(N)} := \left(\hat{\Sigma}_{\mathbf{I}}^{n,\Pi(N)}\right)^{\mathrm{T}}, \text{ for } n = 1, \ldots, N-1,$$

where $k : \mathbb{R} \to [0,1]$ is a decreasing kernel function continuous at zero with with $k(0) = 1$ and $h_N$ is an appropriately chosen band-width parameter[14]. In what follows, and in the statement of Corollary 2.12, we set $k$ to be the truncation kernel $k(x) = \mathbb{1}_{[-1,1]}(x)$ for simplicity, but other choices, such as the Bartlett kernel, might lead to better finite sample properties. For each $I, J \in \mathbf{I}$, if $\hat{\Sigma}_{I,J}^{n,\Pi(N)}$ is consistent for $\Sigma_{I,J}^n$ in $L^2$ with monotonically decreasing rate $r(M) \to 0$ as the *effective* sample size $M = \lfloor N/(n+1) \rfloor \to \infty$, then

$$\|\hat{\Sigma}_{I,J}^{\Pi(N)} - \Sigma_{I,J}\|_{L^2} \leq \sum_{|n| \leq h_N} \left\|\hat{\Sigma}_{I,J}^{n,\Pi(N)} - \Sigma_{I,J}^n\right\|_{L^2} + \left|\sum_{|n| > h_N} \Sigma_{I,J}^n\right|$$

$$\leq \sum_{|n| \leq h_N} r(\lfloor N/(n+1) \rfloor) + \left|\sum_{|n| > h_N} \Sigma_{I,J}^n\right|.$$

If the band-width $h_N \to \infty$ as $N \to \infty$, then $\Sigma_{\mathbf{I}} < \infty$ ensures the second term vanishes. If, moreover, we set the rate at which $h_N \to \infty$ to be slow enough to ensure also the first term converges to zero, then consistency of the estimators $\hat{\Sigma}_{I,J}^{n,\Pi(N)}$, for $n \geq 0$, is inherited by $\hat{\Sigma}_{I,J}^{\Pi(N)}$. Under the assumption $r(M) \sim M^{-v}$ for $v \in (0,1)$, one can set $h_N = N^{v/2}$. We can hence focus on determining under which conditions, other than those of Theorem 2.10.2, the estimator $\hat{\Sigma}_{I,J}^{n,\Pi(N)}$ is consistent for $\Sigma_{I,J}^n$, for any $n \geq 0$ and $I, J \in \mathbf{I}$.

Note that, we can apply the shuffle identity to write, for $I, J \in \mathbf{I}$,

$$\hat{\Sigma}_{I,J}^{0,\Pi(N)} = \sum_{K \in I \sqcup J} \hat{\phi}_K^{\Pi(N)}(T) - \hat{\phi}_I^{\Pi(N)}(T)\hat{\phi}_J^{\Pi(N)}(T),$$

and show consistency of this estimator for $\Sigma_{I,J}^0$ by applying the consistency result for the estimator of the expected signature terms to $K \in I \sqcup J$ for $I, J \in \mathbf{I}$. We now attempt to apply a similar approach to the cross-covariance terms. To do so, we introduce an $\mathbb{R}^{nd}$-valued auxiliary process[15] defined as $\mathbb{Y}^n = \{\mathbf{Y}_t^n, \ t \in [0, (n+1)T]\}$, where

$$\mathbf{Y}_t^n = \begin{pmatrix} \mathbf{X}_{t \wedge T}^1 \\ \mathbf{X}_{(t-T)^+ \wedge T}^2 \\ \cdots \\ \mathbf{X}_{(t-nT)^+ \wedge T}^{n+1} \end{pmatrix}, \ t \in [0, (n+1)T] \iff \mathbf{Y}_t^n = \begin{pmatrix} \mathbf{X}_T^1 \\ \cdots \\ \mathbf{X}_T^i \\ \mathbf{X}_s^{i+1} \\ 0 \\ \cdots \\ 0 \end{pmatrix}, \ t = iT + s, s \in [0,T], 0 \leq i \leq n.$$

By construction, we have that, for any two words $I, J \in \mathbf{I}$ over the letters $\{1, \ldots, d\}$,

$$S^I(\mathbb{X}^1)_{[0,T]} = S^I(\mathbb{Y}^n)_{[0,(n+1)T]} \quad \text{and} \quad S^J(\mathbb{X}^{n+1})_{[0,T]} = S^{nd+J}(\mathbb{Y}^n)_{[0,(n+1)T]}.$$

I.e. we have re-written the two signature components of $\mathbb{X}^1$ and $\mathbb{X}^n$ over $[0,T]$ as time-overlapping signature components of $\mathbb{Y}^n$ over $[0, (n+1)T]$. We can thus apply the shuffle product to deduce

$$S^I(\mathbb{X}^1)_{[0,T]} S^J(\mathbb{X}^n)_{[0,T]} = S^I(\mathbb{Y}^n)_{[0,(n+1)T]} S^{nd+J}(\mathbb{Y}^n)_{[0,(n+1)T]} = \sum_{K \in I \sqcup (nd+J)} S^K(\mathbb{Y}^n)_{[0,(n+1)T]}.$$

---

as estimators for $\Sigma_{\mathbf{I}}^n$ with $n \geq 0$. The reason for using the non-overlapping sample covariance will become apparent once we introduce the process $\mathbb{Y}^n$, which will be used to show consistency of the estimator $\hat{\Sigma}_{\mathbf{I}}^{n,\Pi(N)}$ for $\Sigma_{\mathbf{I}}^n$ by re-applying Theorem 2.10.1. To show consistency of the full sample covariance estimator we would instead require the generalization of such result for time-overlapping expected signature estimators.

[14]In the presence of conditional heteroskedasticity such estimators for the long-run covariance are known as Heteroskedasticity and Autocorrelation Consistent (HAC) estimators (Newey & West, 1987). Note that decomposing the estimation of $2\Sigma_{\mathbf{I}}^n$ into the sum of $\hat{\Sigma}_{\mathbf{I}}^{n,\Pi(N)}$ and its transpose ensures the resulting long-run covariance estimator $\hat{\Sigma}_{\mathbf{I}}^{\Pi(N)}$ is symmetric.

[15]We would like to thank Nicola Muca Cirone for suggesting this clever trick.

We can thus re-write the $(I, J)$-th entry of $\Sigma_{\mathbf{I}}^n$ as

$$
\begin{aligned}
\Sigma_{I,J}^n &= \mathbb{E}\left[S^I(\mathbb{X}^1)_{[0,T]}S^J(\mathbb{X}^{1+n})_{[0,T]}\right] - \mathbb{E}\left[S^I(\mathbb{X}^1)_{[0,T]}\right]\mathbb{E}\left[S^J(\mathbb{X}^{1+n})_{[0,T]}\right] \\
&= \mathbb{E}\left[S^I(\mathbb{Y}^n)_{[0,(n+1)T]}S^{nd+J}(\mathbb{Y}^n)_{[0,(n+1)T]}\right] - \mathbb{E}\left[S^I(\mathbb{X})_{[0,T]}\right]\mathbb{E}\left[S^J(\mathbb{X})_{[0,T]}\right] \\
&= \sum_{K \in I \sqcup (nd+J)} \mathbb{E}\left[S^K(\mathbb{Y}^n)_{[0,(n+1)T]}\right] - \mathbb{E}\left[S^I(\mathbb{X})_{[0,T]}\right]\mathbb{E}\left[S^J(\mathbb{X})_{[0,T]}\right] \\
&= \sum_{K \in I \sqcup (nd+J)} \psi_K((n+1)T) - \phi_I(T)\phi_J(T),
\end{aligned}
$$

and estimate it by

$$
\hat{\Sigma}_{I,J}^{n,\Pi(N)} = \sum_{K \in I \sqcup (nd+J)} \hat{\psi}_K^{\Pi(N;n)}((n+1)T) - \hat{\phi}_I^{\Pi(N)}(T)\hat{\phi}_J^{\Pi(N)}(T),
$$

where, setting $M = \lfloor N/(n+1)\rfloor$,

$$
\hat{\psi}_K^{\Pi(N;n)}((n+1)T) := \frac{1}{M}\sum_{m=1}^M S^K(\mathbb{Y}^{\pi_{N,m;n}})_{[0,(n+1)T]},
$$

and

$$
\Pi(N;n) = \pi_{N,1;n} \cup \ldots \cup ((M-1)(n+1)T + \pi_{N,M;n}),
$$

where for each $m = 1, \ldots, M$,

$$
\pi_{N,m;n} = \pi_{N,(m-1)(n+1)+1} \cup \ldots \cup (nT + \pi_{N,m(n+1)}),
$$

partitions $[0, (n+1)T]$ and

$$
|\Pi(N;n)| := \max_{1 \le m \le M} |\pi_{N,m;n}| = \max_{1 \le m \le M} \max_{1 \le i \le (n+1)} |\pi_{N,(m-1)(n+1)+i}| \le |\Pi(N)|.
$$

In order to apply the consistency result of Theorem 2.10 to $\hat{\psi}_K^{\Pi(N;n)}(nT)$, we need to understand under which additional conditions on $\{\mathbf{X}_t,\ t \ge 0\}$, other than those of Theorem 2.10, the process $\{\mathbf{Y}_t^n,\ t \ge 0\}$, defined by

$$
\mathbf{Y}_t^n = \sum_{m \ge 1}\begin{pmatrix} \mathbf{X}_{(t-(m-1)T)^+ \wedge T}^m \\ \mathbf{X}_{(t-mT)^+ \wedge T}^{m+1} \\ \cdots \\ \mathbf{X}_{(t-(m+n-1)T)^+ \wedge T}^{m+n} \end{pmatrix},\ t \ge 0,
$$

satisfies itself the assumptions of Theorem 2.10. Note that canonical geometricity of the stochastic process $\mathbb{X}$ (with lift-defining sequence of partitions $\rho$, $|\rho| \to 0$) and stationarity of $\{\mathbb{X}^m,\ m \ge 1\}$ imply that each $\mathbb{X}^m$ is a canonical geometric stochastic process and, hence, so is $\mathbb{Y}^n$ (with lift-defining sequence of partitions $\rho^n = \rho \cup (T+\rho) \cup \cdots \cup (nT+\rho)$, $|\rho| \to 0$). Moreover, stationarity and ergodicity of $\{\mathbb{X}^m,\ m \ge 1\}$ imply stationarity and ergodicity of $\{(\mathbb{X}^m, \ldots, \mathbb{X}^{m+n}),\ m \ge 1\}$ and hence of the measurable transformation $\{\mathbb{Y}^{n,m} = f(\mathbb{X}^m, \ldots, \mathbb{X}^{m+n}),\ m \ge 1\}$.

For fixed $n \ge 0$ and $m \ge 1$, each $\pi_{\cdot,m;n} = \{\pi_{N,m;n}, \lfloor N/(n+1)\rfloor \ge m\}$ is a signature-defining sequence for $\mathbb{Y}^n = \mathbb{Y}^{n,1}$ over $[0, (n+1)T]$ since each $\pi_{\cdot,n} = \{\pi_{N,n},\ N \ge n\}$ is a signature-defining sequence for $\mathbb{X}$ over $[0, T]$. Moreover, by construction of $\Pi(N;n)$, $|\Pi(N;n)| \le |\Pi(N)| \to 0$ whenever $|\Pi(N)| \to 0$ as $N \to \infty$.

It thus remains to check that $\mathbb{Y}^n = \mathbb{Y}^{n,1}$ satisfies (A$\alpha$) and (A$\delta$) over $[0, (n+1)T]$ with $\alpha \ge 1/2$, $\delta \ge 1$ and $p > 4k$ with $k = \max_{I \in \mathbf{I}} |I|$. Clearly, as we are now considering the process over an arbitrarily long time span $[0, (n+1)T]$, we will require $\{\mathbf{X}_t,\ t \ge 0\}$ to satisfy (A$\alpha$) and (A$\delta$) with $\alpha \ge 1/2$, $\delta \ge 1$ and $p > 4k$ not just over $[0, T]$ but for any $0 \le s \le t$ with $|t - s| \le T$. This condition is automatically fulfilled when $\{\mathbf{X}_t,\ t \ge 0\}$ is stationary (or, more generally, has jointly stationary increments) and satisfies (A$\alpha$) and (A$\delta$) over $[0, T]$ with $\alpha \ge 1/2$, $\delta \ge 1$ and $p > 4k$ with $k = \max_{I \in \mathbf{I}} |I|$. It turns out that these conditions, only slightly stronger than those already required in Theorem 2.10 for the asymptotic normality of $\hat{\phi}_{\mathbf{I}}^{\Pi(N)}(T)$, are sufficient to show that $\hat{\psi}_K^{\Pi(N;n)}((n+1)T)$ is consistent for $\psi_K((n+1)T)$, and thus $\hat{\Sigma}_{I,J}^{n,\Pi(N)}$ is consistent for $\Sigma_{I,J}^n$ for any $n \ge 0$ and $I, J \in \mathbf{I}$.

To show (A$\alpha$) holds for $\mathbb{Y}^n$ with the same $\alpha \geq 1/2$ as $\{\mathbf{X}_t, \ t \geq 0\}$ note that for $0 \leq s \leq t \leq (n+1)T$,

$$
\begin{aligned}
\|\mathbf{Y}^n_{s,t}\|_{L^p} &\leq \sum_{i=0}^{n} \|\mathbf{X}^{i+1}_{(s-iT)^+ \wedge T, (t-iT)^+ \wedge T}\|_{L^p} \\
&\leq \sum_{i=0}^{n} \|\mathbf{X}_{iT+(s-iT)^+ \wedge T, iT+(t-iT)^+ \wedge T}\|_{L^p} \\
&\lesssim \sum_{i=0}^{n} |(t-iT)^+ \wedge T - (s-iT)^+ \wedge T|^\alpha \lesssim |t-s|^\alpha.
\end{aligned}
$$

Next, to show that (A$\delta$) holds for $\mathbb{Y}^n$ with the same $\delta \geq 1$ as $\{\mathbf{X}_t, \ t \geq 0\}$. Note that for $0 \leq s \leq t \leq (n+1)T$,

$$
\begin{aligned}
\|\mathbb{E}_s[\mathbf{Y}^n_{s,t}]\|_{L^p} &\leq \sum_{i=0}^{n} \|\mathbb{E}_s[\mathbf{X}^{i+1}_{(s-iT)^+ \wedge T, (t-iT)^+ \wedge T}]\|_{L^p} \\
&\leq \sum_{i=0}^{n} \|\mathbb{E}_s[\mathbf{X}_{iT+(s-iT)^+ \wedge T, iT+(t-iT)^+ \wedge T}]\|_{L^p} \\
&\leq \sum_{i=0}^{n} \mathbb{1}_{[0,(i+1)T]}(s) \|\mathbb{E}_{iT+(s-iT)^+ \wedge T}[\mathbf{X}_{iT+(s-iT)^+ \wedge T, iT+(t-iT)^+ \wedge T}]\|_{L^p} \\
&\leq \sum_{i=0}^{n} |(t-iT)^+ \wedge T - (s-iT)^+ \wedge T|^\delta \lesssim |t-s|^\delta.
\end{aligned}
$$

## B.4. Proof of Proposition 2.13

### B.4.1. PROOF OF PROPOSITION 2.13, STATIONARY IMPLICATIONS

The first implication follows directly from the definitions of stationarity and joint stationarity of the increments. It remains to show the latter implies stationarity of $\{\mathbb{X}^n, \ n \geq 1\}$, i.e. Equation (14) implies Equation (6). Under joint stationarity of the increments, for all $k \in \mathbb{N}$, $n_1, \ldots, n_k \in \mathbb{N}$ and $n \geq 0$,

$$
\begin{aligned}
\mathbb{P}(\mathbb{X}^{n_1} &\in A_1, \ldots, \mathbb{X}^{n_k} \in A_k) \\
&= \mathbb{P}(\mathbf{X}_{(n_1-1)T, (n_1-1)T+t_1^1} \in B_1^1, \ldots, \mathbf{X}_{(n_k-1)T, (n_k-1)T+t_{m_k}^k} \in B_{m_k}^k), \\
&= \mathbb{P}(\mathbf{X}_{(n_1+n-1)T, (n_1+n-1)T+t_1^1} \in B_1^1, \ldots, \mathbf{X}_{(n_k+n-1)T, (n_k+n-1)T+t_{m_k}^k} \in B_{m_k}^k), \\
&= \mathbb{P}(\mathbb{X}^{n_1+n} \in A_1, \ldots, \mathbb{X}^{n_k+n} \in A_k),
\end{aligned}
$$

for all $A_1, \ldots, A_k$ cylinder sets of the form

$$
A_j = \{\omega_{[0,T]} \in C([0,T]; \mathbb{R}^d) : \omega(t_1^j) \in B_1^j, \ldots, \omega(t_{m_j}^j) \in B_{m_j}^j\},
$$

for $B_1^j, \ldots, B_{m_j}^j \in \mathcal{B}(\mathbb{R}^d)$, $t_1, \ldots, t_{m_j}^j \in [0,T]$, $m_j \geq 1$. Noting that the collection of the sets $A_1 \times \ldots \times A_k$ is a semi-ring that generates[16] the $\sigma$-algebra $\mathcal{B}^k_{[0,T]}$, we can apply Caratheodory's extension theorem to conclude that (6) holds. □

---

[16]Recall that for a set $I \subseteq \mathbb{R}_+$, the Borel $\sigma$-algebra $\mathcal{B}_I := \mathcal{B}(C(I; \mathbb{R}^d))$ (w.r.t. the topology induced by $\|\cdot\|_\infty$) can be equivalently defined by

$$
\mathcal{B}_I = \sigma\left(\omega \in C(I; \mathbb{R}^d) : \omega(t_1) \in A_1, \ldots, \omega(t_n) \in A_n, \ t_1, \ldots, t_n \in I, \ A_1, \ldots, A_n \in \mathcal{B}(\mathbb{R}^d), \ n \geq 1\right).
$$

Moreover, if $(E, \mathcal{B}(E))$ is a Borel measurable space and we equip $E^k$ with the product topology, then

$$
\mathcal{B}(E^k) = \mathcal{B}(E)^k := \sigma(A_1 \times \cdots \times A_k, \ A_1, \ldots, A_k \in \mathcal{B}(E)) = \sigma(A_1 \times \cdots \times A_k, \ A_1, \ldots, A_k \in \mathcal{G}),
$$

where $\mathcal{G}$ is a generating collection for $\mathcal{B}(E)$, i.e. $\mathcal{B}(E) = \sigma(\mathcal{G})$.

*Remark* B.6. One might expect a similar statement as Proposition 2.13 for ergodicity, but the following counterexample shows that

$$\{\mathbf{X}_t, t \geq 0\} \text{ is ergodic } \not\Longrightarrow \{\mathbb{X}^n, n \geq 1\} \text{ is ergodic.}$$

Let $\{X_t, t \geq 0\}$ be an $\mathbb{R}$-valued process such that

$$X_t = \sin\left(\frac{\pi t}{T} + \phi\right), \quad t \geq 0,$$

and $\phi \sim U([0, 2\pi])$, inducing a probability measure $\mathbb{P}_{[0,T]}^\infty$ on $(\Omega_{[0,T]}^\infty, \mathcal{B}_{[0,T]}^\infty)$ where $\Omega_{[0,T]} = (C([0,T], \mathbb{R})$. The process is stationary and ergodic. Stationarity of $\{X_t, t \geq 0\}$ implies stationarity of $\{\mathbb{X}^n, n \geq 1\}$ by Proposition 2.13. But the shift-invariant set

$$I = \prod_{n \geq 1} \{\omega_{[0,T]}^n \in I_{\geq 0} \cup I_{\leq 0}\} \in \mathcal{B}_{[0,T]}^\infty,$$

where $I_{\geq 0}, I_{\leq 0} \in \mathcal{B}_{[0,T]}$ are given by

$$I_{\geq 0} := \{\omega_{[0,T]} : \omega_{[0,T]}(t) \geq 0, \ \forall t \in [0,T]\},$$
$$I_{\leq 0} := \{\omega_{[0,T]} : \omega_{[0,T]}(t) \leq 0, \ \forall t \in [0,T]\},$$

has measure $\mathbb{P}_{[0,T]}^\infty(I) = \mathbb{P}(\phi \in [\pi/2, \pi] \cup [3\pi/2, 2\pi]) = 1/2 \notin \{0, 1\}$ and hence $\{\mathbb{X}^n, n \geq 1\}$ is not ergodic.

### B.4.2. PROOF OF PROPOSITION 2.13, STRONG MIXING IMPLICATIONS

We start by showing that strong mixing of $\{\mathbf{X}_t, t \geq 0\}$ with strong mixing coefficient $\alpha(s)$, $s \in \mathbb{R}_+$ implies strong mixing of the progressive increment process $\{\mathbf{X}_t^T, t \geq 0\}$ where

$$\mathbf{X}_t^T := \mathbf{X}_{\lfloor t/T \rfloor T, t}, \quad t \geq 0,$$

with strong mixing coefficient $\alpha'(s) \leq \alpha(s - 2T)$, $s \geq 2T$. This follows immediately from the definition of strong mixing and the fact that for $t \geq 0$,

$$\sigma(\mathbf{X}_u^T, u \leq t) \subseteq \sigma(\mathbf{X}_u, u \leq t),$$

and for $s \geq 2T$,

$$\sigma(\mathbf{X}_u^T, u \geq t + s) \subseteq \sigma(\mathbf{X}_u, u \geq \lfloor (t+s)/T \rfloor T) \subseteq \sigma(\mathbf{X}_u, u \geq t + (s - 2T)).$$

Next, we show that strong mixing of $\{\mathbf{X}_t^T, t \geq 0\}$ with strong mixing coefficient $\alpha'(s)$, $s \in \mathbb{R}_+$ implies strong mixing of $\{\mathbb{X}^n, n \geq 1\}$ with strong mixing coefficient $\alpha''(n) \leq \alpha'((n-1)T)$, $n \in \mathbb{N}$, and thus $\alpha''(n) \leq \alpha((n-3)T)$, $n \geq 3$. Let

$$\mathcal{X}_a^b := \sigma(\mathbf{X}_u^T, u \in [a, b]),$$

for $a, b \in \overline{\mathbb{R}}_+$ with $a \leq b$ and let

$$\mathcal{S}_m^n := \sigma(\mathbb{X}^l, m \leq l \leq n),$$

for $m, n \in \overline{\mathbb{N}}$ with $m \leq n$. Then, by definition, for each $n \in \mathbb{N}$,

$$\mathcal{X}_{(n-1)T}^{nT} = \sigma(\mathbf{X}_{(n-1)T, (n-1)T+t}, t \in [0, T]) = \mathcal{S}_n^n.$$

Thus for any $m, n \in \overline{\mathbb{N}}$ with $m \leq n$

$$\mathcal{S}_m^n = \mathcal{X}_{(m-1)T}^{nT}.$$

Letting $k \in \mathbb{N}$ and $A \in \mathcal{S}_{-\infty}^k$, $B \in \mathcal{S}_{k+n}^\infty$ we thus have

$$|\mathbb{P}(A \cap B) - \mathbb{P}(A)\mathbb{P}(B)| \leq \alpha'((n-1)T) \to 0, \quad k \to \infty.$$

$\square$

## B.5. Proof of Theorem 2.14

***Sketch of proof.*** *As in the proof of Theorem 2.10, c.f. Appendix B.2, we can apply the in-fill result given in Theorem 2.8 to show the first term in decomposition (8) vanishes. For the second term, we show the sequence of random variables $\{S^{\mathbf{I}}(\mathbb{X}^n)_{[0,T]}, \ n \geq 1\}$ satisfies a (weak) law of large numbers by verifying the auto-covariance decay condition (38). For each fixed lag $n \geq 1$, we start by bounding the auto-covariance of lagged discretized signatures $\{S^{\mathbf{I}}(\mathbb{X}^{\rho,n})_{[0,T]}, \ n \geq 1\}$. This step crucially relies on the Gaussian assumption when using Isserlis' theorem to compute the expectation of the product of (arbitrarily many) path increments. The required auto-covariance decay condition is then obtained in the in-fill limit along a sequence of signature-defining partitions $\rho$ by an application of Theorem 2.8.*

Note that for any $n \in \mathbb{N}$, $0 \leq s_i \leq t_i$ for $i = 1, \ldots, n$ and $t \geq 0$

$$(\mathbf{X}_{t+s_1,t+t_1}, \ldots, \mathbf{X}_{t+s_n,t+t_n}) \overset{\mathcal{L}}{=} (\mathbf{X}_{s_1,t_1}, \ldots, \mathbf{X}_{s_n,t_n}),$$

since both vectors are normally distributed with means

$$\mathbb{E}[\mathbf{X}_{t+s_i,t+t_i}] = \mathbb{E}[\mathbf{X}_{s_i,t_i}] = 0,$$

for $i = 1, \ldots, n$ and covariances

$$\mathrm{Cov}(\mathbf{X}_{t+s_i,t+t_i}, \mathbf{X}_{t+s_j,t+t_j}) = \mathrm{Cov}(\mathbf{X}_{s_i,t_i}, \mathbf{X}_{s_j,t_j}) = C(|t_i - s_i|, |s_j - t_i|, |t_j - s_j|),$$

for $i, j = 1, \ldots, n$. This implies $\{\mathbf{X}_t, \ t \geq 0\}$ has jointly stationary increments and hence, by Proposition 2.13, $\{\mathbb{X}^n, \ n \geq 1\}$ is stationary. By Proposition B.5 the sequence $\{S^{\mathbf{I}}(\mathbb{X}^n), \ n \geq 1\}$ is defined $\mathbb{P}$-a.s. and is stationary.

Note that by Appendix E.1, $\mathbb{X}$ satisfies (A$\alpha$) for any $p \geq 2$. When $\alpha = 1/2$, it also satisfies (A$\delta$) with $\delta \geq 1$ for all $p \geq 2$ by assumption. We can thus apply Theorem 2.8 to obtain an $L^2$ in-fill asymptotic result along a sequence of dyadic refinements. By the discussion at the start of Appendix B.2, we can thus focus on showing the weak law of large numbers holds for $\{S^{\mathbf{I}}(\mathbb{X}^n), \ n \geq 1\}$.

To apply the weak law of large number for dependent random variables, note that:

1. $S^{\mathbf{I}}(\mathbb{X}^n)_{[0,T]} \in L^2$, by the in-fill asymptotic result.

2. To show $\mathrm{Cov}\left(S^{\mathbf{I}}(\mathbb{X}^n)_{[0,T]}, S^{\mathbf{I}}(\mathbb{X}^m)_{[0,T]}\right) \to 0$ as $|m - n| \to \infty$, by stationarity of $\{\mathbb{X}_n, \ n \geq 1\}$, it is sufficient to show that
$$\mathrm{Cov}\left(S^{k'}(\mathbb{X}^1)_{[0,T]}, S^{k'}(\mathbb{X}^n)_{[0,T]}\right) \to 0, \quad n \to \infty, \tag{38}$$
for any $1 \leq k' \leq k = \max_{I \in \mathbf{I}} |I|$.

To show (38), we wish to find a bound on $\mathrm{Cov}(S^{k'}(\mathbb{X}^1)_{[0,T]}, S^{k'}(\mathbb{X}^n)_{[0,T]})$ vanishing to zero as $n \to \infty$. Note that, by the in-fill asymptotic result, along the signature-defining sequence of dyadic refinements, for all $n \geq 1$ and $1 \leq k' \leq k$,

$$\mathrm{Cov}\left(S^{k'}(\mathbb{X}^{\rho,1})_{[0,T]}, S^{k'}(\mathbb{X}^{\rho,n})_{[0,T]}\right) \to \mathrm{Cov}\left(S^{k'}(\mathbb{X}^1)_{[0,T]}, S^{k'}(\mathbb{X}^n)_{[0,T]}\right), \quad |\rho| \to 0. \tag{39}$$

To bound $\mathrm{Cov}(S^{k'}(\mathbb{X}^1)_{[0,T]}, S^{k'}(\mathbb{X}^n)_{[0,T]})$, we can hence start by bounding the covariance between the discretized signatures $\mathrm{Cov}(S^{k'}(\mathbb{X}^{\rho,1})_{[0,T]}, S^{k'}(\mathbb{X}^{\rho,n})_{[0,T]})$ and then let $|\rho| \to 0$.

Let $\rho$ be a dyadic partition of $[0, T]$ and note the form of the discretized signature

$$S(\mathbb{X}^\rho)_{[0,T]} = \bigotimes_{[u,v] \in \rho} \exp_\otimes \mathbf{X}_{u,v},$$

implies

$$S^{k'}(\mathbb{X}^\rho)_{[0,T]} = \sum_{i_\rho \in \mathcal{M}_\rho^{k'}} \bigotimes_{[u,v] \in \rho} \frac{\mathbf{X}_{u,v}^{\otimes i_{[u,v]}}}{i_{[u,v]}!},$$

where

$$\mathcal{M}_\rho^{k'} := \{i_\rho := \{i_{[u,v]}\}_{[u,v]\in\rho} : \ 0 \le i_{[u,v]} \le k', \ [u,v]\in\rho, \ \sum_{[u,v]\in\rho} i_{[u,v]} = k'\},$$

is the set of multiindices over $\rho$ with sum of components $k'$. Note that we can rewrite this as

$$S^{k'}(\mathbb{X}^\rho)_{[0,T]} = \sum_{\substack{j=1 \\ }}^{k'} \sum_{\substack{i_1,\dots,i_j \ge 1 \\ i_1+\cdots+i_j=k'}} \sum_{[u_1,v_1]<\cdots<[u_j,v_j]\in\rho} \cdots \sum \bigotimes_{l=1}^{j} \frac{\mathbf{X}_{u_l,v_l}^{\otimes i_l}}{i_l!}$$

$$= \sum_{\substack{j=1 \\ }}^{k'} \sum_{\substack{i_1,\dots,i_j \ge 1 \\ i_1+\cdots+i_j=k'}} \sum_{[u_1,v_1]<\cdots<[u_j,v_j]\in\rho} \cdots \sum \left[\prod_{l=1}^{j} \frac{1}{i_l!}\right] \bigotimes_{x=1}^{k'} \mathbf{X}_{u_{l_x},v_{l_x}},$$

where we first split the sum over the number of non-zero $i_{[u,v]}$ for each $i_\rho \in \mathcal{M}_\rho^{k'}$ and then rewrite the tensor product by introducing $(l_1,\cdots,l_{k'}) = (1,\dots,1,2,\dots,2,\dots,j,\dots,j)$, where the index 1 is repeated $i_1$ times, the index 2 is repeated $i_2$ times, and so on. We introduce the notation $[u_1,v_1] <_{m'} [u_2,v_2]$ denoting at least $m'$ intervals between $[u_1,v_1]$ and $[u_2,v_2]$ in $\rho$. When $m'=0$, we equivalently write $<$ or $<_0$ to denote the interval $[u_2,v_2]$ being after $[u_1,v_1]$ in $\rho$.

We then group the above summations over the intervals $[u_1,v_1],\dots,[u_j,v_j]$ over all possible combinations of time intervals with at least one pair less than $m$ steps away in the partition $\rho$. To do so, introduce the set[17] variable $\mathcal{I} \oplus_0 \mathcal{J} \in \mathcal{S}_{j',j,m}$ where $\mathcal{I} \in \{1\} \times \{0,1\}^{j-1}$ is such that $|\mathcal{I}| = \mathcal{I}_1 + \dots + \mathcal{I}_j = j'$ and $\mathcal{J} \in \{1,\dots,m\}^{j-j'}$ where $m$ is such that (A$\theta$) holds. We can then recursively define, for $l = 1,\dots,j$,

$$[u_l,v_l](\mathcal{I} \oplus_0 \mathcal{J}) = \begin{cases} \text{interval } \mathcal{J}_l \text{ steps after } [u_{l-1},v_{l-1}](\mathcal{I} \oplus_0 \mathcal{J}) & \text{if } \mathcal{I}_l = 0, \\ [u_{|\mathcal{I}_{1:l}|}, v_{|\mathcal{I}_{1:l}|}] & \text{if } \mathcal{I}_l = 1, \end{cases}$$

where $|\mathcal{I}_{1:l}| = \mathcal{I}_1 + \dots + \mathcal{I}_l$, and write

$$S^{k'}(\mathbb{X}^\rho)_{[0,T]} = \sum_{\substack{j=1 \\ }}^{k'} \sum_{\substack{i_1,\dots,i_j \ge 1 \\ i_1+\cdots+i_j=k'}} \sum_{j'=1}^{j} \sum_{\mathcal{I}\oplus_0\mathcal{J}\in\mathcal{S}_{j',j,m}} \sum_{[u_1,v_1]<_{m_1}\cdots<_{m_{j'-1}}[u_{j'},v_{j'}]\in\rho} \cdots \sum \left[\prod_{l=1}^{j} \frac{1}{i_l!}\right] \bigotimes_{x=1}^{k'} \mathbf{X}_{[u_{l_x},v_{l_x}](\mathcal{I}\oplus_0\mathcal{J})},$$

where, for each $l' = 1,\dots,j'-1$, we have $[u_{l'},v_{l'}] <_{m_{l'}} [u_{l'+1},v_{l'+1}]$, i.e. there are at least $m_{l'}$ intervals between $[u_{l'},v_{l'}]$ and $[u_{l'+1},v_{l'+1}]$ in $\rho$, where

$$m_{l'} = m_{l'}(\mathcal{I} \oplus_0 \mathcal{J}) = \sum_{l=1}^{j} \mathcal{J}_l \mathbb{1}_{\{|\mathcal{I}_{1:l}|=l', \mathcal{I}_l=0\}}.$$

The structure of the tensor products in the summation is thus

$$\mathbf{X}_{u_1,v_1}^{\otimes i_1} \otimes \mathbf{X}_{u_1^2,v_1^2}^{\otimes i_2} \otimes \cdots \otimes \mathbf{X}_{u_1^{n_1},v_1^{n_1}}^{\otimes i_{n_1}} \otimes \mathbf{X}_{u_2,v_2}^{\otimes i_{n_1+1}} \otimes \mathbf{X}_{u_2^2,v_2^2}^{\otimes i_{n_1+2}} \otimes \cdots \otimes \mathbf{X}_{u_2^{n_2},v_2^{n_2}}^{\otimes i_{n_1+n_2}} \otimes \cdots,$$

where $[u_1,v_1], [u_1^2,v_1^2], \cdots, [u_1^{n_1},v_1^{n_1}]$ are all less than $m$ intervals apart and $[u_2,v_2]$ is at least $m$ intervals after $[u_1^{n_1},v_1^{n_1}]$ (and so on)[18].

Let $n \ge 3$ and define $\rho_n = (n-1)T + \rho$. Then, for each word $I = (w_1,\dots,w_{k'})$ of length $k'$, we can write

$$\text{Cov}\left(S^I(\mathbb{X}^{\rho,1})_{[0,T]}, S^I(\mathbb{X}^{\rho,n})_{[0,T]}\right)$$

---

[17] Here $\mathcal{I} \oplus_0 \mathcal{J}$ denotes pairing elements of $\mathcal{J}$ to the elements of $\mathcal{I}$ equal to 0.

[18] To help intuitive understanding consider the case where $k'=2$ and $m=1$, then the above expression reduces to

$$S^2(\mathbb{X}^\rho)_{[0,T]} = \sum_{[u_1,v_1]\in\rho} \frac{\mathbf{X}_{u_1,v_1}^{\otimes 2}}{2!} + \sum_{[u_1,v_1]\in\rho} \mathbf{X}_{u_1,v_1} \otimes \mathbf{X}_{v_1,w_1} + \sum_{\substack{[u_1,v_1]\in\rho \\ [u_2,v_2]\in\rho \\ [u_1,v_1]<_1[u_2,v_2]}} \mathbf{X}_{u_1,v_1} \otimes \mathbf{X}_{u_2,v_2},$$

$$
= \sum_{\substack{j_1,j_2=1}}^{k'} \sum_{\substack{i_1,\ldots,i_{j_1}\geq 1 \\ i_1+\cdots+i_{j_1}=k'}} \sum_{\substack{e_1,\ldots,e_{j_2}\geq 1 \\ e_1+\cdots+e_{j_2}=k'}} \sum_{j_1'=1}^{j_1} \sum_{j_2'=1}^{j_2} \sum_{\substack{\mathcal{I}_1\oplus_0\mathcal{J}_1\in S_{j_1',j_1,m} \\ \mathcal{I}_2\oplus_0\mathcal{J}_2\in S_{j_2',j_2,m}}}
$$

$$
\sum_{\substack{[u_1,v_1]<_{m_1^1}\cdots<_{m_{j_1'-1}^1}[u_{j_1'},v_{j_1'}]\in\rho \\ [s_1,t_1]<_{m_1^2}\cdots<_{m_{j_2'-1}^2}[s_{j_2'},t_{j_2'}]\in\rho_n}} \cdots \left[\prod_{l=1}^{j_1}\frac{1}{i_l!}\right]\left[\prod_{l=1}^{j_2}\frac{1}{e_l!}\right]\mathrm{Cov}\left(\prod_{x=1}^{k'} X^{(w_x)}_{[u_{l_x^1},v_{l_x^1}](\mathcal{I}_1\oplus_0\mathcal{J}_1)}, \prod_{x=1}^{k'} X^{(w_x)}_{[s_{l_x^2},t_{l_x^2}](\mathcal{I}_2\oplus_0\mathcal{J}_2)}\right)
$$

$$
= \sum_{\substack{j_1,j_2=1}}^{k'} \sum_{\substack{i_1,\ldots,i_{j_1}\geq 1 \\ i_1+\cdots+i_{j_1}=k'}} \sum_{\substack{e_1,\ldots,e_{j_2}\geq 1 \\ e_1+\cdots+e_{j_2}=k'}} \sum_{j_1'=1}^{j_1} \sum_{j_2'=1}^{j_2} \sum_{\substack{\mathcal{I}_1\oplus_0\mathcal{J}_1\in S_{j_1',j_1,m} \\ \mathcal{I}_2\oplus_0\mathcal{J}_2\in S_{j_2',j_2,m}}} \sum_{\substack{[u_1,v_1]<_{m_1^1}\cdots<_{m_{j_1'-1}^1}[u_{j_1'},v_{j_1'}]\in\rho \\ [s_1,t_1]<_{m_1^2}\cdots<_{m_{j_2'-1}^2}[s_{j_2'},t_{j_2'}]\in\rho_n}} \cdots \left[\prod_{l=1}^{j_1}\frac{1}{i_l!}\right]\left[\prod_{l=1}^{j_2}\frac{1}{e_l!}\right]
$$

$$
\sum_{p\in MP^2_{(2,k')}} \prod_{\substack{\{(\delta_1,x_1), \\ (\delta_2,x_2)\}\in p}} \mathrm{Cov}\left(X^{(w_{x_1})\delta_1}_{[u_{l_{x_1}^1},v_{l_{x_1}^1}](\mathcal{I}_1\oplus_0\mathcal{J}_1)}X^{(w_{x_1})(1-\delta_1)}_{[s_{l_{x_1}^2},t_{l_{x_1}^2}](\mathcal{I}_2\oplus_0\mathcal{J}_2)}, X^{(w_{x_2})\delta_2}_{[u_{l_{x_2}^1},v_{l_{x_2}^1}](\mathcal{I}_1\oplus_0\mathcal{J}_1)}X^{(w_{x_2})(1-\delta_2)}_{[s_{l_{x_2}^2},t_{l_{x_2}^2}](\mathcal{I}_2\oplus_0\mathcal{J}_2)}\right),
$$

by using the fact that, for two collections of mean-zero normal random variables $(Z_{0,1},\ldots,Z_{0,k'})$ and $(Z_{1,1},\ldots,Z_{1,k'})$, we can apply Isserlis' theorem (Isserlis, 1918) to show

$$
\mathrm{Cov}\left(Z_{0,1}\cdots Z_{0,k'}, Z_{1,1}\cdots Z_{1,k'}\right)
$$
$$
= \mathbb{E}\left[Z_{0,1}\cdots Z_{0,k'}Z_{1,1}\cdots Z_{1,k'}\right] - \mathbb{E}\left[Z_{0,1}\cdots Z_{0,k'}\right]\mathbb{E}\left[Z_{1,1}\cdots Z_{1,k'}\right]
$$
$$
= \sum_{p\in P^2_{(2,k')}} \prod_{\substack{\{(i_1,i_2), \\ (j_1,j_2)\}\in p}} \mathbb{E}\left[Z_{i_1,i_2}Z_{j_1,j_2}\right] - \left(\sum_{q\in P^2_{k'}}\prod_{\{i,j\}\in q}\mathbb{E}\left[Z_{0,i}Z_{0,j}\right]\right)\left(\sum_{r\in P^2_{k'}}\prod_{\{i,j\}\in r}\mathbb{E}\left[Z_{1,i}Z_{1,j}\right]\right)
$$
$$
= \sum_{p\in MP^2_{(2,k')}} \prod_{\{(i_1,i_2),(j_1,j_2)\}\in p} \mathbb{E}\left[Z_{i_1,i_2}Z_{j_1,j_2}\right]
$$
$$
= \sum_{p\in MP^2_{(2,k')}} \prod_{\{(i_1,i_2),(j_1,j_2)\}\in p} \mathrm{Cov}\left(Z_{i_1,i_2},Z_{j_1,j_2}\right),
$$

where $P^2_{(2,k')}$ denotes the set of all the pairings of $\{0,1\}\times\{1,\ldots,k'\}$, $P^2_{k'}$ denotes the set of all the pairings of $\{1,\ldots,k'\}$ and $MP^2_{(2,k')}$ denotes the set of all the pairings of $\{0,1\}\times\{1,\ldots,k'\}$ that contain at least one "mixed" pair, i.e. for all $p\in MP^2_{(2,k')}$ there exist $\{(i_1,i_2),(j_1,j_2)\}\in p$ such that $i_1\neq j_1$.

Note that each $[u_{l'},v_{l'}]$ for $l'=1,\ldots,j_1'$ appears in at least one covariance term. We proceed by cases:

- If $[u_{l'},v_{l'}]$ appears in a pair with an interval $[u_*,v_*]$ such that $[u_{l'-1},v_{l'-1}]<[u_*,v_*]<[u_{l'+1},v_{l'+1}]$, then

$$
|\mathrm{Cov}(X^{(w)}_{[u_{l'},v_{l'}]},X^{(q)}_{[u_*,v_*]})|\lesssim|v_{l'}-u_{l'}|,
$$

  by applying Assumption (A$\alpha$) with $\alpha\geq 1/2$ and the fact that $\rho$ is dyadic (and hence uniform).

- If $[u_{l'},v_{l'}]$ appears in a pair with an interval $[u_*,v_*]$ such that $[u_*,v_*]\leq[u_{l'-1},v_{l'-1}]$, then we have $|u_{l'}-v_*|\geq$

where $[v_1,w_1]$ is the interval right-contiguous to $[u_1,v_1]$ in $\rho$. If $k'=2$ and $m=2$, instead

$$
S^2(\mathbb{X}^\rho)_{[0,T]} = \sum_{[u_1,v_1]\in\rho}\frac{\mathbf{X}^{\otimes 2}_{u_1,v_1}}{2!} + \sum_{[u_1,v_1]\in\rho}\mathbf{X}_{u_1,v_1}\otimes\mathbf{X}_{v_1,w_1} + \sum_{[u_1,v_1]\in\rho}\mathbf{X}_{u_1,v_1}\otimes\mathbf{X}_{w_1,z_1}
$$
$$
+ \sum_{[u_1,v_1]\in\rho}\sum_{\substack{[u_2,v_2]\in\rho \\ [u_1,v_1]<_2[u_2,v_2]}}\mathbf{X}_{u_1,v_1}\otimes\mathbf{X}_{u_2,v_2},
$$

where $[u_1,v_1],[v_1,w_1],[w_1,z_1]$ are consecutive intervals in $\rho$.

$m/2(|v_* - u_*| + |v_{l'} - u_{l'}|)$ and

$$|\text{Cov}(X^{(w)}_{[u_{l'}, v_{l'}]}, X^{(q)}_{[u_*, v_*]})| \lesssim \theta(|u_{l'} - v_*|)|v_* - u_*||v_{l'} - u_{l'}| \lesssim \theta(|u_{l'} - v_{l'-1}|)|v_* - u_*||v_{l'} - u_{l'}|,$$

by applying ($A\theta$).

- Similarly, if $[u_{l'}, v_{l'}]$ appears in a pair with an interval $[u_*, v_*]$ such that $[u_{l'+1}, v_{l'+1}] \leq [u_*, v_*]$, then we have $|u_* - v_{l'}| \geq m/2(|v_{l'} - u_{l'}| + |v_* - u_*|)$ and

$$|\text{Cov}(X^{(w)}_{[u_{l'}, v_{l'}]}, X^{(q)}_{[u_*, v_*]})| \lesssim \theta(|u_* - v_{l'}|)|v_{l'} - u_{l'}||v_* - u_*| \lesssim \theta(|u_{l'+1} - v_{l'}|)|v_{l'} - u_{l'}||v_* - u_*|.$$

- If $[u_{l'}, v_{l'}]$ appears in a pair with an interval $[s_*, t_*] \in \rho_n$, then

$$|\text{Cov}(X^{(w)}_{[u_{l'}, v_{l'}]}, X^{(q)}_{[s_*, t_*]})| \lesssim \theta((n-2)T)|v_{l'} - u_{l'}||t_* - s_*|,$$

by applying ($A\theta$).

A similar reasoning applies to each $[s_{l'}, t_{l'}]$, for $l' = 1, \dots, j'_2$. Noting that at least one pairing is mixed across $\rho$ and $\rho_n$ we can hence bound

$$|\text{Cov}\left(S^I(\mathbb{X}^{\rho,1})_{[0,T]}, S^I(\mathbb{X}^{\rho,n})_{[0,T]}\right)|$$

$$\lesssim \sum_{\substack{j_1,j_2=1}}^{k'} \sum_{\substack{i_1,\dots,i_{j_1} \geq 1 \\ i_1 + \cdots + i_{j_1} = k'}} \sum_{\substack{e_1,\dots,e_{j_2} \geq 1 \\ e_1 + \cdots + e_{j_2} = k'}} \sum_{j'_1=1}^{j_1} \sum_{j'_2=1}^{j_2} \sum_{\substack{\mathcal{I}_1 \oplus_0 \mathcal{J}_1 \in S_{j'_1, j_1, m} \\ \mathcal{I}_2 \oplus_0 \mathcal{J}_2 \in S_{j'_2, j_2, m}}} \sum_{\substack{[u_1, v_1] < m_1^1 \cdots < m_{j'_1-1}^1 \\ [s_1, t_1] < m_1^2 \cdots < m_{j'_2-1}^2}} \cdots \sum_{\substack{[u_{j'_1}, v_{j'_1}] \in \rho \\ [s_{j'_2}, t_{j'_2}] \in \rho_n}} \left[\prod_{l=1}^{j_1} \frac{1}{i_l!}\right]\left[\prod_{l=1}^{j_2} \frac{1}{e_l!}\right]$$

$$\sum_{p \in MP^2_{(2,k')}} \theta((n-2)T)|v_1 - u_1|\theta(|u_2 - v_1|)|v_2 - u_2| \cdots \theta(|u_{j'_1} - v_{j'_1-1}|)|v_{j'_1} - u_{j'_1}|$$

$$\times |t_1 - s_1|\theta(|s_2 - t_1|)|t_2 - s_2| \cdots \theta(|s_{j'_2} - t_{j'_2-1}|)|t_{j'_2} - s_{j'_2}|$$

$$\lesssim \sum_{\substack{j_1,j_2=1}}^{k'} \sum_{\substack{i_1,\dots,i_{j_1} \geq 1 \\ i_1 + \cdots + i_{j_1} = k'}} \sum_{\substack{e_1,\dots,e_{j_2} \geq 1 \\ e_1 + \cdots + e_{j_2} = k'}} \sum_{j'_1=1}^{j_1} \sum_{j'_2=1}^{j_2} \theta((n-2)T)$$

$$\times \left(\sum_{[u_1,v_1] \in \rho} |v_1 - u_1|\right)\left(\sum_{[u_2,v_2] \in \rho} \theta(|u_2|)|v_2 - u_2|\right) \cdots \left(\sum_{[u_{j'_1}, v_{j'_1}] \in \rho} \theta(|u_{j'_1}|)|v_{j'_1} - u_{j'_1}|\right)$$

$$\times \left(\sum_{[s_1,t_1] \in \rho_n} |t_1 - s_1|\right)\left(\sum_{[s_2,t_2] \in \rho_n} \theta(|s_2|)t_2 - s_2|\right) \cdots \left(\sum_{[s_{j'_2}, t_{j'_2}] \in \rho_n} \theta(|s_{j'_2}|)|t_{j'_2} - s_{j'_2}|\right)$$

$$\to \theta((n-2)T)\left(\sum_{\substack{j_1,j_2=1}}^{k'} \sum_{\substack{i_1,\dots,i_{j_1} \geq 1 \\ i_1 + \cdots + i_{j_1} = k'}} \sum_{\substack{e_1,\dots,e_{j_2} \geq 1 \\ e_1 + \cdots + e_{j_2} = k'}} \sum_{j'_1=1}^{j_1} \sum_{j'_2=1}^{j_2} T^2\left(\int_0^T \theta(t)dt\right)^{j'_1+j'_2-2}\right), \quad |\rho| \to 0.$$

Combining this bound with (39), we can conclude that, for all $n \geq 3$,

$$|\text{Cov}\left(S^I(\mathbb{X}^1)_{[0,T]}, S^I(\mathbb{X}^n)_{[0,T]}\right)| \lesssim \theta((n-2)T) \to 0, \quad n \to \infty,$$

i.e. we have shown (38). Hence, we can apply the weak law of large numbers of dependent random variables and the in-fill asymptotics to obtain the desired consistency result. $\qquad\square$

## C. Variance Reduction via Martingale Correction

In this Section 2.2 we considered a single control obtained by substituting the outermost Stratonovich integral with an Itô integral. In principle, for a word of length $|I| = k \geq 2$, one could consider $2^{k-2}$ distinct controls: for any subset $\mathcal{I} \subseteq \{2, \dots, k-1\}$, one can obtain a control by changing each of the integrals with index in $\mathcal{I} \cup \{k\}$ to Itô integrals. One can then apply the controlled linear regression estimator (with only the intercept term as regressor) described in Appendix G. This family of controls will likely be highly correlated and hence:

- the improvements provided by each additional control might be quite marginal compared to the considerable increase in computational cost;

- the estimator of the (inverse) variance matrix of the controls, needed to make the estimator feasible, might be quite unstable.

Hence, for clarity of exposition and computational ease, throughout the rest of this work we only consider the control variate estimator introduced in Section 2.2, i.e. for a fixed word $I$

$$\hat{\phi}_I^{N,\pi,c}(T) := \frac{1}{N}\sum_{n=1}^{N}\left(S^I(\mathbb{X}^{n,\pi})_{[0,T]} - cS_c^I(\mathbb{X}^{n,\pi})_{[0,T]}\right),\tag{40}$$

where $I_{-1} := (i_1, \ldots, i_{k-1})$ and

$$S_c^I(\mathbb{X}^\pi)_{[0,T]} := \sum_{[u,v]\in\pi} S^{I_{-1}}(\mathbb{X}^\pi)_{[0,u]}X_{u,v}^{(i_k)}.$$

## C.1. Martingale Continuity Criterion

If $\mathbb{X}$ is a martingale, then, by the Burkholder-Davis-Gundy (BDG) inequality (Burkholder et al., 1972), we can write a stronger version of assumptions (A$\alpha$) in terms of the quadratic variation of $\mathbb{X}$, for all $0 \leq s \leq t \leq T$

(A$\alpha$.M) $\|\langle \mathbf{X}\rangle_{s,t}\|_{L^{p/2}} \lesssim |t-s|^{2\alpha}$.

Note that for many martingales assumption (A$\alpha$.M) holds with $\alpha = 1/2$ and hence, since (A$\delta$) holds trivially, we can usually apply Theorem 2.8 under *(ii)*. Some non-trivial degenerate cases exist: For example, consider a one-dimensional mean-zero Gaussian martingale over $[0,1]$ with covariance function $C(s\wedge t)$ where $C$ is the Cantor function. This process has quadratic variation $\langle X\rangle_t = C(t)$ and hence – since the Cantor function is Hölder continuous with Hölder exponent $\log_3(2)$ – assumption (A$\alpha$.M) is satisfied with $\alpha = \frac{1}{2}\log_3 2 \in (1/4, 1/3]$. In this case, one can easily check that (A$\beta$) and (A$\gamma$) hold by combining the independent increments and martingale property of $\mathbb{X}$.

## C.2. Estimating $c_\pi^*$

In Section 2.2, we considered the following estimator for $c_\pi^*$:

$$\hat{c}_{\pi,1}^* = \frac{\sum_{n=1}^{N}S^I(\mathbb{X}^{n,\pi})_{[0,T]}S_c^I(\mathbb{X}^{n,\pi})_{[0,T]}}{\sum_{n=1}^{N}S_c^I(\mathbb{X}^{n,\pi})_{[0,T]}^2}.$$

Alternatively, we can exploit the explicit form of the covariance and variance of the infeasible estimator and approximate

$$\text{Cov}(S^I(\mathbb{X}^\pi)_{[0,T]}, S_c^I(\mathbb{X}^\pi)_{[0,T]}) \approx \text{Cov}(S^I(\mathbb{X})_{[0,T]}, S_c^I(\mathbb{X})_{[0,T]})$$

$$= \sum_{J\in I\sqcup I}\psi_J(T) - \frac{1}{2}\sum_{J\in I\sqcup I_{-2}*((i_{k-1},i_k))}\psi_J(T)$$

$$\approx \sum_{J\in I\sqcup I}\hat{\psi}_J^N(T) - \frac{1}{2}\sum_{J\in I\sqcup I_{-2}*((i_{k-1},i_k))}\hat{\psi}_J^N(T)$$

$$\approx \sum_{J\in I\sqcup I}\hat{\psi}_J^{N,\pi}(T) - \frac{1}{2}\sum_{J\in I\sqcup I_{-2}*((i_{k-1},i_k))}\hat{\psi}_J^{N,\pi}(T)$$

$$\approx \sum_{J\in I\sqcup I}\hat{\psi}_J^{N,\pi,\prime}(T) - \frac{1}{2}\sum_{J\in I\sqcup I_{-2}*((i_{k-1},i_k))}\hat{\psi}_J^{N,\pi,\prime}(T),$$

where

$$\hat{\psi}_J^N(T) := \frac{1}{N}\sum_{n=1}^{N}S^J((\mathbb{X}, \langle\mathbb{X}\rangle)^n)_{[0,T]}, \quad \hat{\psi}_J^{N,\pi}(T) := \frac{1}{N}\sum_{n=1}^{N}S^J((\mathbb{X}, \langle\mathbb{X}\rangle)^{n,\pi})_{[0,T]},$$

and

$$\hat{\psi}_J^{N,\pi,'}(T) := \frac{1}{N} \sum_{n=1}^N S^J((\mathbb{X}^{n,\pi}, \langle \hat{\mathbb{X}} \rangle^{n,\pi}))_{[0,T]},$$

with $\langle \hat{\mathbb{X}} \rangle^\pi = \{\langle \hat{\mathbf{X}} \rangle_t^\pi, \ t \in [0,T]\}$ defined as

$$\langle \hat{\mathbf{X}} \rangle_t^\pi = \sum_{[u',v'] \in \pi_{[0,u]}} \mathbf{X}_{u',v'}^2 + \frac{t-u}{v-u} \mathbf{X}_{u,v}^2, \quad t \in [u,v].$$

Similarly, we can approximate

$$\text{Var}(S_c^I(\mathbb{X}^\pi)_{[0,T]}) \approx \text{Var}(S_c^I(\mathbb{X})_{[0,T]}) \approx \sum_{J \in I_{-1} \sqcup I_{-1}} \hat{\psi}_{J*((i_k,i_k))}^{N,\pi}(T) \approx \sum_{J \in I_{-1} \sqcup I_{-1}} \hat{\psi}_{J*((i_k,i_k))}^{N,\pi,'}(T),$$

and hence we define the second estimator for $c_\pi^*$ as

$$\hat{c}_{\pi,2}^* = \frac{\sum_{J \in I \sqcup I} \hat{\psi}_J^{N,\pi,'}(T) - \frac{1}{2} \sum_{J \in I \sqcup I_{-2}*((i_{k-1},i_k))} \hat{\psi}_J^{N,\pi,'}(T)}{\sum_{J \in I_{-1} \sqcup I_{-1}} \hat{\psi}_{J*((i_k,i_k))}^{N,\pi,'}(T)}$$

$$= \frac{\sum_{n=1}^N \left( \sum_{J \in I \sqcup I} S^J((\mathbb{X}^{n,\pi}, \langle \hat{\mathbb{X}} \rangle^{n,\pi}))_{[0,T]} - \frac{1}{2} \sum_{J \in I \sqcup I_{-2}*((i_{k-1},i_k))} S^J((\mathbb{X}^{n,\pi}, \langle \hat{\mathbb{X}} \rangle^{n,\pi}))_{[0,T]} \right)}{\sum_{n=1}^N \sum_{J \in I_{-1} \sqcup I_{-1}} S^{J*((i_k,i_k))}((\mathbb{X}^{n,\pi}, \langle \hat{\mathbb{X}} \rangle^{n,\pi}))_{[0,T]}}.$$

Whether estimator $\hat{c}_{\pi,1}^*$ or estimator $\hat{c}_{\pi,2}^*$ is more precise, in terms of MSE, depends on the properties of the process $\mathbb{X}$ and the expected signature word $I$ being estimated by (40).

**Lemma C.1.** *Let $\mathbb{X} = \{\mathbf{X}_t, \ t \in [0,T]\}$ be a square-integrable martingale satisfying Assumption ($A\alpha$.M), for some $\alpha \geq 1/2$ and $p = 4k$ where $k = |I|$. Assume that $\pi$ is part of a sequence of refining partitions with mesh vanishing fast enough, i.e.*

$$\sum_{n \geq 1} |\pi_n|^{2\alpha - 1/2} < \infty.$$

*Then the difference between the mean-square errors of the two estimators $\hat{c}_{\pi,1}^*$ and $\hat{c}_{\pi,2}^*$ for $c_\pi^*$ is approximately*

$$\mathbb{E}[(\hat{c}_{\pi,2}^* - c_\pi^*)^2] - \mathbb{E}[(\hat{c}_{\pi,1}^* - c_\pi^*)^2] \approx \frac{1}{N} \frac{\mu_Y}{\mu_Z^3} \left( \frac{\mu_Y}{\mu_Z} (\mathbb{E}[Z_2^2] - \mathbb{E}[Z_1^2]) - 2(\mathbb{E}[YZ_2] - \mathbb{E}[YZ_1]) \right),$$

*where*

$$Y = S^I(\mathbb{X})_{[0,T]} S_c^I(\mathbb{X})_{[0,T]}, \quad Z_1 = S_c^I(\mathbb{X})_{[0,T]}^2, \quad Z_2 = \sum_{J \in I_{-1} \sqcup I_{-1}} S^{J*((i_k,i_k))}((\mathbb{X}, \langle \mathbb{X} \rangle))_{[0,T]},$$

*and $\mu_Y = \mathbb{E}[Y]$, $\mu_Z = \mathbb{E}[Z_1] = \mathbb{E}[Z_2]$.*

*Proof.* See Appendix C.3. $\qquad \square$

In practical applications, the above expression cannot be evaluated exactly, but we can approximate it by its sample estimate

$$\mathbb{E}[(\hat{c}_{\pi,2}^* - c_\pi^*)^2] - \mathbb{E}[(\hat{c}_{\pi,1}^* - c_\pi^*)^2] \propto \frac{1}{N^2} \sum_{n=1}^N \left( \frac{\bar{\mu}_Y}{\bar{\mu}_Z} ((Z_{2,n}^\pi)^2 - (Z_{1,n}^\pi)^2) - \sum_{j=1}^2 (Y_{j,n}^\pi Z_{2,n}^\pi - Y_{j,n}^\pi Z_{1,n}^\pi) \right).$$

where

$$\bar{\mu}_Y = \frac{1}{2N} \left( \sum_{n=1}^N Y_{1,n}^\pi + \sum_{n=1}^N Y_{2,n}^\pi \right), \quad \bar{\mu}_Z = \frac{1}{2N} \left( \sum_{n=1}^N Z_{1,n}^\pi + \sum_{n=1}^N Z_{2,n}^\pi \right),$$

and $Y_{1,n}^{\pi}, Z_{1,n}^{\pi}, Y_{2,n}^{\pi}, Z_{2,n}^{\pi}$ are given in Appendix C.3.

Another important discriminant when choosing between estimators $\hat{c}_{\pi,1}^{*}$ and $\hat{c}_{\pi,2}^{*}$ is usually computational cost. To compute $\hat{c}_{\pi,1}^{*}$ it suffices to regress $\{S^{I}(\mathbb{X}^{n,\pi})_{[0,T]}, \ n = 1, \ldots, N\}$ against $\{S_{c}^{I}(\mathbb{X}^{n,\pi})_{[0,T]}, \ n = 1, \ldots, N\}$. Both samples need to be computed to evaluate the control-variate estimator (40) and thus the extra computational cost of $\hat{c}_{\pi,1}^{*}$ is just the cost of a simple linear regression with sample size $N$, namely $\mathcal{O}(N)$. On the other hand, to compute $\hat{c}_{\pi,2}^{*}$, one needs to compute all the higher-order expected signature estimates $\hat{\psi}_{J}^{N,\pi,\prime}(T)$ with $J \in I \sqcup I$, $J \in I \sqcup I_{-2} * ((i_{k-1}, i_{k}))$ and $J = J' * ((i_{k}, i_{k}))$ for $J' \in I_{-1} \sqcup I_{-1}$, which has (naive) extra computational cost $\mathcal{O}(|\pi|^{-1}Tk(d^{2k} + (d+1)^{2k-1}))$ when parallelizing across the $N$ samples. In the in-fill limit, $|\pi|^{-1}T \gg N$ and, hence, computing $\hat{c}_{\pi,2}^{*}$ is significantly more expensive than computing $\hat{c}_{\pi,1}^{*}$.

## C.3. Proof of Lemma C.1

*Sketch of proof.* To compare the two estimators $\hat{c}_{\pi,1}^{*}$ and $\hat{c}_{\pi,2}^{*}$ we exploit their structure as ratio estimators based on i.i.d. observations of numerator random variables $Y_{1}^{\pi}, Y_{2}^{\pi}$ and denominator random variables $Z_{1}^{\pi}, Z_{2}^{\pi}$ respectively. We first show that the two estimators are both (biased) estimators for $c_{\pi}^{*} = c_{\pi,1}^{*} = c_{\pi,2}^{*}$ where

$$c_{\pi,1}^{*} = \frac{\mathbb{E}[Y_{1}^{\pi}]}{\mathbb{E}[Z_{1}^{\pi}]} \quad \text{and} \quad c_{\pi,2}^{*} = \frac{\mathbb{E}[Y_{2}^{\pi}]}{\mathbb{E}[Z_{2}^{\pi}]}.$$

*The first equality is trivial while the second requires several applications of Theorem 2.8 which are detailed in Lemma C.3. We then derive a simple formula for the mean squared error of ratio estimators in terms of the means and variances of the numerator and denominator random variables. Taking the limit $|\pi| \downarrow 0$ and applying Theorem 2.8 to show the second order statistics of $Y_{1}^{\pi}, Y_{2}^{\pi}, Z_{1}^{\pi}, Z_{2}^{\pi}$ converge to those of $Y_{1}, Y_{2}, Z_{1}, Z_{2}$ yields the desired result.*

Note that both $\hat{c}_{\pi,1}^{*}$ and $\hat{c}_{\pi,2}^{*}$ are ratio estimators of the form

$$\hat{c}_{\pi,j}^{*} = \frac{\bar{Y}_{j}^{\pi}}{\bar{Z}_{j}^{\pi}} = \frac{\sum_{n=1}^{N} Y_{j,n}^{\pi}}{\sum_{n=1}^{N} Z_{j,n}^{\pi}},$$

for random variables $Y_{j,1}^{\pi}, \ldots, Y_{j,N}^{\pi}$ and $Z_{j,1}^{\pi}, \ldots, Z_{j,N}^{\pi}$ with $j = 1, 2$ given by i.i.d. copies of

$$Y_{1}^{\pi} = S^{I}(\mathbb{X}^{\pi})_{[0,T]}S_{c}^{I}(\mathbb{X}^{\pi})_{[0,T]}, \quad Z_{1}^{\pi} = S_{c}^{I}(\mathbb{X}^{\pi})_{[0,T]}^{2},$$

and

$$Y_{2}^{\pi} = \sum_{J \in I \sqcup I} S^{J}((\mathbb{X}^{\pi}, \langle \hat{\mathbb{X}} \rangle^{\pi}))_{[0,T]} - \frac{1}{2} \sum_{J \in I \sqcup I_{-2}*((i_{k-1}, i_{k}))} S^{J}((\mathbb{X}^{\pi}, \langle \hat{\mathbb{X}} \rangle^{\pi}))_{[0,T]},$$

$$Z_{2}^{\pi} = \sum_{J \in I_{-1} \sqcup I_{-1}} S^{J*((i_{k}, i_{k}))}((\mathbb{X}^{\pi}, \langle \hat{\mathbb{X}} \rangle^{\pi}))_{[0,T]}.$$

By the standard theory of ratio estimators, these are biased estimators for

$$c_{\pi,j}^{*} = \frac{\mathbb{E}[Y_{j}^{\pi}]}{\mathbb{E}[Z_{j}^{\pi}]},$$

and applying a first-order Taylor expansion[19] we can approximate the mean squared error as

$$\mathbb{E}[(\hat{c}^*_{\pi,j} - c^*_{\pi,j})^2] \approx \frac{1}{\mathbb{E}[Z^\pi_j]^2}\mathrm{Var}(\bar{Y}^\pi_j) + \frac{\mathbb{E}[Y^\pi_j]^2}{\mathbb{E}[Z^\pi_j]^4}\mathrm{Var}(\bar{Z}^\pi_j) - 2\frac{\mathbb{E}[Y^\pi_j]}{\mathbb{E}[Z^\pi_j]^3}\mathrm{Cov}(\bar{Y}^\pi_j, \bar{Z}^\pi_j)$$

$$\approx \frac{1}{N}\left(\frac{1}{\mathbb{E}[Z^\pi_j]^2}\mathrm{Var}(Y^\pi_j) + \frac{\mathbb{E}[Y^\pi_j]^2}{\mathbb{E}[Z^\pi_j]^4}\mathrm{Var}(Z^\pi_j) - 2\frac{\mathbb{E}[Y^\pi_j]}{\mathbb{E}[Z^\pi_j]^3}\mathrm{Cov}(Y^\pi_j, Z^\pi_j)\right).$$

Note that

$$c^*_{\pi,1} = \frac{\mathbb{E}[S^I(\mathbb{X}^\pi)_{[0,T]}S^I_c(\mathbb{X}^\pi)_{[0,T]}]}{\mathbb{E}[S^I_c(\mathbb{X}^\pi)^2_{[0,T]}]} = \frac{\mathrm{Cov}(S^I(\mathbb{X}^\pi)_{[0,T]}, S^I_c(\mathbb{X}^\pi)_{[0,T]})}{\mathrm{Var}(S^I_c(\mathbb{X}^\pi)^2_{[0,T]})} = c^*_\pi,$$

and, as $|\pi| \to 0$,

$$c^*_{\pi,2} = \frac{\sum_{J \in I \sqcup I}\mathbb{E}[S^J((\mathbb{X}^\pi, \langle\hat{\mathbb{X}}\rangle^\pi))_{[0,T]}] - \frac{1}{2}\sum_{J \in I \sqcup I_{-2}*((i_{k-1},i_k))}\mathbb{E}[S^J((\mathbb{X}^\pi, \langle\hat{\mathbb{X}}\rangle^\pi))_{[0,T]}]}{\sum_{J \in I_{-1} \sqcup I_{-1}}\mathbb{E}[S^{J*((i_k,i_k))}((\mathbb{X}^\pi, \langle\hat{\mathbb{X}}\rangle^\pi))_{[0,T]}]}$$

$$\overset{(46)}{\approx} \frac{\sum_{J \in I \sqcup I}\mathbb{E}[S^J((\mathbb{X}, \langle\mathbb{X}\rangle)^\pi)_{[0,T]}] - \frac{1}{2}\sum_{J \in I \sqcup I_{-2}*((i_{k-1},i_k))}\mathbb{E}[S^J((\mathbb{X}, \langle\mathbb{X}\rangle)^\pi)_{[0,T]}]}{\sum_{J \in I_{-1} \sqcup I_{-1}}\mathbb{E}[S^{J*((i_k,i_k))}((\mathbb{X}, \langle\mathbb{X}\rangle)^\pi)_{[0,T]}]}$$

$$\overset{(43)}{\approx} \frac{\sum_{J \in I \sqcup I}\mathbb{E}[S^J((\mathbb{X}, \langle\mathbb{X}\rangle))_{[0,T]}] - \frac{1}{2}\sum_{J \in I \sqcup I_{-2}*((i_{k-1},i_k))}\mathbb{E}[S^J((\mathbb{X}, \langle\mathbb{X}\rangle))_{[0,T]}]}{\sum_{J \in I_{-1} \sqcup I_{-1}}\mathbb{E}[S^{J*((i_k,i_k))}((\mathbb{X}, \langle\mathbb{X}\rangle))_{[0,T]}]}$$

$$= \frac{\mathrm{Cov}(S^I(\mathbb{X})_{[0,T]}, S^I_c(\mathbb{X})_{[0,T]})}{\mathrm{Var}(S^I_c(\mathbb{X})^2_{[0,T]})}$$

$$\overset{(41),(42)}{\approx} \frac{\mathrm{Cov}(S^I(\mathbb{X}^\pi)_{[0,T]}, S^I_c(\mathbb{X}^\pi)_{[0,T]})}{\mathrm{Var}(S^I_c(\mathbb{X}^\pi)^2_{[0,T]})} = c^*_\pi.$$

We refer to Lemma C.3 for the rigorous justification of the approximations (41), (42), (43), (46). Moreover, as $|\pi| \to 0$, by (41) and (42),

$$Y^\pi_1 \overset{L^2}{\to} S^I(\mathbb{X})_{[0,T]}S^I_c(\mathbb{X})_{[0,T]} =: Y_1, \quad Z^\pi_1 \overset{L^2}{\to} S^I_c(\mathbb{X})^2_{[0,T]} =: Z_1,$$

and by combining (43) and (46),

$$Y^\pi_2 \overset{L^2}{\to} \sum_{J \in I \sqcup I}S^J((\mathbb{X}, \langle\mathbb{X}\rangle))_{[0,T]} - \frac{1}{2}\sum_{J \in I \sqcup I_{-2}*((i_{k-1},i_k))}S^J((\mathbb{X}, \langle\mathbb{X}\rangle))_{[0,T]} =: Y_2,$$

$$Z^\pi_2 \overset{L^2}{\to} \sum_{J \in I_{-1} \sqcup I_{-1}}S^{J*((i_k,i_k))}((\mathbb{X}, \langle\mathbb{X}\rangle))_{[0,T]} =: Z_2.$$

Note that $Y_1 = Y_2 =: Y$ but $Z_1 \neq Z_2$ even though $\mathbb{E}[Z_1] = \mathbb{E}[Z_2] =: \mu_Z$. When $|\pi|$ is small we can thus approximate the MSEs of $\hat{c}^*_{\pi,1}$ and $\hat{c}^*_{\pi,2}$ with respect to $c^*_\pi$ as

$$\mathbb{E}[(\hat{c}^*_{\pi,1} - c^*_\pi)^2] \approx \frac{1}{N}\left(\frac{1}{\mu^2_Z}\sigma^2_Y + \frac{\mu^2_Y}{\mu^4_Z}\mathrm{Var}(Z_1) - 2\frac{\mu_Y}{\mu^3_Z}\mathrm{Cov}(Y, Z_1)\right),$$

_______________

[19]On the set $|\bar{Z}^\pi_j - \mathbb{E}[Z^\pi_j]| < |\mathbb{E}[Z^\pi_j]|$,

$$\hat{c}_{\pi,j} = c^*_{\pi,j}\left(1 + \frac{\bar{Y}^\pi_j - \mathbb{E}[Y^\pi_j]}{\mathbb{E}[Y^\pi_j]}\right)\left(1 + \frac{\bar{Z}^\pi_j - \mathbb{E}[Z^\pi_j]}{\mathbb{E}[Z^\pi_j]}\right)^{-1}$$

$$= c^*_{\pi,j}\left(1 + \frac{\bar{Y}^\pi_j - \mathbb{E}[Y^\pi_j]}{\mathbb{E}[Y^\pi_j]}\right)\left(1 - \frac{\bar{Z}^\pi_j - \mathbb{E}[Z^\pi_j]}{\mathbb{E}[Z^\pi_j]} + \mathcal{O}\left(\left(\frac{\bar{Z}^\pi_j - \mathbb{E}[Z^\pi_j]}{\mathbb{E}[Z^\pi_j]}\right)^2\right)\right)$$

$$= c^*_{\pi,j} + \frac{1}{\mathbb{E}[Z^\pi_j]}(\bar{Y}^\pi_j - \mathbb{E}[Y^\pi_j]) - \frac{\mathbb{E}[Y^\pi_j]}{\mathbb{E}[Z^\pi_j]^2}(\bar{Z}^\pi_j - \mathbb{E}[Z^\pi_j]) + \mathcal{O}\left(\left(\frac{\bar{Y}^\pi_j - \mathbb{E}[Y^\pi_j]}{\mathbb{E}[Y^\pi_j]}\right)\left(\frac{\bar{Z}^\pi_j - \mathbb{E}[Z^\pi_j]}{\mathbb{E}[Z^\pi_j]}\right)\right).$$

For the approximation to be rigorous one needs to assume the probability of the set $\{\omega : |\bar{Z}^\pi_j(\omega) - \mathbb{E}[Z^\pi_j]| \geq |\mathbb{E}[Z^\pi_j]|\}$ approaches 0 faster than the speed at which the MSE conditional on this set explodes as $N \to \infty$.

and

$$\mathbb{E}[(\hat{c}_{\pi,2}^* - c_\pi^*)^2] \approx \frac{1}{N}\left(\frac{1}{\mu_Z^2}\sigma_Y^2 + \frac{\mu_Y^2}{\mu_Z^4}\text{Var}(Z_2) - 2\frac{\mu_Y}{\mu_Z^3}\text{Cov}(Y,Z_2)\right),$$

which differ by

$$\mathbb{E}[(\hat{c}_{\pi,2}^* - c_\pi^*)^2] - \mathbb{E}[(\hat{c}_{\pi,1}^* - c_\pi^*)^2] \approx \frac{1}{N}\frac{\mu_Y}{\mu_Z^3}\left(\frac{\mu_Y}{\mu_Z}(\mathbb{E}[Z_2^2] - \mathbb{E}[Z_1^2]) - 2(\mathbb{E}[YZ_2] - \mathbb{E}[YZ_1])\right).$$

*Remark* C.2. Note that we can "mix" the two estimators and form

$$\hat{c}_{\pi,2,1}^* = \frac{\bar{Y}_2^\pi}{\bar{Z}_1^\pi} \quad \text{and} \quad \hat{c}_{\pi,1,2}^* = \frac{\bar{Y}_1^\pi}{\bar{Z}_2^\pi}.$$

The discussion above ensures that, as $|\pi| \to 0$, $\hat{c}_{\pi,2,1}^*$ and $\hat{c}_{\pi,1,2}^*$ have the same MSEs as $\hat{c}_{\pi,1,1}^* = \hat{c}_{\pi,1}$ and $\hat{c}_{\pi,2,2}^* = \hat{c}_{\pi,2}$, respectively.

**Lemma C.3.** *Assume that $\mathbb{X}$ satisfies (A$\alpha$.M) for some $\alpha \geq 1/2$ and $p = 4k$ where $k = |I|$. Assume that $\pi$ is part of a sequence of refining partitions with mesh vanishing fast enough, i.e.*

$$\sum_{n \geq 1} |\pi_n|^{2\alpha - 1/2} < \infty.$$

*Then the approximations (41), (42), (43), (46) hold as $|\pi| \to 0$.*

*Proof.* By Appendix C.1 $\mathbb{X}$ satisfies (A$\alpha$) with $\alpha \geq 1/2$ and $p = 4k$. Note that since $\mathbb{X}$ is a martingale (A$\delta$) holds trivially and hence we can set $\epsilon = 1/2$.

We can apply Theorem 2.8 to deduce that

$$S^I(\mathbb{X}^\pi)_{[0,T]} \overset{L^4}{\to} S^I(\mathbb{X})_{[0,T]}, \quad |\pi| \to 0.$$

Moreover,

$$S_c^I(\mathbb{X}^\pi)_{[0,T]} \overset{L^4}{\to} S_c^I(\mathbb{X})_{[0,T]}, \quad |\pi| \to 0.$$

since

$$\|S_c^I(\mathbb{X})_{[0,T]} - S_c^I(\mathbb{X}^\pi)_{[0,T]}\|_{L^4}$$

$$\overset{(i)}{\leq} \left\|S_c^I(\mathbb{X})_{[0,T]} - \sum_{[u,v] \in \pi} S^{I_{-1}}(\mathbb{X})_{[0,u]}X_{u,v}^{(i_k)}\right\|_{L^4} + \left\|\sum_{[u,v] \in \pi}(S^{I_{-1}}(\mathbb{X})_{[0,u]} - S^{I_{-1}}(\mathbb{X}^\pi)_{[0,u]})X_{u,v}^{(i_k)}\right\|_{L^4}$$

$$\overset{(ii)}{\leq} \left\|\int_0^T S^{I_{-1}}(\mathbb{X})_{[0,s]}\mathrm{d}X_s^{(i_k)} - \sum_{[u,v] \in \pi} S^{I_{-1}}(\mathbb{X})_{[0,u]}X_{u,v}^{(i_k)}\right\|_{L^4}$$

$$+ 2C_2\left(\sum_{[u,v] \in \pi}\left\|S^{I_{-1}}(\mathbb{X})_{[0,u]} - S^{I_{-1}}(\mathbb{X}^\pi)_{[0,u]}\right\|_{L^{2k/(k-1)}}^2 \left\|X_{u,v}^{(i_k)}\right\|_{L^{2k}}^2\right)^{1/2}$$

$$\overset{(iii)}{\lesssim} \left\|\int_0^T S^{I_{-1}}(\mathbb{X})_{[0,s]}\mathrm{d}X_s^{(i_k)} - \sum_{[u,v] \in \pi} S^{I_{-1}}(\mathbb{X})_{[0,u]}X_{u,v}^{(i_k)}\right\|_{L^4}$$

$$+ 2C_2 \sup_{u \in \pi}\left\|S^{I_{-1}}(\mathbb{X})_{[0,u]} - S^{I_{-1}}(\mathbb{X}^\pi)_{[0,u]}\right\|_{L^{2k/(k-1)}}\left(\sum_{[u,v] \in \pi}|v-u|^{2\alpha}\right)^{1/2}$$

$$\overset{(iv)}{\to} 0, \quad |\pi| \to 0,$$

by applying in $(i)$ the triangle inequality, in $(ii)$ Lemma B.1 with the natural filtration of $\mathbb{X}$, in $(iii)$ Theorem 2.8 to the word $I_{-1}$ with $m = 2k/(k-1)$ and in $(iv)$ the definition of the Itô integral and Remark B.4.

Under these assumptions, we thus have

$$S^I(\mathbb{X}^\pi)_{[0,T]} S^I_c(\mathbb{X}^\pi)_{[0,T]} \xrightarrow{L^2} S^I(\mathbb{X})_{[0,T]} S^I_c(\mathbb{X})_{[0,T]}, \quad |\pi| \to 0, \tag{41}$$

and

$$S^I_c(\mathbb{X}^\pi)^2_{[0,T]} \xrightarrow{L^2} S^I_c(\mathbb{X})^2_{[0,T]}, \quad |\pi| \to 0. \tag{42}$$

Next, we consider the approximation (43). Note that $\mathbb{X}$ satisfies (A$\alpha$) (and trivially (A$\delta$)) with $\alpha \geq 1/2$ and $p = 4k$ while $\langle \mathbb{X} \rangle$ satisfies (A$\alpha$) and (A$\delta$) both with exponent $2\alpha \geq 1$ and $p = 2k$. By using a slightly more general version[20] of Theorem 2.8 applied to the process $(\mathbb{X}, \langle \mathbb{X} \rangle)$, for any word $J \in I \sqcup I$ or $J \in I \sqcup I_{-2} * ((i_{k-1}, i_k))$ or $J = J' * ((i_k, i_k))$ with $J' \in I_{-1} \sqcup I_{-1}$ we have

$$S^J((\mathbb{X}, \langle \mathbb{X} \rangle)^\pi)_{[0,T]} \xrightarrow{L^2} S^J((\mathbb{X}, \langle \mathbb{X} \rangle))_{[0,T]}, \quad |\pi| \to 0, \tag{43}$$

and hence for fixed $N \geq 1$,

$$\hat{\psi}^{N,\pi}_J(T) \xrightarrow{L^2} \hat{\psi}^N_J(T), \quad |\pi| \to 0.$$

Finally, making approximation (46) rigorous is a bit more challenging. Let us start by considering the case where $\langle \mathbb{X} \rangle$ only appears in the outermost integral, i.e. $J = (j_1, \ldots, j_{k'})$, where $j_1, \ldots, j_{k'-1} \in \{1, \ldots, d\}$ and $j_{k'} \in \{(1,1), \ldots, (1,d), \ldots, (d,d)\}$ for some $k' \in \{1, \ldots, 2k-1\}$. Then, for any $\tau \in \pi$,

$$\left\| S^J((\mathbb{X}, \langle \mathbb{X} \rangle)^\pi)_{[0,\tau]} - S^J((\mathbb{X}^\pi, \langle \hat{\mathbb{X}} \rangle^\pi))_{[0,\tau]} \right\|_{L^{4k/(k'+1)}}$$

$$\overset{(i)}{=} \left\| \sum_{i=1}^{k'} \frac{1}{i!} \sum_{[u,v] \in \pi_{[0,\tau]}} S^{J-i}(\mathbb{X}^\pi)_{[0,u]} X^{(j_{k'-i+1})}_{u,v} \cdots X^{(j_{k'-1})}_{u,v} \left( \langle X \rangle^{(j_{k'})}_{u,v} - \langle \hat{X} \rangle^{\pi,(j_{k'})}_{u,v} \right) \right\|_{L^{4k/(k'+1)}}$$

$$\overset{(ii)}{\leq} \sum_{i=1}^{k'} \frac{1}{i!} \left\| \sum_{[u,v] \in \pi_{[0,\tau]}} S^{J-i}(\mathbb{X}^\pi)_{[0,u]} X^{(j_{k'-i+1})}_{u,v} \cdots X^{(j_{k'-1})}_{u,v} \left( \langle X \rangle^{(j_{k'})}_{u,v} - \left( X^{(j_{k'})}_{u,v} \right)^2 \right) \right\|_{L^{4k/(k'+1)}}$$

$$\overset{(iii)}{\leq} \left( \sum_{[u,v] \in \pi_{[0,\tau]}} \left\| S^{J-1}(\mathbb{X}^\pi)_{[0,u]} \left( \langle X \rangle^{(j_{k'})}_{u,v} - \left( X^{(j_{k'})}_{u,v} \right)^2 \right) \right\|^2_{L^{4k/(k'+1)}} \right)^{1/2}$$

$$\quad + \sum_{i=2}^{k'} \frac{1}{i!} \sum_{[u,v] \in \pi_{[0,\tau]}} \left\| S^{J-i}(\mathbb{X}^\pi)_{[0,u]} X^{(j_{k'-i+1})}_{u,v} \cdots X^{(j_{k'-1})}_{u,v} \left( \langle X \rangle^{(j_{k'})}_{u,v} - \left( X^{(j_{k'})}_{u,v} \right)^2 \right) \right\|_{L^{4k/(k'+1)}}$$

$$\overset{(iii)}{\leq} \left( \sum_{[u,v] \in \pi_{[0,\tau]}} \left\| S^{J-1}(\mathbb{X}^\pi)_{[0,u]} \left( \langle X \rangle^{(j_{k'})}_{u,v} - \left( X^{(j_{k'})}_{u,v} \right)^2 \right) \right\|^2_{L^{4k/(k'+1)}} \right)^{1/2}$$

$$\quad + \sum_{i=2}^{k'} \frac{1}{i!} \sum_{[u,v] \in \pi_{[0,\tau]}} \left\| S^{J-i}(\mathbb{X}^\pi)_{[0,u]} X^{(j_{k'-i+1})}_{u,v} \cdots X^{(j_{k'-1})}_{u,v} \left( \langle X \rangle^{(j_{k'})}_{u,v} - \left( X^{(j_{k'})}_{u,v} \right)^2 \right) \right\|_{L^{4k/(k'+1)}}$$

$$\overset{(iv)}{\leq} \sup_{u \in \pi_{[0,\tau]}} \left\| S^{J-1}(\mathbb{X}^\pi)_{[0,u]} \right\|_{L^{4k/(k'-1)}} \left( \sum_{[u,v] \in \pi_{[0,\tau]}} \left\| \langle \mathbb{X} \rangle_{u,v} - \mathbf{X}^{\otimes 2}_{u,v} \right\|^2_{L^{2k}} \right)^{1/2}$$

$$\quad + \sum_{i=2}^{k'} \frac{1}{i!} \sup_{u \in \pi_{[0,\tau]}} \left\| S^{J-i}(\mathbb{X}^\pi)_{[0,u]} \right\|_{L^{4k/(k'-i)}} \sum_{[u,v] \in \pi_{[0,\tau]}} \left\| \mathbf{X}_{u,v} \right\|^{i-1}_{L^{4k}} \left\| \langle \mathbb{X} \rangle_{u,v} - \mathbf{X}^{\otimes 2}_{u,v} \right\|_{L^{2k}}$$

$$\overset{(v)}{\lesssim} |\pi|^{4\alpha-1} \sqrt{\tau} + |\pi|\tau \to 0, \quad |\pi| \to 0,$$

---

[20]To estimate the expected signature term corresponding to the word $I = (i_1, \ldots, i_k)$, it is sufficient to require *(i)*, *(ii)*, *(iii)*, or *(iv)* to be satisfied by $X^{(i_1)}, \ldots, X^{(i_k)}$ with $p_1, \ldots, p_k$ such that $p_1^{-1} + \cdots + p_k^{-1} \leq m^{-1}$. In Theorem 2.8 we considered the case where $p_1 = \cdots = p_k = p$ for clarity.

by using in $(i)$ Lemma B.3, in $(ii)$ the triangle inequality and the definition of $\langle\hat{\mathbb{X}}\rangle^\pi$, in $(iii)$ Lemma B.1 with the natural filtration of $\mathbb{X}$ (with respect to which $\mathbb{X} - \langle\mathbb{X}\rangle$ is a martingale) for the $i=1$ term and the triangle inequality for the $i = 2, \ldots, j$ terms, in $(iv)$ Hölder inequality, in $(v)$ Remark B.4 applied to $\mathbb{X}$ and $J_{-i}$ with $p = 4k$, $m = 4k/(k'-i)$ and $|J_{-i}| = k' - i$ for $i = 1, \ldots, k'$ and

$$\left\|\langle\mathbf{X}\rangle_{u,v} - \mathbf{X}_{u,v}^{\otimes 2}\right\|_{L^{2k}} \leq \|\langle\mathbf{X}\rangle_{u,v}\|_{L^{2k}} + \|\mathbf{X}_{u,v}\|_{L^{4k}}^2 \lesssim |v - u|^{2\alpha}, \tag{44}$$

by assumption (A$\alpha$.M) with $\alpha \geq 1/2$ and $p = 4k$. We have thus shown

$$\sup_{\tau\in\pi}\left\|S^J((\mathbb{X},\langle\mathbb{X}\rangle)^\pi)_{[0,\tau]} - S^J((\mathbb{X}^\pi,\langle\hat{\mathbb{X}}\rangle^\pi))_{[0,\tau]}\right\|_{L^{4k/(k'+1)}} \lesssim |\pi|. \tag{45}$$

This extends to any $J = (j_1, \ldots, j_{k'}, \ldots, j_{k'+m'})$ with $m' \geq 0$ and $1 \leq k', k'+m' \leq 2k-1$, where, as above, $j_{k'} \in \{(1,1), \ldots, (1,d), \ldots, (d,d)\}$ and $j_i \in \{1, \ldots, d\}$ for all other $i \neq k'$. We proceed inductively on $m' = 0, \ldots, 2k - k' - 1$ to show Equation (45) holds for any $J$ of this form in the $L^{4k/(k'+m'+1)}$ norm. The case $m' = 0$ has been covered above. Assume (45) holds for $J_{k'+m}$ with $m \in \{0, \ldots, m'\}$ and $0 \leq m' \leq 2k - k' - 1$, then

$$\left\|S^{J_{k'+m'+1}}((\mathbb{X},\langle\mathbb{X}\rangle)^\pi)_{[0,\tau]} - S^{J_{k'+m'+1}}((\mathbb{X}^\pi,\langle\hat{\mathbb{X}}\rangle^\pi))_{[0,\tau]}\right\|_{L^{4k/(k'+m'+2)}}$$

$$\overset{(i)}{=} \left\|\sum_{i=1}^{m'+1}\frac{1}{i!}\sum_{[u,v]\in\pi_{[0,\tau]}}\left[S^{J_{k'+m'+1-i}}((\mathbb{X},\langle\mathbb{X}\rangle)^\pi)_{[0,u]} - S^{J_{k'+m'+1-i}}((\mathbb{X}^\pi,\langle\hat{\mathbb{X}}\rangle^\pi))_{[0,u]}\right]X_{u,v}^{(j_{k'+m'+2-i})}\cdots X_{u,v}^{(j_{k'+m'+1})}\right.$$

$$+ \sum_{i=m'+2}^{k'+m'+1}\frac{1}{i!}\sum_{[u,v]\in\pi_{[0,\tau]}}S^{J_{k'+m'+1-i}}(\mathbb{X}^\pi)_{[0,u]}$$

$$\left.\times X_{u,v}^{(j_{k'+m'+2-i})}\cdots\left(\langle X\rangle_{u,v}^{(j_{k'})} - \langle\hat{X}\rangle_{u,v}^{\pi,(j_{k'})}\right)\cdots X_{u,v}^{(j_{k'+m'+1})}\right\|_{L^{4k/(k'+m'+2)}}$$

$$\overset{(ii)}{\leq} \left\|\sum_{[u,v]\in\pi_{[0,\tau]}}\left[S^{J_{k'+m'}}((\mathbb{X},\langle\mathbb{X}\rangle)^\pi)_{[0,u]} - S^{J_{k'+m'}}((\mathbb{X}^\pi,\langle\hat{\mathbb{X}}\rangle^\pi))_{[0,u]}\right]X_{u,v}^{(j_{k'+m'+1})}\right\|_{L^{4k/(k'+m'+2)}}$$

$$+ \sum_{i=2}^{m'+1}\frac{1}{i!}\left\|\sum_{[u,v]\in\pi_{[0,\tau]}}\left[S^{J_{k'+m'+1-i}}((\mathbb{X},\langle\mathbb{X}\rangle)^\pi)_{[0,u]} - S^{J_{k'+m'+1-i}}((\mathbb{X}^\pi,\langle\hat{\mathbb{X}}\rangle^\pi))_{[0,u]}\right]\right.$$

$$\left.\times X_{u,v}^{(j_{k'+m'+2-i})}\cdots X_{u,v}^{(j_{k'+m'+1})}\right\|_{L^{4k/(k'+m'+2)}}$$

$$+ \sum_{i=m'+2}^{k'+m'+1}\frac{1}{i!}\left\|\sum_{[u,v]\in\pi_{[0,\tau]}}S^{J_{k'+m'+1-i}}(\mathbb{X}^\pi)_{[0,u]}\right.$$

$$\left.\times X_{u,v}^{(j_{k'+m'+2-i})}\cdots\left(\langle X\rangle_{u,v}^{(j_{k'})} - \left(X_{u,v}^{(j_{k'})}\right)^2\right)\cdots X_{u,v}^{(j_{k'+m'+1})}\right\|_{L^{4k/(k'+m'+2)}}$$

$$\overset{(iii)}{\leq} \sup_{u\in\pi_{[0,\tau]}}\left\|S^{J_{k'+m'}}((\mathbb{X},\langle\mathbb{X}\rangle)^\pi)_{[0,u]} - S^{J_{k'+m'}}((\mathbb{X}^\pi,\langle\hat{\mathbb{X}}\rangle^\pi))_{[0,u]}\right\|_{L^{4k/(k'+m'+1)}}\left(\sum_{[u,v]\in\pi_{[0,\tau]}}\|\mathbf{X}_{u,v}\|_{L^{4k}}^2\right)^{1/2}$$

$$+ \sum_{i=2}^{m'+1}\frac{1}{i!}\sup_{u\in\pi_{[0,\tau]}}\left\|S^{J_{k'+m'+1-i}}((\mathbb{X},\langle\mathbb{X}\rangle)^\pi)_{[0,u]} - S^{J_{k'+m'+1-i}}((\mathbb{X}^\pi,\langle\hat{\mathbb{X}}\rangle^\pi))_{[0,u]}\right\|_{L^{4k/(k'+m'-i+2)}}$$

$$\times\left(\sum_{[u,v]\in\pi_{[0,\tau]}}\|\mathbf{X}_{u,v}\|_{L^{4k}}^i\right)$$

$$+ \sum_{i=m'+2}^{k'+m'+1}\frac{1}{i!}\sup_{u\in\pi_{[0,\tau]}}\left\|S^{J_{k'+m'+1-i}}(\mathbb{X}^\pi)_{[0,u]}\right\|_{L^{4k/(k'+m'+1-i)}}\left(\sum_{[u,v]\in\pi_{[0,\tau]}}\|\mathbf{X}_{u,v}\|_{L^{4k}}^{i-1}\|\langle\mathbf{X}\rangle_{u,v} - \mathbf{X}_{u,v}^{\otimes 2}\|_{L^{2k}}\right)$$

$$\overset{(v)}{\lesssim} |\pi|\sqrt{\tau} + |\pi|\tau \to 0, |\pi| \to 0,$$

where in $(i)$ we have used Lemma B.3, in $(ii)$ triangle inequality, in $(iii)$ Lemma B.1 with the natural filtration of $\mathbb{X}$ for the $i=1$ term, traingle inequality for the $i = 2, \ldots, k' + m' + 1$ terms and Hölder inequality across all terms, in $(iv)$ the

inductive hypothesis for $J_{k'+m}$ with $m = 0, \ldots, m'$ for the first two sups, Remark B.4 applied to $\mathbb{X}$ and $J_i$ with $p = 4k$, $m = 4k/i$ and $|J_i| = i$ for $i = 1, \ldots, k' - 1$ for the third sup and bounds (44) and (A$\alpha$) with $p = 4k$ and $\alpha \geq 1/2$ for the summations over $\pi_{[0,\tau]}$.

We have thus shown that for any word $J \in I \sqcup I$ or $J \in I \sqcup I_{-2} * ((i_{k-1}, i_k))$ or $J = J' * ((i_k, i_k))$ for $J' \in I_{-1} \sqcup I_{-1}$ we have

$$S^J((\mathbb{X}, \langle \mathbb{X} \rangle)^\pi)_{[0,T]} - S^J((\mathbb{X}^\pi, \langle \hat{\mathbb{X}} \rangle^\pi))_{[0,T]} \xrightarrow{L^2} 0, \quad |\pi| \to 0, \tag{46}$$

and hence for fixed $N \geq 1$,

$$\hat{\psi}_J^{N,\pi}(T) - \hat{\psi}_J^{N,\pi,'}(T) \xrightarrow{L^2} 0, \quad |\pi| \to 0.$$

$\square$

# D. Itô processes and diffusions

In this section we consider Itô processes and Itô diffusions: two common classes of models for continuous-time stochastic processes. We start by providing sufficient conditions ensuring Assumption (2.6), needed for the in-fill asymptotics, holds. We then focus on time-homogeneous Itô diffusions and discuss general conditions under which these processes are stationary and strongly mixing, ensuring stationarity and strong mixing of $\{\mathbb{X}^n, \ n \geq 1\}$ under (chop) observations (cf. Proposition 2.13).

## D.1. In-fill conditions

### D.1.1. ITÔ PROCESSES

We consider the case where $\mathbb{X}$ is an Itô process, i.e. satisfies

$$\mathbf{X}_t = \mathbf{X}_0 + \int_0^t \mathbf{b}_s \, \mathrm{d}s + \int_0^t V_s \, \mathrm{d}\mathbf{W}_s, \quad t \in [0, T],$$

where $\mathbf{b} = \{\mathbf{b}_t, \ t \in [0, T]\}$ and $V = \{V_t, \ t \in [0, T]\}$ are progressively measurable $d$- and $d \times q$-dimensional processes such that

$$\sup_{s \in [0,T]} \|\mathbf{b}_s\|_{L^p}, \ \sup_{s \in [0,T]} \|V_s\|_{L^p} < \infty, \tag{47}$$

$\mathbb{W} = \{\mathbf{W}_t, \ t \geq 0\}$ is a $q$-dimensional Brownian motion and $\mathbf{X}_0 \in L^p$. The assumptions on $\mathbf{b}$ and $V$ imply that for all $0 \leq s \leq t \leq T$,

$$\left\| \int_s^t \mathbf{b}_u \mathrm{d}u \right\|_{L^p} \leq \int_s^t \|\mathbf{b}_u\|_{L^p} \, \mathrm{d}u \tag{48}$$

$$\leq \left( \sup_{u \in [0,T]} \|\mathbf{b}_u\|_{L^p} \right) |t - s|, \tag{49}$$

by Minkowski's integral inequality, and

$$\left\| \int_s^t V_u \mathrm{d}\mathbf{W}_u \right\|_{L^p} \lesssim \mathbb{E} \left[ \left( \mathrm{tr} \int_s^t V_u V_u^{\mathrm{T}} \mathrm{d}u \right)^{p/2} \right]^{1/p} = \left\| \int_s^t \|V_u\|^2 \mathrm{d}u \right\|_{L^{p/2}}^{1/2}$$

$$\leq \left( \int_s^t \|V_u\|_{L^p}^2 \, \mathrm{d}u \right)^{1/2} \tag{50}$$

$$\leq \left( \sup_{u \in [0,T]} \|V_u\|_{L^p} \right) |t - s|^{1/2}, \tag{51}$$

by Burkholder-Davis-Gundy (BDG) inequality (Burkholder et al., 1972), the formula for the quadratic variation of the Itô integral, and Minkowski integral inequality. We can show (A$\alpha$) holds with $\alpha = 1/2$ by noting that for all $0 \leq s \leq t \leq T$,

$$\|\mathbf{X}_{s,t}\|_{L^p} \leq \left\| \int_s^t \mathbf{b}_u \mathrm{d}u \right\|_{L^p} + \left\| \int_s^t V_u \mathrm{d}\mathbf{W}_u \right\|_{L^p} \lesssim |t - s| + |t - s|^{1/2} \lesssim |t - s|^{1/2},$$

by combining bounds (49) and (51).

Next, we can show (A$\delta$) holds with $\delta = 1$ by noting that for all $0 \leq s \leq t \leq T$,

$$\|\mathbb{E}_s[\mathbf{X}_{s,t}]\|_{L^p} = \left\|\mathbb{E}_s\left[\int_s^t \mathbf{b}_u \mathrm{d}u\right]\right\|_{L^p} \leq \left\|\int_s^t \mathbf{b}_u \mathrm{d}u\right\|_{L^p} \lesssim |t - s|,$$

where we use the martingale property of Itô integrals, contractive property of conditional expectation and the bound (49).

### D.1.2. ITÔ DIFFUSIONS

Next, assume $\mathbb{X}$ is a (possibly time-inhomogeneous) Itô diffusion, i.e. satisfies

$$d\mathbf{X}_t = f(t, \mathbf{X}_t)\mathrm{d}t + \sigma(t, \mathbf{X}_t)\mathrm{d}\mathbf{W}_t, \quad t \in [0, T],$$

where $\mathbb{W} = \{\mathbf{W}_t, \ t \in [0, T]\}$ is a $q$-dimensional Brownian motion, $f : [0, T] \times \mathbb{R}^d \to \mathbb{R}^d, \sigma : [0, T] \times \mathbb{R}^d \to \mathbb{R}^{d \times q}$ and $\mathbf{X}_0 \in L^p$. Itô diffusions form a subclass of Itô processes, which we already covered in Appendix D.1.1. Here, we give conditions specific to Itô diffusions – i.e. in terms of $f$ and $\sigma$ – which imply condition (47).

- If $f$ and $\sigma$ are uniformly bounded on $[0, T] \times \mathbb{R}^d$, then condition (47) immediately holds.

- Assume $f$ and $\sigma$ are Lipschitz continuous, i.e. for all $s, t \in [0, T]$ and $\mathbf{x}, \mathbf{y} \in \mathbb{R}^d$,

$$\|f(t, \mathbf{x}) - f(s, \mathbf{y})\| \leq K_f \|(t, \mathbf{x}) - (s, \mathbf{y})\| \leq K_f(|t - s| + \|\mathbf{x} - \mathbf{y}\|),$$
$$\|\sigma(t, \mathbf{x}) - \sigma(s, \mathbf{y})\| \leq K_\sigma \|(t, \mathbf{x}) - (s, \mathbf{y})\| \leq K_\sigma(|t - s| + \|\mathbf{x} - \mathbf{y}\|).$$

Then, for all $0 \leq s \leq t \leq T$, we can bound

$$\begin{aligned}
&\|\mathbf{X}_{s,t}\|_{L^p} \\
&= \left\|\int_s^t f(u, \mathbf{X}_u)\mathrm{d}u + \int_s^t \sigma(u, \mathbf{X}_u)\mathrm{d}\mathbf{W}_u\right\|_{L^p} \\
&\overset{(i)}{\lesssim} \int_s^t \|f(u, \mathbf{X}_u)\|_{L^p}\,\mathrm{d}u + \left(\int_s^t \|\sigma(u, \mathbf{X}_u)\|_{L^p}^2\,\mathrm{d}u\right)^{1/2} \\
&\overset{(ii)}{\lesssim} \|f(s, \mathbf{X}_s)\|_{L^p}(t - s) + \int_s^t \|f(u, \mathbf{X}_u) - f(s, \mathbf{X}_s)\|_{L^p}\,\mathrm{d}u \\
&\qquad + \|\sigma(s, \mathbf{X}_s)\|_{L^p}(t - s)^{1/2} + \left(\int_s^t \|\sigma(u, \mathbf{X}_u) - \sigma(s, \mathbf{X}_s)\|_{L^p}^2\,\mathrm{d}u\right)^{1/2} \\
&\overset{(iii)}{\lesssim} \left[\|f(0, \mathbf{0})\| + K_f(s + \|\mathbf{X}_s\|_{L^p})\right](t - s) + K_f \frac{(t - s)^2}{2} + \int_s^t K_f \|\mathbf{X}_{s,u}\|_{L^p}\,\mathrm{d}u \\
&\quad + \left[\|\sigma(0, \mathbf{0})\| + K_\sigma(s + \|\mathbf{X}_s\|_{L^p})\right](t - s)^{1/2} + K_\sigma\left(\int_s^t \|(u, \mathbf{X}_u) - (s, \mathbf{X}_s)\|_{L^p}^2\,\mathrm{d}u\right)^{1/2} \\
&\overset{(iv)}{\lesssim} \left[\|f(0, \mathbf{0})\| + K_f(s + \|\mathbf{X}_s\|_{L^p})\right](t - s) + K_f \frac{(t - s)^2}{2} + \int_s^t K_f \|\mathbf{X}_{s,u}\|_{L^p}\,\mathrm{d}u \\
&\quad + \left[\|\sigma(0, \mathbf{0})\| + K_\sigma(s + \|\mathbf{X}_s\|_{L^p})\right](t - s)^{1/2} + K_\sigma\left(\int_s^t |u - s|^2\mathrm{d}u\right)^{1/2} \\
&\qquad\qquad\qquad\qquad + K_\sigma\left(\int_s^t \|\mathbf{X}_{s,u}\|_{L^p}^2\,\mathrm{d}u\right)^{1/2} \\
&\overset{(v)}{\lesssim} (1 \vee \|\mathbf{X}_s\|_{L^p})(t - s)^{1/2} + \int_s^t \|\mathbf{X}_{s,u}\|_{L^p}\,\mathrm{d}u + \left(\int_s^t \|\mathbf{X}_{s,u}\|_{L^p}^2\,\mathrm{d}u\right)^{1/2} \\
&\overset{(vi)}{\lesssim} (1 \vee \|\mathbf{X}_s\|_{L^p})(t - s)^{1/2} + \left(\int_s^t \|\mathbf{X}_{s,u}\|_{L^p}^2\,\mathrm{d}u\right)^{1/2},
\end{aligned}$$

by using in $(i)$ triangle inequality and Equations (48) and (50), in $(ii)$ triangle inequality, in $(iii)$ Lipschitzianity of $f$ and $\sigma$, in $(iv)$ triangle inequality, in $(v)$ for all $0 \le s \le t \le T$, $s \le T \lesssim 1$ and $(t-s)^{1/2+\epsilon} \le T^\epsilon (t-s)^{1/2} \lesssim (t-s)^{1/2}$ for $\epsilon \ge 0$, and in $(vi)$ Jensen's inequality. Setting $s = 0$, since we assume $\mathbf{X}_0 \in L^p$, this is

$$\|\mathbf{X}_{0,t}\|_{L^p} \lesssim t^{1/2} + \left( \int_0^t \|\mathbf{X}_{0,u}\|_{L^p}^2 \, \mathrm{d}u \right)^{1/2},$$

and we can apply Willett (1964, Lemma 2.2), a nonlinear generalization of the Gronwall inequality, to deduce for all $0 \le t \le T$,

$$\|\mathbf{X}_{0,t}\|_{L^p} \lesssim t^{1/2} + \frac{\left( \int_0^t \exp\{-Cs\} s \, \mathrm{d}s \right)^{1/2}}{1 - \sqrt{1 - \exp\{-Ct\}}}$$
$$\lesssim T^{1/2} + \frac{T}{1 - \sqrt{1 - \exp\{-CT\}}} < \infty.$$

We can hence show the condition for Itô processes (47) holds by noting that $\mathbf{X}_0 \in L^p$ and Lipschitzianity of $f$ and $\sigma$ imply

$$\|f(t, \mathbf{X}_t)\|_{L^p} \le \|f(0, \mathbf{0})\| + K_f(|t| + \|\mathbf{X}_0\|_{L^p} + \|\mathbf{X}_{0,t}\|_{L^p}) < \infty,$$
$$\|\sigma(t, \mathbf{X}_t)\|_{L^p} \le \|\sigma(0, \mathbf{0})\| + K_\sigma(|t| + \|\mathbf{X}_0\|_{L^p} + \|\mathbf{X}_{0,t}\|_{L^p}) < \infty,$$

uniformly in $t \in [0, T]$, and hence (A$\alpha$) and (A$\delta$) hold with $\alpha = 1/2$ and $\delta = 1$, respectively.

- Assume $f$ and $\sigma$ are time-homogeneous such that $f$ is Lipschitz continuous and $\sigma$ is $1/2$-Hölder continuous, i.e. for all $\mathbf{x}, \mathbf{y} \in \mathbb{R}^d$,

$$\|f(\mathbf{x}) - f(\mathbf{y})\| \le K_f \|\mathbf{x} - \mathbf{y}\|,$$
$$\|\sigma(\mathbf{x}) - \sigma(\mathbf{y})\| \le K_\sigma \|\mathbf{x} - \mathbf{y}\|^{1/2}.$$

Then, for all $0 \le s \le t \le T$, we can bound

$$\|\mathbf{X}_{s,t}\|_{L^p} = \left\| \int_s^t f(\mathbf{X}_u) \mathrm{d}u + \int_s^t \sigma(\mathbf{X}_u) \mathrm{d}\mathbf{W}_u \right\|_{L^p}$$
$$\overset{(i)}{\lesssim} \int_s^t \|f(\mathbf{X}_u)\|_{L^p} \, \mathrm{d}u + \left( \int_s^t \|\sigma(\mathbf{X}_u)\|_{L^p}^2 \, \mathrm{d}u \right)^{1/2}$$
$$\overset{(ii)}{\lesssim} \|f(\mathbf{X}_s)\|_{L^p} (t - s) + \int_s^t \|f(\mathbf{X}_u) - f(\mathbf{X}_s)\|_{L^p} \, \mathrm{d}u$$
$$\qquad + \|\sigma(\mathbf{X}_s)\|_{L^p} (t - s)^{1/2} + \left( \int_s^t \|\sigma(\mathbf{X}_u) - \sigma(\mathbf{X}_s)\|_{L^p}^2 \, \mathrm{d}u \right)^{1/2}$$
$$\overset{(iii)}{\lesssim} \left[ \|f(\mathbf{0})\| + K_f \|\mathbf{X}_s\|_{L^p} \right](t - s) + \int_s^t K_f \|\mathbf{X}_{s,u}\|_{L^p} \, \mathrm{d}u$$
$$\qquad + \left[ \|\sigma(\mathbf{0})\| + K_\sigma \|\mathbf{X}_s\|_{L^p}^{1/2} \right](t - s)^{1/2} + K_\sigma^{1/2} \left( \int_s^t \|\mathbf{X}_{s,u}\|_{L^p} \mathrm{d}u \right)^{1/2}$$
$$\overset{(iv)}{\lesssim} \left( 1 \vee \|\mathbf{X}_s\|_{L^p} \vee \|\mathbf{X}_s\|_{L^p}^{1/2} \right)(t - s)^{1/2} + \int_s^t \|\mathbf{X}_{s,u}\|_{L^p}^2 \, \mathrm{d}u + \left( \int_s^t \|\mathbf{X}_{s,u}\|_{L^p} \mathrm{d}u \right)^{1/2},$$

by proceeding as in the previous case. Setting $s = 0$, since we assume $\mathbf{X}_0 \in L^p$, this is

$$\|\mathbf{X}_{0,t}\|_{L^p} \lesssim t^{1/2} + \int_0^t \|\mathbf{X}_{0,u}\|_{L^p} \, \mathrm{d}u + \left( \int_0^t \|\mathbf{X}_{0,u}\|_{L^p} \, \mathrm{d}u \right)^{1/2},$$

and we can apply Dragomir (2003, Theorem 41), another nonlinear generalization of the Gronwall inequality, to deduce for all $0 \leq t \leq T$,

$$\|\mathbf{X}_{0,t}\|_{L^p} \lesssim f(t) < \infty.$$

We can hence show the condition for Itô processes (47) holds by noting that $\mathbf{X}_0 \in L^p$ and the conditions on $f$ and $\sigma$ imply

$$\|f(\mathbf{X}_t)\|_{L^p} \leq \|f(\mathbf{0})\| + K_f(\|\mathbf{X}_0\|_{L^p} + \|\mathbf{X}_{0,t}\|_{L^p}) < \infty,$$
$$\|\sigma(\mathbf{X}_t)\|_{L^p} \leq \|\sigma(\mathbf{0})\| + K_\sigma(\|\mathbf{X}_0\|_{L^p} + \|\mathbf{X}_{0,t}\|_{L^p})^{1/2} < \infty,$$

uniformly in $t \in [0, T]$, and hence (A$\alpha$) and (A$\delta$) hold with $\alpha = 1/2$ and $\delta = 1$ respectively.

### D.2. Long span conditions

#### D.2.1. Itô diffusions

When developing conditions ensuring stationarity and ergodicity of an Itô diffusion it is natural to restrict $\{\mathbf{X}_t,\ t \geq 0\}$ to the case where it is time-homogeneous, i.e. satisfies

$$d\mathbf{X}_t = f(\mathbf{X}_t)dt + \sigma(\mathbf{X}_t)d\mathbf{W}_t, \quad t \geq 0,$$

where $\mathbb{W} = \{\mathbf{W}_t,\ t \geq 0\}$ is a $q$-dimensional Brownian motion, $f : \mathbb{R}^d \to \mathbb{R}^d$ and $\sigma : \mathbb{R}^d \to \mathbb{R}^{d \times q}$. Assume:

- The diffusion coefficient $\sigma : \mathbb{R}^d \to \mathbb{R}^{d \times q}$ is Lipschitz continuous and $\Sigma := \sigma \sigma^{\mathrm{T}} : \mathbb{R}^d \to \mathbb{R}^{d \times d}$ is bounded and uniformly elliptic, i.e.

$$\inf_{x \in \mathbb{R}^d, \xi \in \mathbb{R}^d \setminus \{0\}} \frac{\langle \xi, \Sigma(x)\xi \rangle}{\|\xi\|^2} > 0.$$

  This is a classic PDE condition which ensures the transition densities are "nice", i.e. continuous and bounded away from zero (Friedman, 1964).

- The drift $f : \mathbb{R}^d \to \mathbb{R}^d$ is Lipschitz continuous and has negative radial part at $\infty$, i.e.

$$\limsup_{\|x\| \to \infty} \left\langle f(x), \frac{x}{\|x\|^{\kappa+1}} \right\rangle =: -C_\kappa \in [-\infty, 0),$$

pushing the process towards the origin with strength[21] controlled by $\kappa \in [-1, \infty)$. When $\kappa = -1$ assume further that $2C_{-1} > \sup_{x \in \mathbb{R}^d} \mathrm{Tr}\,\Sigma(x)$ and define

$$\eta_{f,\Sigma}^* := \begin{cases} \infty, & \text{if } \kappa > 0, \\ 2C_0/\|\!|\Sigma|\!\|, & \text{if } \kappa = 0, \\ 2C_\kappa/((1+\kappa)\|\!|\Sigma|\!\|), & \text{if } \kappa \in (-1, 0), \\ \left(2C_{-1} - \sup_{x \in \mathbb{R}^d} \mathrm{Tr}\,\Sigma(x)\right)/\|\!|\Sigma|\!\|, & \text{if } \kappa = -1, \end{cases}$$

where $\|\!|\Sigma|\!\| := \sup_{x \in \mathbb{R}^d} \|\Sigma(x)\|$.

These conditions are enough to ensure there exists a unique invariant probability measure $\mu$ on $\mathbb{R}^d$ with

$$\begin{cases} \int_{\mathbb{R}^d} e^{\eta \|x\|} \mu(dx) < \infty, & \text{if } \kappa \geq 0, \\ \int_{\mathbb{R}^d} e^{\eta \|x\|^{1+\kappa}} \mu(dx) < \infty, & \text{if } \kappa \in (-1, 0), \\ \int_{\mathbb{R}^d} \|x\|^\eta \mu(dx) < \infty, & \text{if } \kappa = -1, \end{cases}$$

---

[21]Note that, for large $\|x\|$, one has $\left\langle f(x), \frac{x}{\|x\|} \right\rangle \approx -C_\kappa \|x\|^\kappa$, and hence the strength of the pull grows as $\|x\|$ increases when $\kappa > 0$ and decays as $\|x\|$ increases when $\kappa < 0$.

for all $\eta \in (0, \eta_{f,\Sigma}^*)$, such that for any $x \in \mathbb{R}^d$ the transition probabilities[22] $\{P_t(x, \cdot), \ t \geq 0\}$ converge to $\mu$ in total variation distance with rates

$$
\|P_t(x, \cdot) - \mu\|_{\text{TV}} \leq \begin{cases} c_1 e^{-c_2 t}(e^{\eta\|x\|} + c_3), & \text{if } \kappa \geq 0, \\ c_1 e^{-c_2 t^{(1+\kappa)/(1-\kappa)}}(e^{\eta\|x\|^{1+\kappa}} + c_3), & \text{if } \kappa \in (-1, 0), \\ c_1(1 + c_2 t)^{-\eta/2}(\|x\|^\eta + c_3), & \text{if } \kappa = -1, \end{cases}
$$

with $c_1, c_2, c_3 > 0$ (Kulik, 2018, Theorem 3.3.4, 3.3.5 and 3.3.6).

Assuming $\mathbf{X}_0 \sim \mu$, the Itô diffusion $\{\mathbf{X}_t, \ t \geq 0\}$ defines a stationary Markov process with $\mathbf{X}_0 \in L^p$ for all $p \geq 2$ when $\kappa > -1$ and for all $2 \leq p < \eta_{f,\Sigma}^*$ when $\kappa = -1$. Recall that, by the discussion in Appendix D.1.2, when $f$ and $\sigma$ are Lipschitz continuous, it is enough to have $\mathbf{X}_0 \in L^p$ to ensure the process $\mathbb{X} = \{\mathbf{X}_t, \ t \in [0, T]\}$ satisfies Assumptions (A$\alpha$) and (A$\delta$) with $\alpha = 1/2$ and $\delta = 1$, implying $\epsilon = 1/2$. By Proposition 2.13 the chain $\{\mathbb{X}^n, \ n \geq 1\}$ is stationary and, hence, it remains to establish strong mixing of $\{\mathbb{X}^n, \ n \geq 1\}$ to apply Theorem 2.10. The strong mixing coefficient of the stationary Markov process $\{\mathbf{X}_t, \ t \geq 0\}$ can be easily[23] bounded by

$$
\alpha(t) \leq \int_{\mathbb{R}^d} \|P_t(x, \cdot) - \mu\|_{\text{TV}} \, \mu(\mathrm{d}x) \lesssim \begin{cases} e^{-c_2 t}, & \text{if } \kappa \geq 0, \\ e^{-c_2 t^{(1+\kappa)/(1-\kappa)}}, & \text{if } \kappa \in (-1, 0), \\ (1 + c_2 t)^{-\eta/2}, & \text{if } \kappa = -1, \end{cases}
$$

and hence the process is strongly mixing. By Proposition 2.13 the chain $\{\mathbb{X}^n, \ n \geq 1\}$ is also strongly mixing with coefficient $\alpha''(n) \leq \alpha((n-3)T), n \geq 3$. It follows immediately that $\{\mathbb{X}^n, \ n \geq 1\}$ is ergodic and hence, we can apply Theorem 2.10.1 to deduce that, letting $|\Pi(N)| \to 0$ as $N \to \infty$ the expected signature estimator (7) is consistent for any expected signature term when $\kappa > -1$ and for all expected signature terms with $|I| < \frac{1}{2}\eta_{f,\Sigma}^*$ when $\kappa = -1$.

Finally, we note that for any $\zeta > 0$

$$
\sum_{n \geq 0} \alpha''(n)^{\zeta/(2+\zeta)} \lesssim \begin{cases} \sum_{n \geq 1} e^{-c_2 nT\zeta/(2+\zeta)} < \infty, & \text{if } \kappa \geq 0, \\ \sum_{n \geq 1} e^{-c_2(nT)^{(1+\kappa)/(1-\kappa)}\zeta/(2+\zeta)} < \infty, & \text{if } \kappa \in (-1, 0), \\ \sum_{n \geq 1} (1 + c_2 nT)^{-\eta\zeta/(4+2\zeta)} = \infty, & \text{if } \kappa = -1, \end{cases}
$$

and hence, if $\kappa > -1$ and $\Pi(N)$ is a sequence of expanding dyadic refinements, we can apply Theorem 2.10.2 to show that the expected signature estimator (7) is also asymptotically normal.

---

[22] For the time-homogeneous Itô diffusion $\{\mathbf{X}_t, \ t \geq 0\}$ these are defined by

$$
P_t(x, A) := \mathbb{P}(\mathbf{X}_t \in A | \mathbf{X}_0 = x) = \mathbb{P}(\mathbf{X}_{s+t} \in A | \mathbf{X}_s = x),
$$

for all $t, s \geq 0$, $x \in \mathbb{R}^d$, $A \in \mathcal{B}(\mathbb{R}^d)$.

[23] If $\{\mathbf{X}_t, \ t \geq 0\}\}$ is a stationary Markov process with stationary distribution $\mu$ and $A \in \mathcal{F}_{-\infty}^t, B \in \mathcal{F}_{t+s}^\infty$ for $t, s \geq 0$,

$$
\begin{aligned}
|\mathbb{P}(A \cap B) - \mathbb{P}(A)\mathbb{P}(B)| &= |\mathbb{E}[\mathbb{1}_A \mathbb{1}_B] - \mathbb{E}[\mathbb{1}_A]\mathbb{E}[\mathbb{1}_B]| \\
&= \left|\mathbb{E}[\mathbb{1}_A \mathbb{E}[\mathbb{E}[\mathbb{1}_B | \mathcal{F}_{-\infty}^{t+s}] | \mathcal{F}_{-\infty}^t]] - \mathbb{E}[\mathbb{1}_A \mathbb{E}[\mathbb{1}_B]]\right| \\
&= \left|\mathbb{E}\left[\mathbb{1}_A \left(\mathbb{E}[\mathbb{E}[\mathbb{1}_B | \mathbf{X}_{t+s}] | \mathbf{X}_t] - \mathbb{E}[\mathbb{E}[\mathbb{1}_B | \mathbf{X}_{t+s}]]\right)\right]\right| \\
&= \left|\mathbb{E}\left[\mathbb{1}_A \left(\int_{\mathbb{R}^d} h_B(x) P_s(\mathbf{X}_t, \mathrm{d}x) - \int_{\mathbb{R}^d} h_B(x)\mu(\mathrm{d}x)\right)\right]\right| \\
&\leq \mathbb{E}\left[\mathbb{1}_A \left|\int_{\mathbb{R}^d} h_B(x)(P_s(\mathbf{X}_t, \mathrm{d}x) - \mu(\mathrm{d}x))\right|\right] \\
&\leq \mathbb{E}\left[\|P_s(\mathbf{X}_t, \cdot) - \mu\|_{\text{TV}}\right] \\
&= \int_{\mathbb{R}^d} \|P_s(x, \cdot) - \mu\|_{\text{TV}} \, \mu(\mathrm{d}x),
\end{aligned}
$$

where $h_B : \mathbb{R}^d \mapsto [0, 1]$ is the measurable function defining the conditional expectation $\mathbb{E}[\mathbb{1}_B | \mathbf{X}_{t+s}] = h(\mathbf{X}_{t+s})$.

# E. Gaussian Processes

We first exploit the properties of Gaussian random variables to show that, if a Gaussian process satisfies (A$\alpha$) for $p = 2$ and some $\alpha > 1/4$, then it satisfies (A$\alpha$) for any $p \geq 2$ and the same $\alpha$. Next, we show that two common examples of Gaussian processes, Ornstein-Uhlenbeck processes and fractional Brownian motion, satisfy the assumptions of Theorem 2.14, i.e. (A$\alpha$) with $p = 2$ and $\alpha \geq 1/2$, (A$\delta$) when $\alpha = 1/2$ and (A$\theta$) for some decreasing $\theta : \mathbb{R}_+ \to \mathbb{R}_+$ with $\theta(t) \to 0$, $t \to \infty$ and $\int_0^T \theta(t)\mathrm{d}t < \infty$ and $m \in \mathbb{N}$.

## E.1. Gaussian Processes Continuity Criterion

Let $\mathbb{X}$ be a Gaussian process with mean function $\mu : [0, T] \to \mathbb{R}^d$. If $\mathbb{X}$ satisfies (A$\alpha$) with $p = 2$ and $\alpha > 1/4$ and $\mu$ is $\alpha$-Hölder continuous, then it satisfies (A$\alpha$) for any $p \geq 2$ and exponent $\alpha$. Note that, by the inclusion of norms in $L^p$ spaces, it suffices to show this holds for arbitrarily large $p$. Choosing $p = 2q$ even, we can write

$$
\|\mathbf{X}_{s,t}\|_{L^{2q}} \leq \|\mathbf{X}_{s,t} - \mu_{s,t}\|_{L^{2q}} + \|\mu_{s,t}\|
$$

$$
\lesssim \left( \mathbb{E}\left[ \left( \sum_{i=1}^d \left| X_{s,t}^{(i)} - \mu_{s,t}^{(i)} \right|^2 \right)^q \right] \right)^{1/2q} + |t-s|^\alpha
$$

$$
= \left( \sum_{\substack{q_1+\ldots+q_d=q \\ q_1,\ldots,q_d \geq 0}} \binom{q}{q_1,\ldots,q_d} \mathbb{E}\left[ \left| X_{s,t}^{(1)} - \mu_{s,t}^{(1)} \right|^{2q_1} \cdots \left| X_{s,t}^{(d)} - \mu_{s,t}^{(d)} \right|^{2q_d} \right] \right)^{1/2q} + |t-s|^\alpha
$$

$$
\overset{(i)}{=} \left( \sum_{\substack{q_1+\ldots+q_d=q \\ q_1,\ldots,q_d \geq 0}} \binom{q}{q_1,\ldots,q_d} \sum_{p \in P_{2q_1,\ldots,2q_d}^2} \prod_{\{i,j\} \in p} \mathrm{Cov}\left( X_{s,t}^{(i)}, X_{s,t}^{(j)} \right) \right)^{1/2q} + |t-s|^\alpha
$$

$$
\overset{(ii)}{\lesssim} \left( \sum_{\substack{q_1+\ldots+q_d=q \\ q_1,\ldots,q_d \geq 0}} \binom{q}{q_1,\ldots,q_d} \sum_{p \in P_{2q_1,\ldots,2m_k}^2} \prod_{\{i,j\} \in p} |t-s|^{2\alpha} \right)^{1/2q} \lesssim |t-s|^\alpha,
$$

where in $(i)$ we apply Isserlis' theorem (Isserlis, 1918) denoting by $P_{2q_1,\ldots,2q_d}^2$ the set of all the pairings of $S = \{1\}^{2q_1} \cup \{2\}^{2q_2} \cup \cdots \cup \{d\}^{2q_d}$, i.e. all distinct ways of partitioning $S$ into $q_1 + \ldots + q_d = q$ pairs, and in $(ii)$ the fact that Assumption (A$\alpha$) with $p = 2$ and $\alpha > 1/4$ implies for all $i, j = 1, \ldots, d$,

$$
\begin{aligned}
\mathrm{Cov}(X_{s,t}^{(i)}, X_{s,t}^{(j)}) &\leq |\mathrm{Cov}(X_{s,t}^{(i)}, X_{s,t}^{(j)})| \\
&\leq |\mathbb{E}[X_{s,t}^{(i)} X_{s,t}^{(j)}]| + |\mu_{s,t}^{(i)} \mu_{s,t}^{(j)}| \\
&\leq \mathbb{E}[|X_{s,t}^{(i)} X_{s,t}^{(j)}|] + |\mu_{s,t}^{(i)}||\mu_{s,t}^{(j)}| \\
&\leq \mathbb{E}[|X_{s,t}^{(i)}|^2]^{1/2}\mathbb{E}[|X_{s,t}^{(j)}|^2]^{1/2} + |\mu_{s,t}^{(i)}||\mu_{s,t}^{(j)}| \\
&\leq \|\mathbf{X}_{s,t}\|^2 + \|\mu_{s,t}\|^2 \lesssim |t-s|^{2\alpha}.
\end{aligned}
$$

## E.2. Gaussian Processes Covariance Decay Condition

### E.2.1. ORNSTEIN-UHLENBECK PROCESS

If $\{\mathbf{X}_t, \ t \geq 0\}$ is a stationary mean-zero $d$-dimensional Ornstein-Uhlenbeck process, i.e. a mean-zero Gaussian process with covariance

$$
C(s,t) := \mathrm{Cov}(\mathbf{X}_s, \mathbf{X}_t) = e^{-A|t-s|}\Sigma,
$$

where $\Sigma = \mathrm{Var}(\mathbf{X}_t)$ and the drift matrix parameter $A \in \mathbb{R}^{d \times d}$ has positive real parts of all eigenvalues, then we can show that:

- (A$\alpha$) holds with $\alpha = 1/2$ and $p = 2$. Note that for all $0 \leq s \leq t \leq T$,

$$
\|\mathbf{X}_{s,t}\|_{L^2}^2 = \mathbb{E}[\mathrm{tr}(\mathbf{X}_{s,t} \otimes \mathbf{X}_{s,t})]
$$

$$
\begin{aligned}
&= \mathrm{tr}(\mathrm{Var}(\mathbf{X}_t) + \mathrm{Var}(\mathbf{X}_s) - 2\mathrm{Cov}(\mathbf{X}_s, \mathbf{X}_t)) \\
&= 2\,\mathrm{tr}((I_d - e^{-A|t-s|})\Sigma) \\
&\leq 2\,\|I_d - e^{-A|t-s|}\|\,\|\Sigma\| \\
&\leq 2\left(\|A\||t-s|e^{\|A\||t-s|}\right)\|\Sigma\| \\
&\lesssim |t-s|.
\end{aligned}
$$

- $(\mathrm{A}\delta)$ holds for any $p \geq 2$ with $\delta = 1$. Using the integral representation of the OU process, one can easily verify that for all $0 \leq s \leq t \leq T$, $\mathbb{E}_s[\mathbf{X}_t] := \mathbb{E}[\mathbf{X}_t|\mathcal{F}_s] = e^{-A|t-s|}\mathbf{X}_s$, where $\{\mathcal{F}_t,\ t \in [0,T]\}$ is the natural filtration of $\mathbb{X}$. Then, for all $0 \leq s \leq t \leq T$,

$$
\|\mathbb{E}_s[\mathbf{X}_{s,t}]\|_{L^p} = \|(e^{-A|t-s|} - I_d)\mathbf{X}_s]\|_{L^p} \leq \|I_d - e^{-A|t-u|}\|\,\|\mathbf{X}_s\|_{L^p} \lesssim |t-s|.
$$

- The covariance of the increments is homogeneous since, for all $u, v, s, t \geq 0$,

$$
\begin{aligned}
\mathrm{Cov}\left(\mathbf{X}_{u,v}, \mathbf{X}_{s,t}\right) &= \mathrm{Cov}\left(\mathbf{X}_v, \mathbf{X}_t\right) - \mathrm{Cov}\left(\mathbf{X}_v, \mathbf{X}_s\right) - \mathrm{Cov}\left(\mathbf{X}_u, \mathbf{X}_t\right) + \mathrm{Cov}\left(\mathbf{X}_u, \mathbf{X}_s\right) \\
&= (e^{-A|t-v|} - e^{-A|s-v|} - e^{-A|t-u|} + e^{-A|s-u|})\Sigma,
\end{aligned}
$$

depends only on the relative distances $|t-v|, |s-v|, |t-u|$ and $|s-u|$.

- The covariance of the increments satisfies Assumption $(\mathrm{A}\theta)$ with $m = 0$ and $\theta(t) = e^{-\lambda_A t}$ where $\lambda_A$ is a constant depending on the drift matrix $A \in \mathbb{R}^{d \times d}$. For all $0 \leq u \leq v < s \leq t$,

$$
\begin{aligned}
\|\mathrm{Cov}\left(\mathbf{X}_{u,v}, \mathbf{X}_{s,t}\right)\| &= \|e^{-A|s-v|}(e^{-A|t-s|} - I_d - e^{-A|t-s|-A|v-u|} + e^{-A|v-u|})\Sigma\| \\
&= \|e^{-A|s-v|}(e^{-A|t-s|} - I_d)(I_d - e^{-A|v-u|})\Sigma\| \\
&= \|e^{-A|s-v|}\|\,\|I_d - e^{-A|t-s|}\|\,\|I_d - e^{-A|v-u|}\|\,\|\Sigma\| \\
&\lesssim e^{-\lambda_A|s-v|}|t-s||v-u|,
\end{aligned}
$$

where, in the last step, we use the fact that $A \in \mathbb{R}^{d \times d}$ has positive real parts of all eigenvalues to find $\lambda_A \in (0, \min_{\lambda \in \sigma(A)} \mathrm{Re}(\lambda))$ such that $\|e^{-At}\| \lesssim e^{-\lambda_A t}$ for all $t \geq 0$.

Note that $(\mathrm{A}\alpha)$ and $(\mathrm{A}\delta)$ could have been alternatively established by noting that the OU process is an Itô diffusion with Lipschitz continuous coefficients and applying the results of Appendix D.1.2.

### E.2.2. FRACTIONAL BROWNIAN MOTION

If $\{X_t^H,\ t \geq 0\}$ is a (one-dimensional) fractional Brownian motion with Hurst parameter $H > 1/2$, i.e. a mean-zero Gaussian process with covariance

$$
C^H(s,t) := \mathrm{Cov}(X_s^H, X_t^H) = \frac{1}{2}(|t|^{2H} + |s|^{2H} - |t-s|^{2H}),
$$

then we can show that:

- $(\mathrm{A}\alpha)$ holds with $\alpha = H > 1/2$ (and $p = 2$) since $\|X_{s,t}^H\|_{L^2} = |t-s|^H$.

- The covariance of the increments is homogeneous since for all $u, v, s, t \geq 0$,

$$
\begin{aligned}
\mathrm{Cov}\left(X_{u,v}^H, X_{s,t}^H\right) &= \mathrm{Cov}\left(X_v^H, X_t^H\right) - \mathrm{Cov}\left(X_v^H, X_s^H\right) - \mathrm{Cov}\left(X_u^H, X_t^H\right) + \mathrm{Cov}\left(X_u^H, X_s^H\right) \\
&= \frac{1}{2}(|s-v|^{2H} + |t-u|^{2H} - |t-v|^{2H} - |s-u|^{2H}),
\end{aligned}
$$

depends only on the relative distances $|t-v|, |s-v|, |t-u|$ and $|s-u|$.

- The covariance of the increments satisfies Assumption (A$\theta$) with $m = 3$ and $\theta(t) = t^{2H-2}$. For all $0 \leq u \leq v < s \leq t$ with $|s - v| \geq 3/2(|t - s| + |v - u|)$,

$$
\left| \text{Cov} \left( X_{u,v}^H, X_{s,t}^H \right) \right|
$$
$$
= \frac{1}{2} \left| |s - v|^{2H} + |t - u|^{2H} - |t - v|^{2H} - |s - u|^{2H} \right|
$$
$$
= \frac{1}{2} \Bigg| |s - v|^{2H}
$$
$$
+ \sum_{n=0}^{\infty} \frac{(2H) \cdots (2H - n + 1)}{n!} |s - v|^{2H-n} (|t - u| - |s - v|)^n
$$
$$
- \sum_{n=0}^{\infty} \frac{(2H) \cdots (2H - n + 1)}{n!} |s - v|^{2H-n} (|t - v| - |s - v|)^n
$$
$$
- \sum_{n=0}^{\infty} \frac{(2H) \cdots (2H - n + 1)}{n!} |s - v|^{2H-n} (|s - u| - |s - v|)^n \Bigg|
$$
$$
= \frac{1}{2} \left| \sum_{n=2}^{\infty} \frac{(2H) \cdots (2H - n + 1)}{n!} |s - v|^{2H-n} \left[ (|t - s| + |v - u|)^n - |t - s|^n - |v - u|^n \right] \right|
$$
$$
= \frac{1}{2} |s - v|^{2H-2} \left| \sum_{n=2}^{\infty} \frac{(2H) \cdots (2H - n + 1)}{n!} |s - v|^{-(n-2)} \sum_{j=1}^{n-1} \binom{n}{j} |t - s|^j |v - u|^{n-j} \right|
$$
$$
\leq \frac{1}{2} |s - v|^{2H-2} |t - s| |v - u|
$$
$$
\times \sum_{n=2}^{\infty} \frac{|(2H) \cdots (2H - n + 1)|}{(n-2)!} |s - v|^{-(n-2)} \sum_{j=0}^{n-2} \binom{n-2}{j} |t - s|^j |v - u|^{(n-2)-j}
$$
$$
\leq \frac{1}{2} |s - v|^{2H-2} |t - s| |v - u| |2H| |2H - 1| \sum_{n=0}^{\infty} |s - v|^{-n} (|t - s| + |v - u|)^n
$$
$$
\leq \frac{1}{2} |s - v|^{2H-2} |t - s| |v - u| |2H| |2H - 1| \left( 1 - \frac{|t - s| + |v - u|}{|s - v|} \right)^{-1}
$$
$$
\leq |2H| |2H - 1| |s - v|^{2H-2} |t - s| |v - u|,
$$

by using the fact that that $x \mapsto f(x) = x^{2H}$ is analytic for any $x > 0$ and $|s - v| \geq 3/2(|t - s| + |v - u|)$.

## F. Machine Learning Algorithms with Expected Signatures

Signatures of paths have found widespread use in the machine learning community, with applications ranging from character recognition (Graham, 2013; Xie et al., 2018) to medical diagnosis (Pérez Arribas et al., 2018). Taking the signature of a stream of data is essentially a feature extraction method mapping raw stream-like data to a lower-dimensional but highly-informative latent space. The theoretical foundations for their efficacy range from the characterization result of Hambly & Lyons (2005) to the universal approximation theorem (Levin et al., 2016, Theorem 3.1). As discussed in the introduction, when dealing with collections of paths, the characterization results of Fawcett (2003), Chevyrev & Lyons (2016) and Chevyrev & Oberhauser (2018) give strong theoretical justification for the use of the expected signature as a feature extraction method.

While dealing with a collection of paths is arguably a less common setting than a single stream of data, the literature still provides a wide range of machine learning algorithms leveraging expected signatures[24]. These cover many different tasks (from distributional regression to generative modeling) and applications (from ECG classification to option pricing). In this section, we review five of these machine learning algorithms discussing how the martingale correction introduced in Section 2.2 can be applied to the expected signature computation step to improve performance. Before diving into each

---

[24]The GPES algorithms, discussed in Section 3.2.1, actually takes as input a single stream of data and applies a data augmentation technique to form a collection of paths.

specific application in detail, we discuss some general considerations on the use of the martingale correction term in practice.

### F.1. Martingale Correction in Applications

In Section 2.2, we considered the same framework as in the rest of this work, namely the setting where $\mathbb{X}$ is a continuous-time stochastic process and $\mathbb{X}^\pi$ is a piecewise-linear interpolation of the discrete-time observation of such process along the partition $\pi$. It is important to note that, while some applications have a natural underlying latent continuous-time model (Lyons et al., 2021) others do not (Lemercier et al., 2021). In any case, we do not necessarily require the "background" continuous-time model $\mathbb{X}$ to be defined to apply the control-variate estimator (40) with $c = \hat{c}_{1,\pi}^*$. Letting $\pi = \{0 = t_0 < t_1 < \cdots < t_M = T\}$, one can easily see that it is sufficient for the discrete-time process $\{\mathbf{X}_m = \mathbf{X}_{t_m},\ m = 0, \ldots, M\}$ to be a discrete-time martingale with respect to $\mathcal{F}_m = \sigma(\mathbf{X}_1, \ldots, \mathbf{X}_m)$ for the control variate estimator to have the same bias but lower variance than the naive expected signature estimator. In what follows all path observations are inevitably sampled at discrete points in time and hence, abusing notation slightly, in some places we drop the dependence on the partition $\pi$. Whether $\mathbb{X}$ is a continuous-time process or a sequence of observations in discrete time should be clear from the context.

Machine learning methods based on signature methods often apply augmentations to the raw streams of data before computing the signature, cf. Lyons & McLeod (2024, Section 2.5). For example, a path augmentation which is often found to improve model performance is the lead-lag transform. Combining the previous observation on discrete-time martingales with Remark 2.15, we can easily see that the control variate expected signature estimator can also be employed when the lead-lag augmentation is applied to the raw data, i.e. when the $d$-dimensional discrete-time martingale $\{\mathbf{X}_m,\ m = 1, \ldots, M\}$ is embedded into the $2d$-dimensional process

$$\mathbb{X}' = \{(\mathbf{X}_1, \mathbf{X}_1), (\mathbf{X}_2, \mathbf{X}_1), (\mathbf{X}_2, \mathbf{X}_2), \ldots, (\mathbf{X}_M, \mathbf{X}_{M-1}), (\mathbf{X}_M, \mathbf{X}_M)\} = \{\mathbf{X}'_{m'},\ m' = 2, \ldots, 2M + 2\}.$$

Note that for each $m' = 2, \ldots, 2M + 2$, if $m' = 2m$ then $\mathbf{X}'_{m'} = (\mathbf{X}_m, \mathbf{X}_m)$, $\mathbf{X}'_{m'+1} = (\mathbf{X}_{m+1}, \mathbf{X}_m)$ and

$$\mathbb{E}[\mathbf{X}'_{m'+1} | \mathbf{X}'_1, \ldots, \mathbf{X}'_{m'}] = (\mathbb{E}[\mathbf{X}_{m+1} | \mathbf{X}_1, \ldots, \mathbf{X}_m], \mathbf{X}_m) = (\mathbf{X}_m, \mathbf{X}_m),$$

and if $m' = 2m + 1$ then $\mathbf{X}'_{m'} = (\mathbf{X}_{m+1}, \mathbf{X}_m)$, $\mathbf{X}'_{m'+1} = (\mathbf{X}_{m+1}, \mathbf{X}_{m+1})$ and

$$\mathbb{E}[\mathbf{X}'_{m'+1} | \mathbf{X}'_1, \ldots, \mathbf{X}'_{m'}] = (\mathbf{X}_{m+1}, \mathbf{X}_{m+1}).$$

and hence the leading components, i.e. the first $d$ entries of $\mathbb{X}'$, form a discrete-time martingale with respect to the natural filtration of $\mathbb{X}'$. By Remark 2.15, we can hence apply the control variate expected signature estimator (40) for any word $I = (i_1, \ldots, i_k) \in \{1, \ldots, 2d\}^k$ with $i_k \in \{1, \ldots, d\}$.

Finally, in some applications, we may not have a strong prior on whether the process being modeled is a martingale or not. In this case, we may consider the martingale correction as a model configuration hyperparameter to be tuned, just like the lead-lag path augmentation discussed above. In the model training phase we can then apply a cross-validation procedure to learn whether applying the martingale correction to (some of) the expected signature terms improves the performance of the model.

*Remark* F.1. Both the signature transform and the expected signature transform are general methods applicable to any machine learning task dealing with (collections of) streams of data. These can thus always be used as out-of-the-box feature extraction methods when little domain knowledge is available. On the other hand, when task-specific information is known, incorporating such knowledge in the machine learning model will most likely improve performance.

*Remark* F.2. It is important to note there is also a wide range of machine learning methods based on signature kernels (Kiraly & Oberhauser, 2019; Chevyrev & Oberhauser, 2018; Lemercier et al., 2021; Salvi et al., 2021). This bypasses the need to explicitly estimate the expected signature and hence we cannot directly apply the martingale correction developed in Section 2.2.

### F.2. Algorithms

#### F.2.1. TIME SERIES CLASSIFICATION (TRIGGIANO & ROMITO, 2024)

In the Gaussian Process augmented Expected Signature (GPES) classifier, the input stream $\mathbf{x} \in \mathbb{R}^{d \times M_1}$ is interpreted as a discrete-time realization of a Gaussian process $\mathbb{X} \sim \mathrm{GP}(\mu(t), \Sigma(t))$ at points $\pi_1 = \{0 = t_1 < \ldots < t_{M_1} = T\}$, i.e. a realization of $\mathbb{X}^\pi$. Values of the process over a fixed set of in-fill points $\pi_2 = \{s_1 < \ldots < s_{M_2}\}$ can thus be

sampled[25] from the conditional distribution of $\mathbb{X}^{\pi_2}$ given the input $\mathbb{X}^{\pi_1}$, i.e. the conditional distribution $\mathbb{X}^{\pi_2}|\mathbb{X}^{\pi_1} = \mathbf{x} \sim \mathcal{N}(\mu_{\mathbf{x},\pi_1,\pi_2}, \Sigma_{\mathbf{x},\pi_1,\pi_2})$. The expected signature of the process $\mathbb{X}$ is then estimated from a collection of samples $\mathbb{X}^{1,\pi_1\cup\pi_2}, \ldots, \mathbb{X}^{N,\pi_1\cup\pi_2}$ such that $\mathbb{X}^{n,\pi_1} = \mathbf{x}$ and $\mathbb{X}^{n,\pi_2} \sim \mathcal{N}(\mu_{\mathbf{x},\pi_1,\pi_2}, \Sigma_{\mathbf{x},\pi_1,\pi_2})$ for $n = 1, \ldots, N$. In Triggiano & Romito (2024), the authors emphasize the theoretical and empirical importance of the tensor normalization introduced in Chevyrev & Oberhauser (2018), ensuring the resulting expected robust signature is characteristic for a larger class of processes. An important component of the GPES model is thus the (truncated at level $k$) tensor normalization $\lambda_C : T^K(\mathbb{R}^d) \to T^K(\mathbb{R}^d)$, controlled by the hyperparameter $C$. For more details on the effect of the tensor normalization procedure and a sensitivity analysis[26] with respect to the hyper-parameter $C$ we refer to Triggiano & Romito (2024). When applying the martingale correction to the GPES model[27] we subtract the correction term $\hat{c}_1^* S_c^{\mathbf{I}}(\mathbb{X}^{k,\pi_1\cup\pi_2})_{[0,T]}$ to each $S^{\mathbf{I}}(\mathbb{X}^{k,\pi_1\cup\pi_2})_{[0,T]}$, i.e. *before* taking the empirical expectation over the paths. This modification of the original algorithm highlighted in green in Algorithm 1. The final layer of the GPES model then maps the expected signature to a class by a combination of a linear transformation and a softmax output activation. The forward pass through the GPES model is summarized in Algorithm 1.

---

**Algorithm 1** Gaussian Process augmented Expected Signature (GPES) classifier, forward pass

---

**hyperparameters** Signature truncation level $k \in \mathbb{N}$, tensor normalization parameter $C \in \mathbb{R}_+$, data augmentation size $N \in \mathbb{N}$, in-fill partition $\pi_2 \in \Delta_{[0,T]}^{M_2}$ s.t. $M_2 \in \mathbb{N}$.

**parameters** Biases $\mathbf{b}_\mu \in \mathbb{R}^{d_\mu}$, $\mathbf{b}_\Sigma \in \mathbb{R}^{d_\Sigma}$, $\mathbf{b}_{\text{out}} \in \mathbb{R}^{d_{\text{out}}}$ and weights $W_\mu \in \mathbb{R}^{d_\mu \times d_{\text{in}}}$, $W_\Sigma \in \mathbb{R}^{d_\Sigma \times d_{\text{in}}}$, $W_{\text{out}} \in \mathbb{R}^{d_{\text{out}} \times d_{\text{sig}}}$ where $d_{\text{in}} \leftarrow (d M_1 + M_1 + M_2)$, $d_\mu \leftarrow d M_2$, $d_\Sigma \leftarrow d M_2(d M_2 + 1)/2$, $d_{\text{sig}} \leftarrow (d + \ldots + d^k)$ and $d_{\text{out}} \leftarrow |\mathcal{C}|$.

**input** $\mathbf{x} \in \mathbb{R}^{d \times M_1}$ and $\pi_1 \in \Delta_{[0,T]}^{M_1}$.

1: $\mu_{\mathbf{x},\pi_1,\pi_2} \leftarrow \mathbf{b}_\mu + W_\mu(\mathbf{x}, \pi_1, \pi_2)$.
2: $L_{\mathbf{x},\pi_1,\pi_2} \leftarrow \mathbf{b}_\Sigma + W_\Sigma(\mathbf{x}, \pi_1, \pi_2)$ and $\Sigma_{\mathbf{x},\pi_1,\pi_2} \leftarrow L_{\mathbf{x},\pi_1,\pi_2} L_{\mathbf{x},\pi_1,\pi_2}^{\mathsf{T}}$.
3: **for** $n \in \{1, \ldots, N\}$ **do**
4:     $\mathbb{X}^{n,\pi_1} \leftarrow \mathbf{x}$.
5:     Sample $\mathbb{X}^{n,\pi_2} \sim \mathcal{N}(\mu_{\mathbf{x},\pi_1,\pi_2}, \Sigma_{\mathbf{x},\pi_1,\pi_2})$.
6:     Signature of $\mathbb{X}^{n,\pi_1\cup\pi_2}$: $\mathbf{S}^n = S^{\leq k}(\mathbb{X}^{n,\pi_1\cup\pi_2})_{[0,T]} - \hat{c}_1^* S_c^{\leq k}(\mathbb{X}^{n,\pi_1\cup\pi_2})_{[0,T]} \in \mathbb{R}^{d_{\text{sig}}}$.
7:     Tensor normalization: $\mathbf{S}^n \leftarrow \lambda_C(\mathbf{S}^n)$.
8: **end for**
9: Expected signature $\mathbf{ES} \leftarrow \dfrac{1}{N} \sum_{n=1}^N \mathbf{S}^n$.

**output** $\hat{c} \leftarrow \text{softmax}(\mathbf{b}_{\text{out}} + W_{\text{out}}\mathbf{ES})$.

---

Note that, unlike classic Gaussian process regression where the prior mean is assumed to be constant $\mu(t) \equiv \mu$ and the prior covariance function

$$\Sigma : [0, T] \to \mathbb{R}^{d \times d},$$

is parameterized by a kernel and posteriors are computed by combining standard properties of the multivariate normal distribution and the kernel trick[28], in the GPES model the conditional mean and covariance functions

$$(\mathbf{x}, \pi_1, \pi_2) \in \mathbb{R}^{d \times M_1} \times \Delta_{[0,T]}^{M_1} \times \Delta_{[0,T]}^{M_2} \cong \mathbb{R}^{d M_1 + M_1 + M_2} \quad \mapsto \quad \begin{cases} \mu_{\mathbf{x},\pi_1,\pi_2} \in \mathbb{R}^{d \times M_2} \cong \mathbb{R}^{d M_2}, \\ \Sigma_{\mathbf{x},\pi_1,\pi_2} \in \mathcal{M}(\mathbb{R}^{d \times M_2}) \cong \mathbb{R}^{d M_2 \times d M_2}, \end{cases}$$

are parametrized by linear transformations

$$\mu_{\mathbf{x},\pi_1,\pi_2} = \mathbf{b}_\mu + W_\mu(\mathbf{x}, \pi_1, \pi_2),$$
$$\Sigma_{\mathbf{x},\pi_1,\pi_2} = L_{\mathbf{x},\pi_1,\pi_2} L_{\mathbf{x},\pi_1,\pi_2}^{\mathsf{T}}, \quad L_{\mathbf{x},\pi_1,\pi_2} = \mathbf{b}_\Sigma + W_\Sigma(\mathbf{x}, \pi_1, \pi_2),$$

---

[25]Super-sampling the input data to a collection of realizations from a Gaussian process, can be effectively understood as a regularization by noise technique.

[26]Choosing a very large value of $C$ is in practice equivalent to not applying a tensor normalization.

[27]Note that the GPES algorithm estimates the expected signature conditional on $\mathbb{X}^{n,\pi_1} = \mathbf{x}$, so technically the martingale correction is biasing the estimator.

[28]The Gaussian process regression model is then fitted by tuning the kernel hyperparameters (either via maximum likelihood or cross-validation).

where $L_{\mathbf{x},\pi_1,\pi_2}$ is a lower triangular matrix. The parameters $\mathbf{b}_\mu, \mathbf{b}_\Sigma, W_\mu, W_\Sigma$ are learned along with the output layer parameters $\mathbf{b}_{\text{out}}, W_{\text{out}}$ in the training phase via numerical optimization, in Triggiano & Romito (2024) the authors use simple stochastic gradient descent (SGD). If the timestamps $\pi_1$ of the observations $\mathbf{x}$ are not provided, they need to be fixed and can be regarded as hyperparameters of the model. The (way the) in-fill partition (is chosen) is instead always chosen a-priori, with a natural choice for $\pi_2$ being the set of mid-points of $\pi_1$. Other model hyperparameters include the signature truncation level $k$, the size of the augmentation $N$ and the constant $C$ controlling the strength of the tensor normalization, as well as the training procedure's hyperparameters (learning rate, batch size etc.).

In Section 3.2.1 we replicate the synthetic data experiments of Triggiano & Romito (2024). These consist of three datasets:

(FBM) Two equally balanced classes with samples generated according to a standard Brownian motion and a fractional Brownian motion with Hurst parameter $H = 0.26$ (both in dimension $d = 1$).

(OU) Two equally balanced classes with samples generated according to two different Ornstein-Uhlenbeck (OU) processes (both in dimension $d = 1$).

(Bidim) Six equally balanced classes with samples generated according to six different bi-dimensional stochastic processes ($d = 2$).

When fitting the models we take the optimal hyperparameters cross-validated by Triggiano & Romito (2024)[29] and apply cross-validated SGD to the training dataset. That is, we use 80% of the training dataset to iterate through SGD parameter updates, while keeping the remaining 20% of the training dataset (the validation set) to determine when the procedure has converged without overfitting. As described in (Triggiano & Romito, 2024) the presence of the tensor normalization step often leads to exploding gradients in the training procedure. We thus repeat the SGD routine over 5 different parameter initializations and pick the model with best validation performance.

### F.2.2. PRICING PATH-DEPENDENT DERIVATIVES (LYONS ET AL., 2021)

While the authors of Lyons et al. (2021) consider a more general setting, for brevity, we focus on the case where the (discounted) price process $\mathbb{X}$ is assumed to be a semimartingale. Let $\mathbb{X} = \{\mathbf{X}_t,\ t \in [0,T]\}$ be a semimartingale on the probability space $(\Omega = C([0,T],\mathbb{R}), \mathcal{F}, \mathbb{P})$ and denote by $\hat{\Omega}_T^{\text{LL}}$ the set of realized (time and lead-lag augmented) price signatures, i.e.

$$\hat{\Omega}_T^{\text{LL}} = \{S(\hat{\mathbb{X}}^{\text{LL}})_{[0,T]} \in T((\mathbb{R}^4)) : \hat{\mathbb{X}} = \{(t, \mathbf{X}_t),\ t \geq 0\}\},$$

where we refer to Lyons et al. (2021, Definition 2.14 and Example 2.15) for the definition of the lead-lag augmentation but, for the purposes of our discussion, it is sufficient to note that it is uniquely defined through Stratonovich integration. The authors then consider the market $(\hat{\Omega}_T^{\text{LL}}, \mathcal{B}(\hat{\Omega}_T^{\text{LL}}), \{\mathcal{F}_t,\ t \in [0,T]\}, \hat{\mathbb{P}}^{\text{LL}})$ where $\hat{\mathbb{P}}^{\text{LL}}$ is the push-forward of $\mathbb{P}$ onto $\hat{\Omega}_T^{\text{LL}}$. By defining the set of derivative payoffs as all measurable $F : \hat{\Omega}_T^{\text{LL}} \to \mathbb{R}$, i.e. for a given price realization $\mathbb{X}$ the holder of the derivative receives $F(S(\hat{\mathbb{X}}^{\text{LL}})_{[0,T]})$, in Lyons et al. (2021, Proposition 4.5) the authors use the universality of the signature to show that any *continuous* payoff $F$ can be arbitrarily well approximated by a linear payoff[30]. In particular, this implies the price of any such $F$ can be decomposed as

$$\mathbb{E}^{\mathbb{Q}}[Z_T F(S(\hat{\mathbb{X}}^{\text{LL}})_{[0,T]})] \approx \langle f, Z_T \mathbb{E}^{\mathbb{Q}}[S(\hat{\mathbb{X}}^{\text{LL}})_{[0,T]}]\rangle,$$

for a set of linear coefficients $f \in T((\mathbb{R}^4)^*)$ where $\mathbb{Q}$ is a pricing measure for $\mathbb{X}$ and $Z_T$ is a deterministic discount factor over $[0,T]$. The set of signature payoffs $\{S^I(\hat{\mathbb{X}}^{\text{LL}})_{[0,T]},\ I \in \mathcal{W}(\{1,2,3,4\})\}$ can thus be understood as a set of Arrow-Debreu securities spanning the set of continuous path-dependent derivatives $F$. Similarly, in Lyons et al. (2021, Proposition 4.6) the authors show that linear trading strategies are dense in the space of admissible trading strategies $\mathcal{A}$, here defined as the set of all continuous functions $\theta : S(\hat{\mathbb{X}})_{[0,t]} \mapsto \theta(S(\hat{\mathbb{X}})_{[0,t]})$ over the stopped at $t \in [0,T]$ time-augmented price path signatures, i.e.

$$\theta(S(\hat{\mathbb{X}})_{[0,t]}) \approx \langle \ell, S(\hat{\mathbb{X}})_{[0,t]}\rangle, \quad \forall t \in [0,T],$$

---

[29]The only hyperparameter we modify is the truncation level $k$ which we set to 4 for computational reasons (in the original paper the optimal value was found to be 5 or 6, depending on the dataset). The hyperparameters an the training and testing routines used to produce the results in Table 1 can be found at https://github.com/lorenzolucchese/gp-esig-classifier.

[30]Note that the approximation of $F$ by $f$ does not depend on the choice of probability measure $\mathbb{P}$, i.e. it is a pathwise density result. In Algorithm 2 and Algorithm 3 we thus assume that $f$ has been estimated offline (i.e. for any model) by linearly regressing $F(\omega)$ against $\omega$ for a large set of $\omega \in \hat{\Omega}_T^{\text{LL}}$.

for a set of linear coefficients $\ell \in T((\mathbb{R}^2)^*)$. These two results are then combined in Lyons et al. (2021, Theorem 4.7) to show that the solution of the quadratic $\mathbb{P}$-hedging problem[31]

$$\theta^* = \underset{\theta \in \mathcal{A}}{\operatorname{argmin}} \, \mathbb{E}^{\mathbb{P}}\left[\left(F(S(\hat{\mathbb{X}}^{\mathrm{LL}})_{[0,T]}) - p_0 - \int_0^T \theta(S(\hat{\mathbb{X}})_{[0,t]})\mathrm{d}\mathbf{X}_t\right)^2\right], \tag{52}$$

can be arbitrarily well approximated by the solution of a linear signature quadratic hedging problem, i.e.

$$\theta^*(S(\hat{\mathbb{X}})_{[0,t]}) \approx \langle \ell^*, S(\hat{\mathbb{X}})_{[0,t]}\rangle, \quad t \in [0, T],$$

where $\ell^* \in T((\mathbb{R}^2)^*)$ can be computed by

$$\ell^* = \underset{\ell \in T((\mathbb{R}^2)^*)}{\operatorname{argmin}} \, \langle (f - p_0 \varnothing + \ell\mathbf{4})^{\sqcup\!\sqcup 2}, \mathbb{E}^{\mathbb{P}}[S(\hat{\mathbb{X}}^{\mathrm{LL}})_{[0,T]}]\rangle.$$

---

**Algorithm 2** Pricing Path-Dependent Derivatives with Expected Signatures

---

**hyperparameters** Signature truncation levels $k$, number of Monte Carlo samples $N \in \mathbb{N}$.
**parameters** Risk-neutral measure $\mathbb{Q}$, deterministic discount factor $Z_T \in \mathbb{R}^+$, linear approximator $f \in T^k((\mathbb{R}^4)^*)$.
**input** Derivative payoff $F: \hat{\Omega}_T^{\mathrm{LL}} \to \mathbb{R}$.
 1: Sample $N$ trajectories $\mathbb{X}^1, \ldots, \mathbb{X}^N \sim \mathbb{Q}$.
 2: Compute time-augmented lead-lag transforms $\hat{\mathbb{X}}^{n,\mathrm{LL}}$ for $n \in \{1, \ldots, N\}$.
 3: Compute $\mathbf{S}^n \in T^k((\mathbb{R}^4))$ s.t. $\mathbf{S}_I^n \leftarrow S^I(\hat{\mathbb{X}}^{n,\mathrm{LL}})_{[0,T]}, |I| \leq k$ for $n \in \{1, \ldots, N\}$.
 4: Estimate $\Phi \in T^k((\mathbb{R}^4))$ s.t. $\Phi_I \leftarrow \hat{\phi}_I^{N,\hat{c}_1}(T), |I| \leq k$ from $\{\hat{\mathbb{X}}^{n,\mathrm{LL}}\}_{n=1}^N$.
**output** Price $p = \langle f, Z_T\Phi\rangle$.

---

**Algorithm 3** Hedging Path-Dependent Derivatives with Expected Signatures

---

**hyperparameters** Signature truncation levels $k$, number of Monte Carlo samples $N \in \mathbb{N}$.
**parameters** Real-world measure $\mathbb{P}$, initial capital $p_0 \in \mathbb{R}$, linear approximator $f \in T^k((\mathbb{R}^4)^*)$.
**input** Derivative payoff $F: \hat{\Omega}_T^{\mathrm{LL}} \to \mathbb{R}$.
 1: Sample $N$ trajectories $\mathbb{X}^1, \ldots, \mathbb{X}^N \sim \mathbb{P}$.
 2: Compute time-augmented lead-lag transforms $\hat{\mathbb{X}}^{n,\mathrm{LL}}$ for $n \in \{1, \ldots, N\}$.
 3: Compute $\mathbf{S}^n \in T^k((\mathbb{R}^4))$ s.t. $\mathbf{S}_I^n \leftarrow S^I(\hat{\mathbb{X}}^{n,\mathrm{LL}})_{[0,T]}, |I| \leq k$ for $n \in \{1, \ldots, N\}$.
 4: Estimate $\Phi \in T^k((\mathbb{R}^4))$ s.t. $\Phi_I \leftarrow \hat{\phi}_I^{N,\hat{c}_1}(T), |I| \leq k$ from $\{\hat{\mathbb{X}}^{n,\mathrm{LL}}\}_{n=1}^N$.
 5: $\hat{\ell}^* \leftarrow \inf_{\ell \in T^{\lfloor k/2 \rfloor}((\mathbb{R}^2)^*)} \langle (f - p_0\varnothing + \ell\mathbf{4})^{\sqcup\!\sqcup 2}, \Phi\rangle$.
**output** Hedging strategy $t \mapsto \langle \hat{\ell}^*, S(\hat{\mathbb{X}})_{[0,t]}\rangle, \; t \in [0, T]$.

---

These theoretical results suggest both a pricing and a hedging strategy for path-dependent derivatives based on expected signatures[32], summarized in Algorithm 2 and Algorithm 3. Both algorithms make use of expected signature estimation via Monte Carlo simulations, an approach that provides a classic setting for applying the martingale correction described in Section 2.2 (recall that by Remark 2.15 and the Lead-Lag discussion in Appendix F.1 we apply the correction only to signature terms with the process $\mathbb{X}$ appearing in the outer integral). Note that price processes under $\mathbb{P}$ (as considered in the hedging setting) are usually not martingales and hence it is not clear whether the variance reduction for the Monte Carlo estimator of the expected signature obtained via the martingale correction offsets the introduced bias. On the other hand, under $\mathbb{Q}$, the fundamental theorem of option pricing ensures the discounted asset price process $\mathbb{X}$ is a (local) martingale and hence the martingale correction is exact.

---

[31]For conciseness here we only discuss the quadratic hedging problem, in Lyons et al. (2021) the authors obtain results for general polynomials which allows them to also approximate the optimal hedge under exponential utility.

[32]As discussed in Pérez Arribas et al. (2018, Section 6.1) the expected signature under the measure $\mathbb{Q}$ can alternatively be estimated in a model-free way from the market prices of a large enough set of exotic derivatives, yielding the implied expected signature. Here we assume the measures $\mathbb{P}$ and $\mathbb{Q}$ have been appropriately calibrated to market data.

F.2.3. DISTRIBUTIONAL REGRESSION FOR STREAMS (LEMERCIER ET AL., 2021)

The machine learning model we consider in Section 3.2.3 is perhaps the most natural one when working with expected signatures. Introduced in Lemercier et al. (2021), the Signature of the pathwise Expected Signature (SES) model aims to learn a map from a collection of paths $\mathbb{X}^1, \ldots, \mathbb{X}^N \in \mathcal{X} \subseteq \mathrm{Lip}([0, T], \mathbb{R}^d)$, understood as an empirical measure on path space $\mu = \frac{1}{N} \sum_{n=1}^N \delta_{\mathbb{X}^n} \in \mathcal{P}(\mathcal{X})$, to a corresponding scalar value $f(\mu) \in \mathbb{R}$. The task of learning such function $f : \mathcal{P}(\mathcal{X}) \to \mathbb{R}$ from a finite number of (possibly noisy) observations $\{(\mu_i, y_i)\}_{i \in \mathcal{I}_{\text{train}}}$ is known as distributional regression. Under appropriate conditions, by combining the characterizing property of the expected signature and the universality of the signature, the authors show that linear functionals on the signature of the pathwise expected signature are universal for weakly continuous functions $f : \mathcal{P}(\mathcal{X}) \to \mathbb{R}$ (Lemercier et al., 2021, Theorem 3.2). The training and testing of the SES model is summarized in Algorithm 4 with the step at which can apply the martingale correction highlighted in green.

---

**Algorithm 4** Signature of pathwise Expected Signature (SES), training and testing

**hyperparameters** Signature truncation levels: $k_1, k_2 \in \mathbb{N}$. Linear regression regularizers.
**input** $\{(\{\mathbb{X}^n\}_{n \in \mathcal{N}_i}, y_i)\}_{i \in \mathcal{I}_{\text{train}}}, \{(\{\mathbb{X}^n\}_{n \in \mathcal{N}_i}, y_i)\}_{i \in \mathcal{I}_{\text{test}}}$ where $\mathbb{X}^n \in \mathbb{R}^{d \times M}$ and $y_i \in \mathbb{R}$.
 1: $d_1 \leftarrow d + \cdots + d^{k_1}$ and $d_2 \leftarrow d_1 + \cdots + d_1^{k_2}$.
 2: **for** $i \in \mathcal{I}_{\text{train}} \cup \mathcal{I}_{\text{test}}$ **do**
 3:   Pathwise expected signature of $\{\mathbb{X}^n\}_{n \in \mathcal{N}_i}$: $\Phi^i \in \mathbb{R}^{d_1 \times M}, \Phi^i_{I,m} \leftarrow \hat{\phi}^{\mathcal{N}_i, \hat{c}_1^*}_I(t_m), |I| \leq k_1, 1 \leq m \leq M$.
 4:   Signature of $\Phi^i$: $\mathbf{S}^i \leftarrow S^{\leq k_2}(\Phi^i)_{[0,T]} \in \mathbb{R}^{d_2}$.
 5: **end for**
 6: Fit linear regression: $\hat{\boldsymbol{\beta}} = (\hat{\beta}_0, \ldots, \hat{\beta}_{d_2}) \leftarrow \mathrm{LinearRegressionFit}(\{(\mathbf{S}^i, y_i)\}_{i \in \mathcal{I}_{\text{train}}})$.
 7: Predict using fitted linear regression: $\{\hat{y}_i\}_{i \in \mathcal{I}_{\text{test}}} \leftarrow \mathrm{LinearRegressionPredict}(\hat{\boldsymbol{\beta}}, \{\mathbf{S}^i\}_{i \in \mathcal{I}_{\text{test}}})$.
**output** Performance metric: $L(\{\hat{y}_i\}_{i \in \mathcal{I}_{\text{test}}}, \{y_i\}_{i \in \mathcal{I}_{\text{test}}})$.

---

In Section 3.2.3 we repeat two of the synthetic data experiments conducted in Lemercier et al. (2021), analyzing the performance of the SES model without and with martingale correction (MC). In the first experiment (Lemercier et al., 2021, Section 5.2), the task is to infer the temperature of an ideal gas from the paths of $N = 20$ particles moving in a box. The dynamics of the system are inevitably linked to the radius of the particles, with larger particle radii resulting in more frequent collisions. Two settings[33] are therefore considered, one with smaller particle radii $r_1 = 0.35 \times \sqrt[3]{V/N}$ and one with larger particle radii $r_2 = 0.65 \times \sqrt[3]{V/N}$. The second experiment (Lemercier et al., 2021, Section 5.3) concerns the estimation of the mean-reversion parameter in a rough volatility model. More precisely, the task is to infer the value of $a \in [10^{-6}, 1]$ from a sample $\{\sigma^n_\pi\}_{n=1}^N$ of (discretely observed) paths $\sigma^n = \{\sigma^n_t, \ t \in \pi\}$ over the partition $\pi = \{0, 0.01, \ldots, 2\}$ with continuous-time dynamics

$$\mathrm{d}Z_t = -a(Z_t - \mu)\mathrm{d}t + \nu \mathrm{d}B_t^H, \quad \sigma_t = \exp Z_t, \quad t \in [0, 2],$$

where $\{B_t^H, \ t \in [0, 2]\}$ is a fractional Brownian motion with Hurst parameter $H = 0.2$, $\mu = 0.5$, $\nu = 0.3$ and $Z_0 = 0.5$. The performance of the model is evaluated with increasingly large collections of paths as inputs, i.e. $N = 20, 50, 100$. As expected, the model becomes more accurate as the number of paths increases. We refer to Lemercier et al. (2021) for more details on the two experimental setups.

In both experiments, we keep the same training-evaluation pipeline as the one considered in the original paper, namely nested $k$-fold cross-validation with 5 outer folds for evaluation and 3 inner folds for hyperparameter selection (including the signature truncation $k_1$ and a Lasso regularization parameter). The code used to produce the results of Table 2 and Table 3 is available at `https://github.com/lorenzolucchese/distribution-regression-streams`.

F.2.4. SYSTEMATIC TRADING (FUTTER ET AL., 2023)

The last application we consider is also motivated by a financial application and can be understood as a natural extension of the quadratic hedging problem (52). In Futter et al. (2023) the authors consider the same setup as in Lyons et al. (2021) but allow the trading strategy $\theta \in \mathcal{A}$ to depend on the signature of the augmented process

$$\hat{\mathbb{Z}} = \{(t, \mathbf{X}_t, \mathbf{f}_t), \ t \in [0, T]\} \in C([0, T], \mathbb{R}^{1+d+q}),$$

---

[33]$V = 3\mathrm{cm}^3$ denotes the volume of the box in which the particles are moving.

where $\mathbb{X} = \{\mathbf{X}_t, \ t \in [0,T]\} \in C([0,T], \mathbb{R}^d)$ is a $d$-dimensional price process and $\mathbb{F} = \{\mathbf{f}_t, \ t \in [0,T]\} \in C([0,T], \mathbb{R}^q)$ is a set of $q$ observable but not tradable factors (or signals) influencing $\mathbb{X}$, and to trade in $d$ assets. Again motivated by a universality result of the signature the authors approximate any such trading strategy by a linear one, i.e.

$$\theta_i(S(\hat{\mathbb{Z}})_{[0,t]}) \approx \langle \ell_i, S(\hat{\mathbb{Z}})_{[0,t]} \rangle, \quad i \in \{1, \ldots, d\}, \ t \in [0,T],$$

for some set of linear functionals $\ell_1, \ldots, \ell_d \in T((\mathbb{R}^{1+d+q})*)$. The authors then show that an explicit solution to the path-dependent mean-variance problem

$$\ell_1^*, \ldots, \ell_d^* = \underset{\substack{\ell_1, \ldots, \ell_d \in T^k((\mathbb{R}^{1+d+q})^*) \\ \mathrm{Var}(\mathrm{PnL}_T) \leq \Delta}}{\operatorname{argmin}} \mathbb{E}^{\mathbb{P}}[\mathrm{PnL}_T], \quad \text{where} \quad \mathrm{PnL}_T = \sum_{i=1}^{d} \int_0^T \langle \ell_i, S(\hat{\mathbb{Z}})_{[0,t]} \rangle \mathrm{d}X_t^i,$$

for arbitrary truncation level $k \in \mathbb{N}$ can be read from the entries of

$$\frac{1}{2\lambda_\Delta} \Sigma_{\mathrm{sig}}^{-1} \mu_{\mathrm{sig}} \in \mathbb{R}^{d_{\mathrm{sig}}}, \quad d_{\mathrm{sig}} = d + \ldots + d(1 + d + q)^k,$$

where $\mu_{\mathrm{sig}} \in \mathbb{R}^{d_{\mathrm{sig}}}$ and $\Sigma_{\mathrm{sig}} \in \mathbb{R}^{d_{\mathrm{sig}} \times d_{\mathrm{sig}}}$ and $\lambda_\Delta \in \mathbb{R}_+$ only depend[34] on the expected signature $\mathbb{E}^{\mathbb{P}}[S(\hat{\mathbb{Z}}^{\mathrm{LL}})_{[0,T]}]$.

A standard way of applying the sig-trading strategy in practice is given in Algorithm 5. Note that in real financial markets price and signal paths cannot be resampled and hence the collection $\{(\mathbb{X}^n, \mathbb{F}^n)\}_{n=1}^N$ can only be obtained by chopping-and-shifting a single long observation $\{(\mathbf{X}_t, \mathbf{f}_t), \ t \in [0, NT]\}$. In this respect, the sig-trading algorithm provides a striking example of a setting where the sampling scheme cannot be considered i.i.d. and hence one needs to resort to the results of Theorem 2.10 to obtain theoretical guarantees for the expected signature estimator.

A silent but fundamental assumption[35] is that the market dynamics of the collection of *past* price-factor paths $\{(\mathbb{X}^n, \mathbb{F}^n)\}_{n=1}^N$ used to estimate the expected signature will be the same as those of the *future* price-factor process $(\mathbb{X}^*, \mathbb{F}^*)$ to which the trading strategy will be applied. Using the martingale correction for estimating (some of the entries of) the expected signature, induces a bias to "ignore" the drift component of the signature term. For example, the first level of the martingale-corrected expected signature is always zero. This might be a desirable feature to avoid over-fitting the trading strategy to spurious drifts in the data (for example in the price processes $\mathbb{X}$ (Buehler et al., 2022)) while capturing higher order effects. As with other applications discussed in this section, the usefulness of the martingale correction in Algorithm 5 can be empirically cross-validated.

---

**Algorithm 5** Signature Trading

**hyperparameters** Signature truncation level $k$, maximum variance $\Delta \in \mathbb{R}_+$.
**input** Collection of price-factor paths $\{(\mathbb{X}^n, \mathbb{F}^n)\}_{n=1}^N$ where $\mathbb{X}^n = \{\mathbf{X}_t^n, \ t \in [0,T]\} \in C([0,T], \mathbb{R}^d)$ and $\mathbb{F}^n = \{\mathbf{f}_t^n, \ t \in [0,T]\} \in C([0,T], \mathbb{R}^q)$ for $n \in \{1, \ldots, N\}$.
1: Set $\hat{\mathbb{Z}} \leftarrow \{(t, \mathbf{X}_t, \mathbf{f}_t), \ t \in [0,T]\}$.
2: Compute time-augmented lead-lag transforms $\hat{\mathbb{Z}}^{n,\mathrm{LL}}$ for $n \in \{1, \ldots, N\}$.
3: Estimate $\Phi \in T^k((\mathbb{R}^{1+d+q}))$ s.t. $\Phi_I \leftarrow \hat{\phi}_I^{N,\hat{c}_1}(T), |I| \leq k$ from $\{\hat{\mathbb{Z}}^{n,\mathrm{LL}}\}_{n=1}^N$.
4: Compute $\lambda_\Delta, \mu_{\mathrm{sig}}, \Sigma_{\mathrm{sig}}$ from corresponding entries of $\Phi$.
5: Extract $\hat{\ell}_i^*$ from corresponding entries of $(2\lambda_\Delta \Sigma_{\mathrm{sig}})^{-1} \mu_{\mathrm{sig}}$ for $i \in \{1, \ldots, d\}$.
**output** Trading strategy $t \mapsto \langle \hat{\ell}_i^*, S(\hat{\mathbb{Z}}^*)_{[0,t]} \rangle, \ i \in \{1, \ldots, d\}, \ t \in [0,T]$ for new $(\mathbb{X}^*, \mathbb{F}^*)$.

---

# G. Controlled Linear Regression

In this section, we introduce the notion of controlled linear regression. The main rationale is to exploit as much information as possible in the training phase to make the coefficient estimators as precise as possible. We start by considering the following linear model

$$y = \beta_1 x_1 + \ldots + \beta_p x_p + \epsilon = \mathbf{x}^\mathsf{T} \boldsymbol{\beta} + \epsilon,$$

---

[34]As in the mean-variance optimal portfolio, the variance scaling parameter $\lambda_\Delta \in \mathbb{R}_+$ also depends on the target variance $\Delta$.

[35]Clearly, this assumption is not specific to the sig-trading strategy but applies to any trading strategy that tries to learn patterns from the past to profit in the future.

and assume we observe the training data $\{(y_n, x_{n,1}, \ldots, x_{n,p}), \ n = 1, \ldots, N\}$, i.e.

$$y_n = \beta_1 x_{n,1} + \ldots + \beta_p x_{n,p} + \epsilon_n = \mathbf{x}_n^{\mathrm{T}}\boldsymbol{\beta} + \epsilon_n, \quad n = 1, \ldots, N,$$

where, given the design matrix $\mathbf{X} \in \mathbb{R}^{N \times p}$, the errors $\{\epsilon_n, \ n = 1, \ldots, N\}$ are

- mean zero, i.e. $\mathbb{E}[\epsilon_n | \mathbf{X}] = 0$ for $n = 1, \ldots, N$,

- homoskedastic, i.e. $\mathbb{E}[\epsilon_n^2 | \mathbf{X}] = \sigma^2 \in [0, \infty)$ for $n = 1, \ldots, N$, and

- uncorrelated, i.e. $\mathbb{E}[\epsilon_n \epsilon_m | \mathbf{X}] = 0$ for $n, m = 1, \ldots, N$ with $n \neq m$.

For any test observation $\mathbf{x}_* = (x_{*,1}, \ldots, x_{*,p}) \in \mathbb{R}^p$ we thus have the best possible prediction (in terms of MSE) for $y_*$ is

$$\mathbb{E}[y_* | \mathbf{x}_*] = \mathbf{x}_*^{\mathrm{T}}\boldsymbol{\beta} =: \hat{y}_*(\boldsymbol{\beta}),$$

and plugging in an estimator $\hat{\boldsymbol{\beta}}$ for $\boldsymbol{\beta}$ yields the predictor

$$\hat{y}_*(\hat{\boldsymbol{\beta}}) = \mathbf{x}_*^{\mathrm{T}}\hat{\boldsymbol{\beta}}.$$

Note that, under the assumptions introduced above, the mean squared error of such a predictor can be decomposed as

$$\mathbb{E}\left[(\hat{y}_*(\hat{\boldsymbol{\beta}}) - y_*)^2 \big| \mathbf{X}, \mathbf{x}_*\right] = \mathbb{E}\left[\epsilon_*^2 | \mathbf{X}, \mathbf{x}_*\right] + \mathbb{E}\left[(\hat{y}_*(\hat{\boldsymbol{\beta}}) - \hat{y}_*(\boldsymbol{\beta}))^2 \big| \mathbf{X}, \mathbf{x}_*\right]$$

$$= \sigma^2 + \mathbb{E}\left[(\hat{y}_*(\hat{\boldsymbol{\beta}}) - \hat{y}_*(\boldsymbol{\beta}))^2 \big| \mathbf{X}, \mathbf{x}_*\right].$$

Under the assumptions discussed above minimizing the mean squared error of the predictor relative to the target $y_*$ is thus equivalent to minimizing the mean squared error of the predictor relative to the infeasible best prediction $\hat{y}_*(\boldsymbol{\beta})$.

### G.1. Controlled Ordinary Least Squares (OLS) estimation

**Classic OLS estimation**    The usual OLS estimator for $\boldsymbol{\beta} = (\beta_1, \ldots, \beta_p) \in \mathbb{R}^p$ is given by[36]

$$\hat{\boldsymbol{\beta}}_{\mathbf{X}} = (\mathbf{X}^{\mathrm{T}}\mathbf{X})^{-1}\mathbf{X}^{\mathrm{T}}\mathbf{y},$$

which, by the Gauss-Markov theorem, is known to be the best linear unbiased estimator (BLUE): for any $\boldsymbol{\lambda} = (\lambda_1, \ldots, \lambda_p) \in \mathbb{R}^p$,

$$\mathbb{E}\left[(\boldsymbol{\lambda}^{\mathrm{T}}\hat{\boldsymbol{\beta}}_{\mathbf{X}} - \boldsymbol{\lambda}^{\mathrm{T}}\boldsymbol{\beta})^2 \big| \mathbf{X}\right] = \min_{\tilde{\boldsymbol{\beta}}_{\mathbf{X}} \in \mathrm{LUE}(\mathbf{X}, \mathbf{y})} \mathbb{E}\left[(\boldsymbol{\lambda}^{\mathrm{T}}\tilde{\boldsymbol{\beta}}_{\mathbf{X}} - \boldsymbol{\lambda}^{\mathrm{T}}\boldsymbol{\beta}) \big| \mathbf{X}\right],$$

where $\mathrm{LUE}(\mathbf{X}, \mathbf{y})$ is the set of all linear and unbiased estimator for $\boldsymbol{\beta}$, i.e. $\tilde{\boldsymbol{\beta}}_{\mathbf{X}} = C(\mathbf{X})\mathbf{y}$ for some $\mathbf{X}$-measurable matrix $C(\mathbf{X}) \in \mathbb{R}^{p \times N}$ and $\mathbb{E}[\tilde{\boldsymbol{\beta}}_{\mathbf{X}} | \mathbf{X}] = \boldsymbol{\beta}$. By applying the BLUE property, we can show that $\hat{y}_*(\hat{\boldsymbol{\beta}}_{\mathbf{X}})$ is the best[37] predictor across all predictors formed from linear and unbiased estimators, i.e.

$$\mathbb{E}\left[(\hat{y}_*(\hat{\boldsymbol{\beta}}_{\mathbf{X}}) - \hat{y}_*(\boldsymbol{\beta}))^2 \big| \mathbf{X}, \mathbf{x}_*\right] \leq \mathbb{E}\left[(\hat{y}_*(\tilde{\boldsymbol{\beta}}_{\mathbf{X}}) - \hat{y}_*(\boldsymbol{\beta}))^2 \big| \mathbf{X}, \mathbf{x}_*\right],$$

for all $\tilde{\boldsymbol{\beta}}_{\mathbf{X}} \in \mathrm{LUE}(\mathbf{X}, \mathbf{y})$. Note that here we applied the mean-zero uncorrelated errors assumption.

**Controlled OLS estimation**    Let us now assume we can additionally observe the "control" random variables $\{\mathbf{z}_n = (z_{n,1}, \ldots, z_{n,k}) \in \mathbb{R}^k, \ n = 1, \ldots, N\}$. We shall assume the controls are available for training, i.e. when estimating $\boldsymbol{\beta}$, but for predicting, i.e. when forecasting $y_*$ we will have access to $\mathbf{x}_*$ but not to $\mathbf{z}_*$. Given the original design matrix $\mathbf{X} \in \mathbb{R}^{N \times p}$, we now assume the errors and the controls are jointly

- mean zero, i.e. $\mathbb{E}[(\epsilon_n, \mathbf{z}_n) | \mathbf{X}] = \mathbf{0} \in \mathbb{R}^{k+1}$ for $n = 1, \ldots, N$,

---

[36]Here, and in all other estimators discussed in this section, the dependence on $\mathbf{y}$ is dropped from the notation.

[37]In terms of mean squared error (MSE). Recall that for unbiased estimators, the MSE is equal to the estimator's variance.

- homoskedastic, i.e. $\mathbb{E}[(\epsilon_n, \mathbf{z}_n)^{\otimes 2}|\mathbf{X}] = \Sigma \in \mathbb{R}^{(k+1)\times(k+1)}$ for $n = 1, \ldots, N$ for some $\Sigma$ symmetric positive definite,

- uncorrelated, i.e. $\mathbb{E}[(\epsilon_n, \mathbf{z}_n) \otimes (\epsilon_m, \mathbf{z}_m)|\mathbf{X}] = \mathbf{0} \in \mathbb{R}^{(k+1)\times(k+1)}$ for $n, m = 1, \ldots, N$ with $n \neq m$.

In what follows we partition

$$\Sigma = \begin{pmatrix} \sigma^2 & \Sigma_{y,\mathbf{z}} \\ \Sigma_{\mathbf{z},y} & \Sigma_{\mathbf{z}} \end{pmatrix},$$

*Remark* G.1. Throughout the whole section, unless stated otherwise, we consider the original design matrix $\mathbf{X}$ and the new observation $\mathbf{x}_* \in \mathbb{R}^p$ to be fixed with random controls $\mathbf{Z}$ and errors $\epsilon$.

As discussed above, fixing the design matrix $\mathbf{X} \in \mathbb{R}^{N \times p}$ and a test observation $\mathbf{x}_* = (x_{*,1}, \ldots, x_{*,p}) \in \mathbb{R}^p$, the predictor

$$\hat{y}_*(\hat{\boldsymbol{\beta}}_{\mathbf{X}}) = \mathbf{x}_*^{\mathsf{T}} \hat{\boldsymbol{\beta}}_{\mathbf{X}},$$

is unbiased for the statistic $\hat{y}_*(\boldsymbol{\beta}) = \mathbf{x}_*^{\mathsf{T}} \boldsymbol{\beta} \in \mathbb{R}$. We hence introduce the control variate predictor

$$\hat{y}_*(\hat{\boldsymbol{\beta}}_{\mathbf{X}}, \mathbf{Z}, \boldsymbol{\lambda}) = \hat{y}_*(\hat{\boldsymbol{\beta}}_{\mathbf{X}}) + \boldsymbol{\lambda}_1^{\mathsf{T}} \mathbf{Z} \boldsymbol{\lambda}_2,$$

where $\mathbf{Z} \in \mathbb{R}^{N \times k}$ is the control design matrix while $\boldsymbol{\lambda}_1 \in \mathbb{R}^N$ and $\boldsymbol{\lambda}_2 \in \mathbb{R}^k$ are measurable in $\mathbf{X}$ and $\mathbf{x}_*$. Under the assumptions discussed above the controlled predictor can be shown to be unbiased and attains a minimum[38] in variance when

$$\boldsymbol{\lambda}_1^* = \mathbf{X}(\mathbf{X}^{\mathsf{T}}\mathbf{X})^{-1}\mathbf{x}_* \quad \text{and} \quad \boldsymbol{\lambda}_2^* = -\Sigma_{\mathbf{z}}^{-1}\Sigma_{\mathbf{z},y}.$$

---

[38]Note that since $\mathbf{Z}$ and $\epsilon$ are assumed to be jointly spherical given $\mathbf{X}$ (and $\mathbf{x}_*$) the variance of the controlled predictor is given by

$$\begin{aligned}
&\mathrm{Var}(\hat{y}_*(\hat{\boldsymbol{\beta}}_{\mathbf{X}}, \mathbf{Z}, \boldsymbol{\lambda})|\mathbf{X}, \mathbf{x}_*) \\
&= \mathrm{Var}(\mathbf{x}_*^{\mathsf{T}}(\mathbf{X}^{\mathsf{T}}\mathbf{X})^{-1}\mathbf{X}^{\mathsf{T}}(\mathbf{X}\boldsymbol{\beta} + \epsilon) + \boldsymbol{\lambda}_1^{\mathsf{T}}\mathbf{Z}\boldsymbol{\lambda}_2|\mathbf{X}, \mathbf{x}_*) \\
&= \mathbf{x}_*^{\mathsf{T}}(\mathbf{X}^{\mathsf{T}}\mathbf{X})^{-1}\mathbf{X}^{\mathsf{T}}\mathbb{E}[\epsilon\epsilon^{\mathsf{T}}|\mathbf{X}, \mathbf{x}_*]\mathbf{X}(\mathbf{X}^{\mathsf{T}}\mathbf{X})^{-1}\mathbf{x}_* + 2\mathbf{x}_*^{\mathsf{T}}(\mathbf{X}^{\mathsf{T}}\mathbf{X})^{-1}\mathbf{X}^{\mathsf{T}}\mathbb{E}[\epsilon\boldsymbol{\lambda}_2^{\mathsf{T}}\mathbf{Z}^{\mathsf{T}}|\mathbf{X}, \mathbf{x}_*]\boldsymbol{\lambda}_1 + \boldsymbol{\lambda}_1^{\mathsf{T}}\mathbb{E}[\mathbf{Z}\boldsymbol{\lambda}_2\boldsymbol{\lambda}_2^{\mathsf{T}}\mathbf{Z}^{\mathsf{T}}|\mathbf{X}, \mathbf{x}_*]\boldsymbol{\lambda}_1 \\
&= \sigma^2\mathbf{x}_*^{\mathsf{T}}(\mathbf{X}^{\mathsf{T}}\mathbf{X})^{-1}\mathbf{x}_* + 2\boldsymbol{\lambda}_2^{\mathsf{T}}\Sigma_{\mathbf{z},y}\mathbf{x}_*^{\mathsf{T}}(\mathbf{X}^{\mathsf{T}}\mathbf{X})^{-1}\mathbf{X}^{\mathsf{T}}\boldsymbol{\lambda}_1 + \boldsymbol{\lambda}_2^{\mathsf{T}}\Sigma_{\mathbf{z}}\boldsymbol{\lambda}_2\boldsymbol{\lambda}_1^{\mathsf{T}}\boldsymbol{\lambda}_1.
\end{aligned}$$

Setting partial derivatives in $\boldsymbol{\lambda}_1$ and $\boldsymbol{\lambda}_2$ equal to zero

$$\partial_{\boldsymbol{\lambda}_1} \mathrm{Var}(\hat{y}_*(\hat{\boldsymbol{\beta}}_{\mathbf{X}}, \mathbf{Z}, \boldsymbol{\lambda})|\mathbf{X}, \mathbf{x}_*) = 2\boldsymbol{\lambda}_2^{\mathsf{T}}\Sigma_{\mathbf{z},y}\mathbf{x}_*^{\mathsf{T}}(\mathbf{X}^{\mathsf{T}}\mathbf{X})^{-1}\mathbf{X}^{\mathsf{T}} + 2\boldsymbol{\lambda}_2^{\mathsf{T}}\Sigma_{\mathbf{z}}\boldsymbol{\lambda}_2\boldsymbol{\lambda}_1 = 2\boldsymbol{\lambda}_2^{\mathsf{T}}(\Sigma_{\mathbf{z},y}\mathbf{x}_*^{\mathsf{T}}(\mathbf{X}^{\mathsf{T}}\mathbf{X})^{-1}\mathbf{X}^{\mathsf{T}} + \Sigma_{\mathbf{z}}\boldsymbol{\lambda}_2\boldsymbol{\lambda}_1) = 0,$$

$$\partial_{\boldsymbol{\lambda}_2} \mathrm{Var}(\hat{y}_*(\hat{\boldsymbol{\beta}}_{\mathbf{X}}, \mathbf{Z}, \boldsymbol{\lambda})|\mathbf{X}, \mathbf{x}_*) = 2\boldsymbol{\lambda}_1^{\mathsf{T}}\mathbf{X}(\mathbf{X}^{\mathsf{T}}\mathbf{X})^{-1}\mathbf{x}_*\Sigma_{\mathbf{z},y} + 2\boldsymbol{\lambda}_1^{\mathsf{T}}\boldsymbol{\lambda}_1\Sigma_{\mathbf{z}}\boldsymbol{\lambda}_2 = 2\boldsymbol{\lambda}_1^{\mathsf{T}}(\mathbf{X}(\mathbf{X}^{\mathsf{T}}\mathbf{X})^{-1}\mathbf{x}_*\Sigma_{\mathbf{z},y} + \boldsymbol{\lambda}_1\Sigma_{\mathbf{z}}\boldsymbol{\lambda}_2) = 0,$$

yields as non-trivial (i.e. such that $\boldsymbol{\lambda}_1, \boldsymbol{\lambda}_2 \neq 0$) stationary point $\boldsymbol{\lambda}_1^* = \mathbf{X}(\mathbf{X}^{\mathsf{T}}\mathbf{X})^{-1}\mathbf{x}_*$ and $\boldsymbol{\lambda}_2^* = -\Sigma_{\mathbf{z}}^{-1}\Sigma_{\mathbf{z},y}$. By taking second derivatives we can compute the Hessian at the stationary point to be

$$\partial_{\boldsymbol{\lambda}}\partial_{\boldsymbol{\lambda}^{\mathsf{T}}} \mathrm{Var}(\hat{y}_*(\hat{\boldsymbol{\beta}}_{\mathbf{X}}, \mathbf{Z}, \boldsymbol{\lambda})|\mathbf{X}, \mathbf{x}_*)\Big|_{\boldsymbol{\lambda} = \boldsymbol{\lambda}_*} = 2\begin{pmatrix} \Sigma_{y,\mathbf{z}}\Sigma_{\mathbf{z}}^{-1}\Sigma_{\mathbf{z},y}I_{N\times N} & -\mathbf{X}(\mathbf{X}^{\mathsf{T}}\mathbf{X})^{-1}\mathbf{x}_*\Sigma_{y,\mathbf{z}} \\ -\Sigma_{\mathbf{z},y}\mathbf{x}_*^{\mathsf{T}}(\mathbf{X}^{\mathsf{T}}\mathbf{X})^{-1}\mathbf{X}^{\mathsf{T}} & \mathbf{x}_*(\mathbf{X}^{\mathsf{T}}\mathbf{X})^{-1}\mathbf{x}_*\Sigma_{\mathbf{z}} \end{pmatrix},$$

which is positive (semi)definite since $\Sigma$ is positive definite and hence $\boldsymbol{\lambda}^*$ is a minimum. To show the assumption that $\Sigma$ is positive definite implies the Hessian is positive (semi)definite we use the following result from linear algebra theory: a symmetric block matrix

$$\mathbf{M} = \begin{pmatrix} \mathbf{A} & \mathbf{B} \\ \mathbf{B}^{\mathsf{T}} & \mathbf{D} \end{pmatrix},$$

where $\mathbf{A}$ is square is positive (semi)definite if and only $\mathbf{A}$ and $\mathbf{D} - \mathbf{B}^{\mathsf{T}}\mathbf{A}^{-1}\mathbf{B}$ are positive (semi)definite. Applying this result to the Hessian at $\boldsymbol{\lambda}_*$ note that $\mathbf{A} = \Sigma_{y,\mathbf{z}}\Sigma_{\mathbf{z}}^{-1}\Sigma_{\mathbf{z},y}I_{N\times N}$ is trivially positive (semi)definite since $\Sigma_{y,\mathbf{z}}\Sigma_{\mathbf{z}}^{-1}\Sigma_{\mathbf{z},y} \geq 0$ by positive definitness of $\Sigma_{\mathbf{z}}$, and

$$\mathbf{D} - \mathbf{B}^{\mathsf{T}}\mathbf{A}^{-1}\mathbf{B} = \mathbf{x}_*(\mathbf{X}^{\mathsf{T}}\mathbf{X})^{-1}\mathbf{x}_*\left(\Sigma_{\mathbf{z}} - \frac{\Sigma_{\mathbf{z},y}\Sigma_{y,\mathbf{z}}}{\Sigma_{y,\mathbf{z}}\Sigma_{\mathbf{z}}^{-1}\Sigma_{\mathbf{z},y}}\right),$$

is positive (semi)definite since $\mathbf{x}_*(\mathbf{X}^{\mathsf{T}}\mathbf{X})^{-1}\mathbf{x}_* \geq 0$ by positive definitness of $\mathbf{X}^{\mathsf{T}}\mathbf{X}$ and $\Sigma_{\mathbf{z}} - (\Sigma_{y,\mathbf{z}}\Sigma_{\mathbf{z}}^{-1}\Sigma_{\mathbf{z},y})^{-1}\Sigma_{\mathbf{z},y}\Sigma_{y,\mathbf{z}}$ is positive (semi)definite by applying the converse of the previous statement to the positive (semi)definite matrix

$$\mathbf{M}' = \begin{pmatrix} \Sigma_{y,\mathbf{z}}\Sigma_{\mathbf{z}}^{-1}\Sigma_{\mathbf{z},y} & \Sigma_{y,\mathbf{z}} \\ \Sigma_{\mathbf{z},y} & \Sigma_{\mathbf{z}} \end{pmatrix}.$$

For any $\mathbf{x}_* = (x_{*,1}, \ldots, x_{*,p}) \in \mathbb{R}^p$ we thus have

$$\hat{y}_*(\hat{\boldsymbol{\beta}}_{\mathbf{X}}, \mathbf{Z}, \boldsymbol{\lambda}^*) = \mathbf{x}_*^{\mathsf{T}}(\mathbf{X}^{\mathsf{T}}\mathbf{X})^{-1}\mathbf{X}^{\mathsf{T}}\mathbf{y} - \mathbf{x}_*^{\mathsf{T}}(\mathbf{X}^{\mathsf{T}}\mathbf{X})^{-1}\mathbf{X}^{\mathsf{T}}\mathbf{Z}\Sigma_{\mathbf{z}}^{-1}\Sigma_{\mathbf{z},y} = \mathbf{x}_*^{\mathsf{T}}\hat{\boldsymbol{\beta}}_{\mathbf{X},\mathbf{Z},\Sigma} = \hat{y}_*(\hat{\boldsymbol{\beta}}_{\mathbf{X},\mathbf{Z},\Sigma}),$$

yielding the infeasible (in the sense that it depends on the unknown quantity $\Sigma$) estimator

$$\hat{\boldsymbol{\beta}}_{\mathbf{X},\mathbf{Z},\Sigma} = (\mathbf{X}^{\mathsf{T}}\mathbf{X})^{-1}\mathbf{X}^{\mathsf{T}}(\mathbf{y} - \mathbf{Z}\Sigma_{\mathbf{z}}^{-1}\Sigma_{\mathbf{z},y}) = \hat{\boldsymbol{\beta}}_{\mathbf{X}} - (\mathbf{X}^{\mathsf{T}}\mathbf{X})^{-1}\mathbf{X}^{\mathsf{T}}\mathbf{Z}\Sigma_{\mathbf{z}}^{-1}\Sigma_{\mathbf{z},y}.$$

Note that, given $\mathbf{X}$, $\hat{\boldsymbol{\beta}}_{\mathbf{X},\mathbf{Z},\Sigma}$ is unbiased (by mean zero property of the controls). Moreover, for any test observation $\mathbf{x}_* = (x_{*,1}, \ldots, x_{*,p}) \in \mathbb{R}^p$,

$$\begin{aligned}
&\mathrm{Var}(\mathbf{x}_*^{\mathsf{T}}\hat{\boldsymbol{\beta}}_{\mathbf{X},\mathbf{Z},\Sigma}|\mathbf{X}, \mathbf{x}_*) \\
&= \mathrm{Var}(\mathbf{x}_*^{\mathsf{T}}\hat{\boldsymbol{\beta}}_{\mathbf{X}}|\mathbf{X}, \mathbf{x}_*) + \mathrm{Var}(\mathbf{x}_*^{\mathsf{T}}(\mathbf{X}^{\mathsf{T}}\mathbf{X})^{-1}\mathbf{X}^{\mathsf{T}}\mathbf{Z}\Sigma_{\mathbf{z}}^{-1}\Sigma_{\mathbf{z},y}|\mathbf{X}, \mathbf{x}_*) \\
&\qquad\qquad\qquad\qquad\qquad\quad - 2\,\mathrm{Cov}(\mathbf{x}_*^{\mathsf{T}}\hat{\boldsymbol{\beta}}_{\mathbf{X}}, \mathbf{x}_*^{\mathsf{T}}(\mathbf{X}^{\mathsf{T}}\mathbf{X})^{-1}\mathbf{X}^{\mathsf{T}}\mathbf{Z}\Sigma_{\mathbf{z}}^{-1}\Sigma_{\mathbf{z},y}|\mathbf{X}, \mathbf{x}_*) \\
&= \sigma^2\mathbf{x}_*^{\mathsf{T}}(\mathbf{X}^{\mathsf{T}}\mathbf{X})^{-1}\mathbf{x}_* + \Sigma_{y,\mathbf{z}}\Sigma_{\mathbf{z}}^{-1}\Sigma_{\mathbf{z},y}\mathbf{x}_*^{\mathsf{T}}(\mathbf{X}^{\mathsf{T}}\mathbf{X})^{-1}\mathbf{x}_* - 2\,\Sigma_{y,\mathbf{z}}\Sigma_{\mathbf{z}}^{-1}\Sigma_{\mathbf{z},y}\mathbf{x}_*^{\mathsf{T}}(\mathbf{X}^{\mathsf{T}}\mathbf{X})^{-1}\mathbf{x}_* \\
&= (\sigma^2 - \Sigma_{y,\mathbf{z}}\Sigma_{\mathbf{z}}^{-1}\Sigma_{\mathbf{z},y})\mathbf{x}_*^{\mathsf{T}}(\mathbf{X}^{\mathsf{T}}\mathbf{X})^{-1}\mathbf{x}_* \\
&\leq \sigma^2\mathbf{x}_*^{\mathsf{T}}(\mathbf{X}^{\mathsf{T}}\mathbf{X})^{-1}\mathbf{x}_* = \mathrm{Var}(\mathbf{x}_*^{\mathsf{T}}\hat{\boldsymbol{\beta}}_{\mathbf{X}}|\mathbf{X}, \mathbf{x}_*),
\end{aligned}$$

with equality iff $\Sigma_{\mathbf{z},y} = 0$. In other words, as long as the controls are correlated with the target, we obtain a better prediction by using $\hat{\boldsymbol{\beta}}_{\mathbf{X},\mathbf{Z},\Sigma}$ instead of the OLS estimator $\hat{\boldsymbol{\beta}}_{\mathbf{X}}$ and the quality of the prediction increases as the correlation between $y$ and $\mathbf{z}$ increases. The variance reduction factor is constant across test observations and is given by

$$\frac{\mathrm{Var}(\mathbf{x}_*^{\mathsf{T}}\hat{\boldsymbol{\beta}}_{\mathbf{X},\mathbf{Z},\Sigma}|\mathbf{X}, \mathbf{x}_*)}{\mathrm{Var}(\mathbf{x}_*^{\mathsf{T}}\hat{\boldsymbol{\beta}}_{\mathbf{X}}|\mathbf{X}, \mathbf{x}_*)} = \left(1 - \frac{\Sigma_{y,\mathbf{z}}\Sigma_{\mathbf{z}}^{-1}\Sigma_{\mathbf{z},y}}{\sigma^2}\right).$$

*Remark* G.2. Note that when $\mathbf{X} = \mathbb{1} \in \mathbb{R}^N$, $\mathbf{x}_* = 1$ and $\mathbf{Z} \in \mathbb{R}^N$, we estimate $\mu_* = \mathbb{E}[y]$ with the OLS estimator $\hat{\mu}_* = \hat{\beta} = \bar{\mathbf{y}}$ and the simplest control variate estimator

$$\hat{\mu}_*^c = \bar{\mathbf{y}} - \frac{\mathrm{Cov}(y,z)}{\mathrm{Var}(z)}\bar{\mathbf{z}},$$

since $\boldsymbol{\lambda}_1^* = \frac{1}{N}\mathbb{1} \in \mathbb{R}^N$ and $\lambda_2^* = -\frac{\mathrm{Cov}(y,z)}{\mathrm{Var}(z)}$, which has reduced variance by a factor of $(1 - \mathrm{Corr}(y,z))$.

In practice, the correlation matrix $\Sigma$ is usually unknown; hence, to make the estimator $\hat{\boldsymbol{\beta}}_{\mathbf{X},\mathbf{Z},\Sigma}$ feasible, we need to estimate it. Under the assumptions discussed above, the most natural candidate is given by the sample estimates

$$\hat{\Sigma}_{\mathbf{z},y} = \frac{1}{N}\mathbf{Z}^{\mathsf{T}}\mathbf{y} \quad \text{and} \quad \hat{\Sigma}_{\mathbf{z}} = \frac{1}{N}\mathbf{Z}^{\mathsf{T}}\mathbf{Z},$$

yielding the feasible estimator

$$\hat{\boldsymbol{\beta}}_{\mathbf{X},\mathbf{Z},\hat{\Sigma}} = (\mathbf{X}^{\mathsf{T}}\mathbf{X})^{-1}\mathbf{X}^{\mathsf{T}}(I - \mathbf{Z}(\mathbf{Z}^{\mathsf{T}}\mathbf{Z})^{-1}\mathbf{Z}^{\mathsf{T}})\mathbf{y}.$$

This can be equivalently understood as regressing $\mathbf{y}$ on the control $\mathbf{Z}$ (i.e. projecting $\mathbf{y}$ onto the space spanned by $\mathbf{Z}$) and then regressing the residual onto $\mathbf{X}$. The resulting estimator will likely be biased (given $\mathbf{X}$), with its exact finite sample properties depending on the distribution of $(\mathbf{X}, \mathbf{Z})|\epsilon$.

When $\epsilon$ depends linearly on $\mathbf{Z}$, i.e.

$$\epsilon = \mathbf{Z}\boldsymbol{\alpha} + \boldsymbol{\eta},$$

we can write

$$\mathbf{y} = \mathbf{X}\boldsymbol{\beta} + \mathbf{Z}\boldsymbol{\alpha} + \boldsymbol{\eta},$$

and hence we know that the joint OLS estimator obtained from the design matrix $(\mathbf{X}\ \mathbf{Z}) \in \mathbb{R}^{N \times (p+k)}$, i.e.

$$\begin{pmatrix} \hat{\boldsymbol{\beta}}_{\mathbf{X},\mathbf{z}} \\ \hat{\boldsymbol{\alpha}}_{\mathbf{X},\mathbf{z}} \end{pmatrix} = \left[\begin{pmatrix} \mathbf{X}^{\mathsf{T}} \\ \mathbf{Z}^{\mathsf{T}} \end{pmatrix}(\mathbf{X}\ \ \mathbf{Z})\right]^{-1}\begin{pmatrix} \mathbf{X}^{\mathsf{T}} \\ \mathbf{Z}^{\mathsf{T}} \end{pmatrix}\mathbf{y} = \begin{pmatrix} \mathbf{X}^{\mathsf{T}}\mathbf{X} & \mathbf{X}^{\mathsf{T}}\mathbf{Z} \\ \mathbf{Z}^{\mathsf{T}}\mathbf{X} & \mathbf{Z}^{\mathsf{T}}\mathbf{Z} \end{pmatrix}^{-1}\begin{pmatrix} \mathbf{X}^{\mathsf{T}}\mathbf{y} \\ \mathbf{Z}^{\mathsf{T}}\mathbf{y} \end{pmatrix}.$$

By using some simple algebraic manipulations for block matrices, we can extract the first $p$ entries of the joint OLS estimator, i.e. the estimator for $\boldsymbol{\beta}$, as

$$\hat{\boldsymbol{\beta}}_{\mathbf{X},\mathbf{Z}} = \hat{\boldsymbol{\beta}}_{\mathbf{X},\mathbf{Z},\hat{\Sigma}'} = \hat{\boldsymbol{\beta}} - (\mathbf{X}^{\mathsf{T}}\mathbf{X})^{-1}\mathbf{X}^{\mathsf{T}}\mathbf{Z}(\mathbf{Z}^{\mathsf{T}}\mathbf{Z} - \mathbf{Z}^{\mathsf{T}}\mathbf{X}(\mathbf{X}^{\mathsf{T}}\mathbf{X})^{-1}\mathbf{X}^{\mathsf{T}}\mathbf{Z})^{-1}(\mathbf{Z}^{\mathsf{T}}\mathbf{y} - \mathbf{Z}^{\mathsf{T}}\mathbf{X}(\mathbf{X}^{\mathsf{T}}\mathbf{X})^{-1}\mathbf{X}^{\mathsf{T}}\mathbf{y}),$$

i.e. $\hat{\boldsymbol{\beta}}_{\mathbf{X},\mathbf{Z},\Sigma}$ with $\Sigma$ estimated by

$$\hat{\Sigma}'_{\mathbf{z},y} = \frac{1}{N}\mathbf{Z}^{\mathsf{T}}(I - \mathbf{X}(\mathbf{X}^{\mathsf{T}}\mathbf{X})^{-1}\mathbf{X}^{\mathsf{T}})\mathbf{y} \quad \text{and} \quad \hat{\Sigma}'_{\mathbf{z}} = \frac{1}{N}\mathbf{Z}^{\mathsf{T}}(I - \mathbf{X}(\mathbf{X}^{\mathsf{T}}\mathbf{X})^{-1}\mathbf{X}^{\mathsf{T}})\mathbf{Z}.$$

Note these are the covariance and variance estimators obtained when projecting $\mathbf{Z}$ onto the orthogonal complement of the space spanned by $\mathbf{X}$. Under the assumptions introduced above, these are also unbiased for $\Sigma_{\mathbf{z},y}$ and $\Sigma_{\mathbf{z}}$. This provides a second feasible controlled estimator for $\boldsymbol{\beta}$.

If we fix both $\mathbf{X}$ and $\mathbf{Z}$ and assume $\boldsymbol{\eta}$ satisfies the Gauss-Markov conditions, then the joint OLS estimator is the BLUE for $(\boldsymbol{\beta}, \boldsymbol{\alpha})$. Extending the classic OLS estimator $\hat{\boldsymbol{\beta}}_{\mathbf{X}}$ to $(\hat{\boldsymbol{\beta}}_{\mathbf{X}}, \hat{\boldsymbol{\alpha}}_{\mathbf{X},\mathbf{Z}})$ we obtain another linear and unbiased estimator for $(\boldsymbol{\beta}, \boldsymbol{\alpha})$. It follows from the Gauss-Markov theorem that for any $\mathbf{x}_* \in \mathbb{R}^p$, setting $\boldsymbol{\lambda} = (\mathbf{x}_*, \mathbb{E}[\mathbf{z}_*|\mathbf{X}, \mathbf{Z}, \mathbf{x}_*] = \mathbf{0})$, one has

$$\mathbb{E}\left[(\hat{y}_*(\hat{\boldsymbol{\beta}}_{\mathbf{X},\mathbf{Z}}) - \hat{y}_*(\boldsymbol{\beta}))^2 \big| \mathbf{X}, \mathbf{Z}, \mathbf{x}_*\right] \leq \mathbb{E}\left[(\hat{y}_*(\hat{\boldsymbol{\beta}}_{\mathbf{X}}) - \hat{y}_*(\boldsymbol{\beta}))^2 \big| \mathbf{X}, \mathbf{Z}, \mathbf{x}_*\right],$$

and hence, by the tower property of conditional expectation,

$$\mathbb{E}\left[(\hat{y}_*(\hat{\boldsymbol{\beta}}_{\mathbf{X},\mathbf{Z}}) - \hat{y}_*(\boldsymbol{\beta}))^2 \big| \mathbf{X}, \mathbf{x}_*\right] \leq \mathbb{E}\left[(\hat{y}_*(\hat{\boldsymbol{\beta}}_{\mathbf{X}}) - \hat{y}_*(\boldsymbol{\beta}))^2 \big| \mathbf{X}, \mathbf{x}_*\right].$$

Using the forms of the variances of $\hat{\boldsymbol{\beta}}_{\mathbf{X}}$ and $\hat{\boldsymbol{\beta}}_{\mathbf{X},\mathbf{Z}}$ we can compute

$$\mathbb{E}\left[(\hat{y}_*(\hat{\boldsymbol{\beta}}_{\mathbf{X},\mathbf{Z}}) - \hat{y}_*(\boldsymbol{\beta}))^2 \big| \mathbf{X}, \mathbf{x}_*\right] = \sigma^2 \mathbf{x}_*^{\mathsf{T}} \mathbb{E}[(\mathbf{X}^{\mathsf{T}}(I - \mathbf{Z}(\mathbf{Z}^{\mathsf{T}}\mathbf{Z})^{-1}\mathbf{Z}^{\mathsf{T}})\mathbf{X})^{-1} \big| \mathbf{X}]\mathbf{x}_*,$$

$$\mathbb{E}\left[(\hat{y}_*(\hat{\boldsymbol{\beta}}_{\mathbf{X}}) - \hat{y}_*(\boldsymbol{\beta}))^2 \big| \mathbf{X}, \mathbf{x}_*\right] = \sigma^2 \mathbf{x}_*^{\mathsf{T}}(\mathbf{X}^{\mathsf{T}}\mathbf{X})^{-1}\mathbf{x}_*,$$

and thus quantify the MSE reduction factor as

$$\frac{\mathbb{E}\left[(\hat{y}_*(\hat{\boldsymbol{\beta}}_{\mathbf{X},\mathbf{Z}}) - \hat{y}_*(\boldsymbol{\beta}))^2 \big| \mathbf{X}, \mathbf{x}_*\right]}{\mathbb{E}\left[(\hat{y}_*(\hat{\boldsymbol{\beta}}_{\mathbf{X}}) - \hat{y}_*(\boldsymbol{\beta}))^2 \big| \mathbf{X}, \mathbf{x}_*\right]} = \frac{\mathbf{x}_*^{\mathsf{T}} \mathbb{E}[(\mathbf{X}^{\mathsf{T}}(I - \mathbf{Z}(\mathbf{Z}^{\mathsf{T}}\mathbf{Z})^{-1}\mathbf{Z}^{\mathsf{T}})\mathbf{X})^{-1} \big| \mathbf{X}]\mathbf{x}_*}{\mathbf{x}_*^{\mathsf{T}}(\mathbf{X}^{\mathsf{T}}\mathbf{X})^{-1}\mathbf{x}_*},$$

which we note does not depend on $\sigma^2$.

Given $\mathbf{X}$ and $\mathbf{Z}$, augmenting the first feasible controlled estimator $\hat{\boldsymbol{\beta}}_{\mathbf{X},\mathbf{Z},\hat{\Sigma}}$ to an estimator for $(\boldsymbol{\beta}, \boldsymbol{\alpha})$, yields a linear but biased (at least in the $\boldsymbol{\beta}$ components) estimator. Whether $\hat{\boldsymbol{\beta}}_{\mathbf{X},\mathbf{Z},\hat{\Sigma}}$ or $\hat{\boldsymbol{\beta}}_{\mathbf{X},\mathbf{Z}}$ yields a better predictor cannot thus be deduced from the Gauss-Markov theorem. As we will see in the numerical experiments discussed in the next section, which estimator performs better depends on the properties of the data generating process.

### G.2. Simulation study

In the previous section we introduced two feasible control estimators, $\hat{\boldsymbol{\beta}}_{\mathbf{X},\mathbf{Z},\hat{\Sigma}}$ and $\hat{\boldsymbol{\beta}}_{\mathbf{X},\mathbf{Z}}$, for the parameters $\boldsymbol{\beta} \in \mathbb{R}^p$. We showed that when

$$\boldsymbol{\epsilon} = \mathbf{Z}\boldsymbol{\alpha} + \boldsymbol{\eta},$$

and $\boldsymbol{\eta}|\mathbf{Z}, \mathbf{X}$ is mean-zero, uncorrelated and homoskedastic then $\hat{y}_*(\hat{\boldsymbol{\beta}}_{\mathbf{X},\mathbf{Z}})$ is always a better predictor than the one formed by the classic OLS estimator $\hat{y}_*(\hat{\boldsymbol{\beta}}_{\mathbf{X}})$. This leaves unanswered the question of how $\hat{\boldsymbol{\beta}}_{\mathbf{X},\mathbf{Z},\hat{\Sigma}}$ performs relative to $\hat{\boldsymbol{\beta}}_{\mathbf{X},\mathbf{Z}}$ (and $\hat{\boldsymbol{\beta}}_{\mathbf{X}}$). We address this question empirically in Table 4 with the following experimental setup.

We consider $N = 1,000$ i.i.d. samples from the model

$$y = \beta_0 + \beta_1 x_1 + \beta_2 x_2 + \epsilon,$$

with $\beta_0 = -1, \beta_1 = 6, \beta_2 = 8$ and

$$\epsilon = \sigma(\rho z + \sqrt{1 - \rho^2}\eta),$$

with $\eta \sim \mathcal{N}(0,1) \perp z \sim \mathcal{N}(0,1)$ independently of $\mathbf{X}$. We thus have $\text{Corr}(y, z|\mathbf{x}) = \text{Corr}(\epsilon, z) = \rho$ and $\epsilon \sim \mathcal{N}(0, \sigma^2) \perp \mathbf{X}$. We fix $N = 1,000$ samples $x_1 \sim \mathcal{N}(0,1) \perp x_2 \sim \mathcal{N}(1,1)$ to obtain $\mathbf{X} \in \mathbb{R}^{N \times 3}$ and evaluate the performance of the predictor on the new observation $x_1^* = 0, x_2^* = 1$. For each estimator, we report an estimate of the root mean square error

$$\text{RMSE}(\hat{y}_*(\hat{\boldsymbol{\beta}}), \hat{y}_*(\boldsymbol{\beta})) = \sqrt{\mathbb{E}\left[(\hat{y}_*(\hat{\boldsymbol{\beta}}) - \hat{y}_*(\boldsymbol{\beta}))^2 | \mathbf{X}, \mathbf{x}_*\right]},$$

obtained over $10,000$ Monte Carlo samples (i.e. keeping $\mathbf{X}$ and $\mathbf{x}^*$ fixed and resampling $\mathbf{Z} \in \mathbb{R}^N$ and $\boldsymbol{\eta} \in \mathbb{R}^N$). We highlight with a single asterisk (*) RMSEs that are lower than the uncontrolled OLS estimator's RMSE with high statistical significance (t-test p-value across the Monte Carlo samples less than 0.001). When one of the two proposed estimators outperforms the other with high statistical significance we highlight the corresponding RMSE with double asterisks (**).

| $\sigma$ | $\rho$ | $\hat{\boldsymbol{\beta}}_{\mathbf{X}}$ RMSE | $\hat{\boldsymbol{\beta}}_{\mathbf{X},\mathbf{Z},\hat{\Sigma}}$ RMSE | (% of $\hat{\boldsymbol{\beta}}_{\mathbf{X}}$) | $\hat{\boldsymbol{\beta}}_{\mathbf{X},\mathbf{Z}}$ RMSE | (% of $\hat{\boldsymbol{\beta}}_{\mathbf{X}}$) | $\hat{\boldsymbol{\beta}}_{\mathbf{X},\mathbf{Z},\Sigma}$ RMSE | (% of $\hat{\boldsymbol{\beta}}_{\mathbf{X}}$) |
|---|---|---|---|---|---|---|---|---|
| | 0.00 | 0.1572 | 0.1579 | 100.46% | 0.1573 | 100.03% | 0.1572 | 100.00% |
| | 0.25 | 0.1588 | 0.1550* | 97.57% | 0.1538** | 96.86% | 0.1489 | 93.75% |
| 5 | 0.50 | 0.1581 | 0.1368* | 86.48% | 0.1358** | 85.86% | 0.1186 | 75.00% |
| | 0.75 | 0.1592 | 0.1052* | 66.08% | 0.1043** | 65.52% | 0.0697 | 43.75% |
| | 1.00 | 0.1606 | 0.0164* | 10.21% | 0.0000** | 0.01% | 0.0000 | 0.00% |
| | 0.00 | 0.3144 | 0.3146 | 100.06% | 0.3145 | 100.03% | 0.3144 | 100.00% |
| | 0.25 | 0.3177 | 0.3082* | 97.03% | 0.3077** | 96.86% | 0.2978 | 93.75% |
| 10 | 0.50 | 0.3163 | 0.2719* | 85.97% | 0.2715* | 85.86% | 0.2372 | 75.00% |
| | 0.75 | 0.3185 | 0.2088* | 65.57% | 0.2086* | 65.52% | 0.1393 | 43.75% |
| | 1.00 | 0.3211 | 0.0164* | 5.11% | 0.0000** | 0.00% | 0.0000 | 0.00% |
| | 0.00 | 0.6289 | 0.6286 | 99.96% | 0.6291 | 100.03% | 0.6289 | 100.00% |
| | 0.25 | 0.6353 | 0.6154* | 96.86% | 0.6154* | 96.86% | 0.5956 | 93.75% |
| 20 | 0.50 | 0.6325 | 0.5429* | 85.83% | 0.5431* | 85.86% | 0.4744 | 75.00% |
| | 0.75 | 0.6369 | 0.4169* | 65.46% | 0.4173* | 65.52% | 0.2786 | 43.75% |
| | 1.00 | 0.6422 | 0.0164* | 2.55% | 0.0000** | 0.00% | 0.0000 | 0.00% |
| | 0.00 | 1.2578 | 1.2569 | 99.93% | 1.2582 | 100.03% | 1.2578 | 100.00% |
| | 0.25 | 1.2707 | 1.2300** | 96.80% | 1.2307* | 96.86% | 1.1913 | 93.75% |
| 40 | 0.50 | 1.2651 | 1.0853** | 85.79% | 1.0862* | 85.86% | 0.9488 | 75.00% |
| | 0.75 | 1.2738 | 0.8336** | 65.44% | 0.8346* | 65.52% | 0.5573 | 43.75% |
| | 1.00 | 1.2845 | 0.0164* | 1.28% | 0.0000** | 0.00% | 0.0000 | 0.00% |

*Table 4.* RMSEs of the classic OLS estimator $\hat{\boldsymbol{\beta}}_{\mathbf{X}}$, the two feasible control estimators $\hat{\boldsymbol{\beta}}_{\mathbf{X},\mathbf{Z},\hat{\Sigma}}$ and $\hat{\boldsymbol{\beta}}_{\mathbf{X},\mathbf{Z}}$, and the infeasible optimal control estimator $\hat{\boldsymbol{\beta}}_{\mathbf{X},\mathbf{Z},\Sigma}$. A single asterisk (*) indicates feasible RMSEs that are lower than the uncontrolled classic OLS estimator's RMSE with high statistical significance (t-test p-value across the 10,000 Monte Carlo samples less than 0.001). A double asterisk (**) indicates the feasible control estimator's RMSE is lower than the other feasible control estimator's RMSE with high statistical significance.

As expected, as the correlation between the control and the target $\rho$ increases, the performance gain obtained by using either of the feasible control estimators increases. The joint-OLS estimator outperforms the standard OLS estimator by the same amount across different signal-to-noise ratios while $\hat{\boldsymbol{\beta}}_{\mathbf{X},\mathbf{Z},\hat{\Sigma}}$'s relative performance changes. Comparing $\hat{\boldsymbol{\beta}}_{\mathbf{X},\mathbf{Z},\hat{\Sigma}}$ and $\hat{\boldsymbol{\beta}}_{\mathbf{X},\mathbf{Z}}$ we note the results suggest that for high signal-to-noise ratios[39] the joint-OLS estimator $\hat{\boldsymbol{\beta}}_{\mathbf{X},\mathbf{Z}}$ slightly outperforms $\hat{\boldsymbol{\beta}}_{\mathbf{X},\mathbf{Z},\hat{\Sigma}}$ while for lower signal-to-noise ratios the latter estimator performs marginally better. Consider the two edge cases:

---

[39]Here we define the signal-to-noise ratio as $\text{std}(\mathbf{x}^\mathsf{T}\boldsymbol{\beta})/\sigma$ where $\text{std}(\mathbf{x}^\mathsf{T}\boldsymbol{\beta})$ measures the variablity of the explainable part of the model (signal) and $\sigma$ measures the variability in the error $\epsilon$ (noise). In Table 4 the signal component of the model is kept fixed and hence higher $\sigma$ denotes lower signal-to-noise ratio.

- when there is no error, i.e. $\sigma = 0$, $\hat{\beta}_{\mathbf{X},\mathbf{Z},\hat{\Sigma}}$ performs worse than $\hat{\beta}_{\mathbf{X}}$ ($= \hat{\beta}_{\mathbf{X},\mathbf{Z}}$) as it is adding uninformative variability to the estimator;

- when there is no signal, i.e. $\mathbf{X} = \mathbb{1} \in \mathbb{R}^N$, $\hat{\beta}_{\mathbf{X},\mathbf{Z},\hat{\Sigma}}$ and $\hat{\beta}_{\mathbf{X},\mathbf{Z}}$ both reduce to the classic control variates estimators, cf. Remark G.2, but the former is more precise as it estimates $\lambda_2^* = -\Sigma_{\mathbf{z}}^{-1}\Sigma_{\mathbf{z},y}$ using the knowledge that the control is mean zero.

Next, we investigate the empirical performance of $\hat{\beta}_{\mathbf{X},\mathbf{Z},\hat{\Sigma}}$ and $\hat{\beta}_{\mathbf{X},\mathbf{Z}}$ when the dependency between $\mathbf{Z}$ and $\epsilon$ is non-linear but $(\mathbf{Z}, \epsilon)|\mathbf{X}$ are still jointly mean-zero, uncorrelated (across samples) and homoskedastic. Keeping the same design matrix $\mathbf{X}$ as in Table 4 and the same parameters $\boldsymbol{\beta} = (-1, 6, 8)$ we now let

$$\epsilon = \sigma \kappa f(z) + \sigma \sqrt{1 - \kappa^2} \eta,$$

with $\eta \sim \mathcal{N}(0,1) \perp z \sim \mathcal{N}(0,1)$ independently of $\mathbf{X}$. By choosing $f(z)$ such that $\mathbb{E}[f(z)] = 0$ and $\mathbb{E}[f(z)^2] = 1$ we have $\mathbb{E}[\epsilon] = 0$ and $\mathbb{E}[\epsilon^2] = \sigma^2$. Moreover, choosing $\kappa = \text{Cov}(z, f(z))^{-1}\rho$ ensures that $\text{Corr}(y, z|\mathbf{x}) = \text{Corr}(\epsilon, z) = \rho$. We investigate the following three dependence functions:

(sq) $f(z) = \dfrac{z^2 + z - 1}{\sqrt{3}}$;

(cube) $f(z) = \dfrac{z^3}{\sqrt{15}}$;

(exp) $f(z) = \dfrac{e^z - \sqrt{e}}{\sqrt{e^2 - e}}$.

The code used to produce the results of Table 4 and Table 5 can be found at https://github.com/lorenzolucchese/controlled-linear-regression.

We also experimented with multiplicative noise ($\epsilon \propto f(z)\,\eta$), heavier tailed errors ($\eta \sim t_3$) and different dataset sizes ($N \in \{100, 1000, 10000\}$). The results are similar to the ones reported in Table 4 and Table 5: the two control estimators outperform the classic OLS estimator as long as $\rho > 0$ while the differences in performance between the two control estimators are statistically significant but small.

| $f$ | $\rho$ | $\hat{\beta}_{\mathbf{X}}$ RMSE | $\hat{\beta}_{\mathbf{X},\mathbf{Z},\hat{\Sigma}}$ RMSE | (% of $\hat{\beta}_{\mathbf{X}}$) | $\hat{\beta}_{\mathbf{X},\mathbf{Z}}$ RMSE | (% of $\hat{\beta}_{\mathbf{X}}$) | $\hat{\beta}_{\mathbf{X},\mathbf{Z},\Sigma}$ RMSE | (% of $\hat{\beta}_{\mathbf{X}}$) |
|---|---|---|---|---|---|---|---|---|
| (sq) | 0.00 | 0.3144 | 0.3146 | 100.06% | 0.3145 | 100.03% | 0.3144 | 100.00% |
| | 0.25 | 0.3174 | 0.3084* | 97.14% | 0.3075** | 96.88% | 0.2976 | 93.75% |
| | 0.50 | 0.3202 | 0.2772* | 86.57% | 0.2761** | 86.25% | 0.2401 | 75.00% |
| | 0.75 | | | | | | | |
| | 1.00 | | | | | | | |
| (cube) | 0.00 | 0.3144 | 0.3146 | 100.06% | 0.3145 | 100.03% | 0.3144 | 100.00% |
| | 0.25 | 0.3184 | 0.3088* | 97.00% | 0.3083* | 96.83% | 0.2985 | 93.75% |
| | 0.50 | 0.3184 | 0.2736* | 85.95% | 0.2734* | 85.86% | 0.2388 | 75.00% |
| | 0.75 | 0.3222 | 0.2125* | 65.96% | 0.2123* | 65.90% | 0.1409 | 43.75% |
| | 1.00 | | | | | | | |
| (exp) | 0.00 | 0.3144 | 0.3146 | 100.06% | 0.3145 | 100.03% | 0.3144 | 100.00% |
| | 0.25 | 0.3178 | 0.3085* | 97.07% | 0.3078** | 96.85% | 0.2979 | 93.75% |
| | 0.50 | 0.3182 | 0.2741* | 86.13% | 0.2734** | 85.92% | 0.2387 | 75.00% |
| | 0.75 | 0.3191 | 0.2091* | 65.53% | 0.2081** | 65.20% | 0.1396 | 43.75% |
| | 1.00 | | | | | | | |

*Table 5.* RMSEs of the classic OLS estimator $\hat{\beta}_{\mathbf{X}}$, the two feasible control estimators $\hat{\beta}_{\mathbf{X},\mathbf{Z},\hat{\Sigma}}$ and $\hat{\beta}_{\mathbf{X},\mathbf{Z}}$, and the infeasible optimal control estimator $\hat{\beta}_{\mathbf{X},\mathbf{Z},\Sigma}$. A single asterisk (*) indicates feasible RMSEs that are lower than the uncontrolled classic OLS estimator's RMSE with high statistical significance (t-test p-value across the 10,000 Monte Carlo samples less than 0.001). A double asterisk (**) indicates the feasible control estimator's RMSE is lower than the other feasible control estimator's RMSE with high statistical significance. Empty values indicate setups that are not achievable for the given correlation $\rho$ and dependence function $f$.

