# OpenReview forum: "Learning with Expected Signatures: Theory and Applications"
_ICML.cc/2025/Conference — ICML 2025 oral_

### Official Review · Reviewer_d8FL · 2025-03-10

**Overall Recommendation:** 5

**Summary:**

This paper establishes a rigorous framework for the expected signature of stochastic processes, proving consistency and asymptotic normality under a double asymptotic regime—where the discretization mesh ($\pi$) tends to zero (in-fill asymptotics) and the number of observations $N$ increases (long-span asymptotics). It also introduces a martingale-based variance reduction technique that further improves finite-sample performance, with empirical validations.

## Update After Rebuttal
I raised my score as a result of a discussion with the authors (see below).

**Claims And Evidence:**

This paper makes several notable contributions that extend and deepen their theoretical utility in machine learning.

(i) Bridging Discrete and Continuous Frameworks:
The paper rigorously establishes conditions under which the empirical expected signature estimator—computed from discretely observed, piecewise linear approximations of a continuous-time process—converges (in a double-asymptotic sense) to the true expected signature of the underlying latent continuous-time stochastic process. The authors establish consistency and asymptotic normality (Thm. 2.8 / 2.10) under regularity, mixing, and moment conditions. Detailed proofs in the Appendix use techniques from Rough Path theory and stochastic analysis.

(ii) Variance Reduction via Martingale Correction:
Recognizing that the classical expected signature estimator can suffer from high variance in finite samples, the paper introduces a modified estimator that employs a control variate derived from the properties of the signature in the martingale setting. By leveraging the Stratonovich integral’s properties, this martingale correction leads to significantly lower mean squared error. The authors back up this theoretical finding with empirical experiments demonstrating improved performance in tasks such as time series classification and option pricing.

**Essential References Not Discussed:**

Essential references are adequately discussed.

**Experimental Designs Or Analyses:**

See the above “Methods and Evaluation Criteria” section.

**Methods And Evaluation Criteria:**

While the authors evaluate their approach on a classification task, there are practical concerns regarding scalability in modern machine learning settings due to the curse of dimensionality. Specifically, when a $d$-dimensional time series is mapped to its signature truncated at level p, the resulting feature vector has dimension $\sum_{k=0}^{p} d^k$. This growth can become prohibitive for high-dimensional and multimodal data—for example, video streams where d might be on the order of $256 \times 256$ or more.

Moreover, for many classification tasks, task-specific representations (such as those learned by deep neural networks that selectively discard irrelevant information) may achieve superior performance compared to employing a full sufficient statistic like the expected signature.

Despite these practical limitations, it is important to note that the primary focus of the paper is on theoretical contributions. Given this theoretical orientation, the experimental evaluations serve primarily to validate the underlying mathematical results rather than to claim state-of-the-art performance in classification. Consequently, while the potential practical challenges of scaling signature methods to very high-dimensional data are acknowledged, they should not be considered grounds for rejection of the paper.

**Other Comments Or Suggestions:**

The paper was an enjoyable read. Please note that my comments represent my initial impressions and may include misunderstandings. I welcome further discussion on these points and am open to revising my score once my questions and concerns are adequately addressed.

**Other Strengths And Weaknesses:**

The paper employs several technical terms—such as BV, the shuffle property, p-variation, and words $I$—without providing sufficient introductory guidance or intuitive explanations. While these concepts may be well-known to experts in stochastic processes and Rough Path theory, the ICML audience is broad and interdisciplinary. To enhance clarity and accessibility, the authors are encouraged to include a concise glossary or an appendix that offers reader-friendly definitions and intuitive explanations of these key terms.

**Questions For Authors:**

Do you plan to make your implementation code publicly available in the near future? Given the complexity of the theory, having accessible code would greatly help in ensuring reproducibility.

**Relation To Broader Scientific Literature:**

This paper bridges the gap between the theoretical framework of expected signatures—defined via nested iterated integrals in continuous time—and their practical estimation from discrete time series data. By rigorously establishing unbiased estimation and asymptotic properties for the expected signature, the authors connect foundational ideas from Rough Path theory with estimation techniques common in practice, thus broadening the future research of signature methods in machine learning.

**Theoretical Claims:**

I have reviewed the high-level theoretical claims and proof sketch but there remains a possibility that oversights exist.

---

> ### Author Rebuttal · Authors · 2025-03-30
>
> We would like to thank the reviewer for the comments. We believe the reviewer well understood the main contributions of the paper, which they highlighted in "Claims and Evidence".
>
> **Methods And Evaluation Criteria**
>
> In practical applications the (expected) signature transform may face the curse of dimensionality. Nevertheless estimation of the expected signature is still a relevant topic as, at least for low dimensional time-series, it provides a theoretically motivated domain-agnostic baseline model. Moreover, ongoing work is being carried out to "a priori" select a transformation of the $\sum\_{k=0}^p d^k$ truncated (expected) signature feature vector retaining its essential characteristics. For example in [2] the authors introduce a linear state-space transformation such that redundant information can be easily removed from the signature vector by (a further) truncation.
>
> Usually, a domain-specific architecture – for example, in the case of image/video data, a convolutional layer targeting the known topological structure of the input – will likely improve empirical model performance (cf. Remark E.1). In such cases, the (expected) signature can be used as a layer in a neural network, see [3]. When working with video streams, one might consider a sequential architecture with an initial embedding layer mapping the data to a lower dimensional space before computing the (expected) signature. The empirical applications discussed in this paper provide a set of simple benchmark tasks on which to evaluate the performance of the martingale correction (such simple models do not and should not be understood as trying to achieve SOTA results). The scope of this work was not to develop new expected signature-based architectures but rather, as well understood by the reviewer, to fill a gap in the theory and, under suitable conditions, provide an improved empirical estimator for the theoretical expected signature.
>
> [2] Bayer and Redmann, Dimension reduction for path signatures, preprint, https://arxiv.org/pdf/2412.14723
>
> [3] Bonnier et al., Deep Signature Transforms, NeurIPS 2019, https://arxiv.org/pdf/1905.08494
>
> **Other Strengths And Weaknesses**
>
> We shall add intuitive definitions of technical terms in the appendix, the following is an extract from said glossary.
>
> *Informal glossary*
>
> The $p$-variation of a path is a measure of its regularity. For the purpose of our discussion it suffices to note that paths that have finite $p$-variation for low $p$ are more regular. A bounded variation (BV) path is a path with finite 1-variation (also known as total variation), this regularity ensures there exists a well-defined notion of integral against this path (e.g. a piecewise linear paths or continuously differentiable path) and hence we can easily define its signature as in Equation (2). Many interesting stochastic processes (e.g. those driven by Brownian motion) have infinite 1-variation (i.e. are not BV) but have finite $p$-variation for all $p>2$ and hence defining their signature requires rough path theory.
>
> The shuffle property of the signature is an algebraic property stating that the product of two signature terms is a linear combination of higher-order signature terms. More precisely, the product of the signature terms corresponding to words $I$ and $J$ is the sum of all signature terms indexed by words $K$ of length $|I|+|J|$ obtained by interleaving $I$ and $J$. In the context of the discussion on page 1 this means that all moments of the signature can be written as linear combinations of higher order expected signature terms.
>
> A word $I=(i\_1, \ldots, i\_n)$ with $i\_1, \ldots, i\_n\in\\{1, \ldots, d\\}$ is a multi-index used to denote an entry of the signature, i.e. a real-valued number. The length of the word, i.e. $|I| = n$, denotes to which level ($n$-dimensional tensor) of the signature the entry belongs. For example $S^I(\mathbb{X})\_{[0,T]}$ where $I=(1,2)$ denotes the $(1, 2)$-entry of the second level of the signature (a matrix).
>
> [...]
>
> **Questions For Authors**
>
> Yes, we will contribute public code in the following two ways:
> - An independent code repository containing methods to estimate expected signatures using both the simple empirical estimator and the martingale corrected estimator (compatible with `numpy` and `torch` arrays, using `iisignature` and `signatory` for signature computations). This directory will also contain the scripts needed to reproduce the option-pricing experiment of Section 3.2.2.
> - Forks of the original repositories for the other two experiments adding the option to use the martingale corrected estimator, which can be used to reproduce the results of Sections 3.2.1 and 3.2.3.
>
> We will refer to the public repositories in the final version of the paper. This information was redacted from the first submission in adherence with the conference's double blind review policy.

---

> > ### Comment · Reviewer_d8FL · 2025-04-03
> >
> > I greatly appreciate your thorough and insightful comments. Your feedback has clarified all of my concerns, and, in light of discussions with other reviewers, I have raised my score accordingly. I look forward to the final version of the paper and the accompanying code repository.

---

### Official Review · Reviewer_v9sL · 2025-03-14

**Overall Recommendation:** 5

**Summary:**

The authors explore an interesting and young topic with in ML, that is, signature-based methods for ML. Signature methods have been quite useful as a sort of preprocessing stage in ML pipelines to synthesize long time-series data among other applications.
The authors theoretically explore the expected signature and it's relation to the empirical expected signature providing a useful bridge between theory and practice.
In this work they explore how to improve the consistency of such empirical estimators.
Lastly, the explore a few relevant applications.

## Update after rebuttal
I think this is a significant work that can establish future downstream applications making use of the promising tool of signatures from rough paths theory. I increased my score to 5 - strong accept.

**Claims And Evidence:**

To the best of my knowledge all the claims are supported, however, I am not a expert in rough path theory and only performed a quick review of the arguments and proofs.

**Essential References Not Discussed:**

N/A

**Experimental Designs Or Analyses:**

From what I read for all three time series experiments they seem sound.

**Methods And Evaluation Criteria:**

The authors explore several interesting and relevant experiments. The evaluation seems to be fair and illustrates the strength of the theory as they are able to reduce the variance of their estimator.

**Other Comments Or Suggestions:**

* The footnotes in page 6 are quite long particular footnote 6. Perhaps 6 could be moved to the appendix (or part of it).
* I think highlighting key points and contributions would make the paper easier to read.
* Likewise, a further discussion of motivation for design decisions in constructing the estimator would improve readability.

**Other Strengths And Weaknesses:**

## Strengths
* Well written paper
* Addresses an important area of research
* Theoretical results could be quite important for future research in the coming years

## Weaknesses
* No conclusion at the end to reiterate contributions
* Paper is quite dense (to be expected) and the organization will good is not exemplary
* I think [1] is a good example of illustrating complicated topics to the layman.
* Numerous equations in the main paper are unnumbered which is bad practice for future citations.
* The motivations for several key design decisions are either missing or not communicated clearly. Especially, for work of this type it is highly important that you make your key ideas as easy to understand as possible.
* While the appendices are quite detailed and show important derivations, this results in the paper being extremely long (I understand, however, that this is the current trend in math-heavy ML research).

[1] Morrill et al., *Neural Rough Differential Equations for Long Time Series*, ICML 2021 https://arxiv.org/pdf/2009.08295

**Questions For Authors:**

1. Does $S(\mathbb X)\_{[0,T]}$ denote $S^k(\mathbb X)_{[0,T]}$ as $k \to \infty$? This is my understanding but I don't see it stated clearly in the main text.
2. Why is the equation on line 285 unnumbered? It seems quite important to the paper.
3. What is the motivation for constructing the equation mentioned above, why do you choose this corrector? I have some thoughts, but this motivation should be included in the writing.

N.B., I generally like this paper and enjoyed reading it; however, the lack of clarity and explanation of motivations prohibits me from giving this paper the highest rating.

**Relation To Broader Scientific Literature:**

This work seems quite relevant to research which uses long time-series data.

**Theoretical Claims:**

I performed a preliminary read through of the argument, however, I did not have the time nor expertise to extensively review the 40+ pages of appendices in detail.

---

> ### Author Rebuttal · Authors · 2025-03-30
>
> We would like to thank the reviewer for the comments. We believe the reviewer well understood the main contributions of the paper, which they highlighted in "Summary".
>
> **Other Strengths And Weaknesses**
> - Conclusions were not added due to space constraints, in case of acceptance we shall include the following short conclusion reiterating the main contributions of the paper. "In this paper we established new estimation results for the expected signature, a model-free embedding for collections of data streams. Our consistency and asymptotic normality results bridge the gap between the theoretically "optimal" continuous-time expected signature and the empirical discrete-time estimator that can be computed from data. Moreover, we introduced a simple modification of such estimator with significantly better finite sample properties under the assumption of martingale observations. Our empirical results suggest the modified estimator might improve the performance of models employing expected signature computations even when the underlying data generating process is not necessarily a martingale."
> - Inspired by the provided reference [1], in case of acceptance, we shall include a simple diagram to visually illustrate the main contributions of the paper in the introduction (unfortunately we cannot upload the image in this rebuttal).
> - We generally followed the convention that all the equations that are cross-referenced in the paper are numbered and the others are not.
> - See answer to question 3. below.
> - The results in the appendix provide the detailed proofs of the results discussed in the main paper. These details though are not strictly necessary to understand the main contributions of this paper but we still believe a good mathematical proof should leave as little details unexplained as possible. This is why we would like to keep the level of detail presented in the appendix as is.
>
> **Other Comments Or Suggestions**
> - We will move the discussion contained in this footnote to the main text as it provides the main motivation for using the martingale correction term, see also the answer to question 3. below.
> - As discussed above, we will highlight key points and contributions further by adding an intuitive diagram and a short conclusion.
> - This is addressed in the answer to question 3. below.
>
> **Questions For Authors**
> - In our notation $S^k(\mathbb{X})\_{[0,T]} \in (\mathbb{R}^d)^{\otimes k}$ is the $k$-th level of the signature, i.e. $$S(\mathbb{X})\_{[0,T]} = (S^0(\mathbb{X})\_{[0,T]}, S^1(\mathbb{X})\_{[0,T]}, S^2(\mathbb{X})\_{[0,T]}, \ldots),$$ where the signature $S(\mathbb{X})\_{[0,T]}$ lives in the tensor algebra $T((\mathbb{R}^d)) = \oplus\_{i=0}^\infty (\mathbb{R}^d)^{\otimes k}$. We believe the reviewer was referring to the notation used in [1] where $S^k(\mathbb{X})\_{[0,T]}$ denotes the $k$-th level *truncated* signature, i.e. the mathematical object living in $T^k((\mathbb{R}^d))= \oplus\_{i=0}^k (\mathbb{R}^d)^{\otimes i}$, and hence $S(\mathbb{X})\_{[0,T]}$ is indeed retrieved by taking the limit as $k\rightarrow\infty$.
> - This equation was left unnumbered due to formatting reasons. We agree with the reviewer that this is quite an important equation and thus we will number it.
> - This corrector (aka control) was chosen based on a control variate argument: by adding an appropriately signed (and scaled) mean-zero control to the estimator we preserve the original estimator's bias while reducing its variance (the stronger the correlation between the control and the original estimator, the larger the reduction in variance). The form of the signature of a martingale yields a natural candidate for the choice of such control variate: by substituting the outermost Stratonovich integral with an Ito integral we obtain a mean-zero term which, by construction, should be highly correlated with the original estimator, thus leading to a significant reduction in variance. This is the content of footnote 6 on page 6. As this provides the main motivation for choosing the form of the corrected estimator we shall move the discussion contained therein to the main text.

---

> > ### Comment · Reviewer_v9sL · 2025-04-02
> >
> > I thank the authors for their detailed response. I still believe equations should be numbered even if they aren't cross referenced in the paper as future researchers may wish to reference them. I don't have any other concerns and I believe this is a good paper for ICML.

---

### Official Review · Reviewer_YgTV · 2025-03-14

**Overall Recommendation:** 4

**Summary:**

An empirical estimate of the expected signature of a stochastic process depends on the number of observed paths $N$ and the partition $\pi$ on which the paths are observed. This paper shows that under suitable conditions, the empirical estimate of the expected signature of a canonical geometric stochastic process is consistent and asymptotically normal in the double asymptotic limit of $N\rightarrow\infty$ and $|\pi|\rightarrow 0$. Brownian motion, fractional Brownian motion (fBM), a bidimensional continuous-time autoregressive process, and the price-variance dynamics of a Heston model are shown to meet the conditions for the empirical estimator to be consistent and all but fBM are shown to meet the conditions for the empirical estimator to be asymptotically normal. Furthermore, this paper shows that when the stochastic process is a martingale, you can introduce a zero-mean correction term which reduces the variance of the empirical estimator. The paper concludes by highlighting that the theoretical results deepen our understanding of existing machine learning approaches which use the expected signature and by demonstrating that the martingale correction leads to improved empirical performance, even for processes which are not martingale.

## Update after rebuttal

The authors addressed my minor stylistic comments and concerns regarding applying the method to non-martingale processes. I am happy to recommend acceptance after the rebuttal.

**Claims And Evidence:**

The main claims of this paper are theoretical, and I discuss these and the provided proofs in more detail in the relevant section. The evidence provided to demonstrate the empirical benefits of the martingale correction is clear and convincing. However, the paper does not provide a theoretical or intuitive justification for the improved empirical performance observed on non-martingale processes.

**Essential References Not Discussed:**

I am not aware of any missing essential references.

**Experimental Designs Or Analyses:**

The design of all experiments seems sound.

**Methods And Evaluation Criteria:**

A good demonstration of the benefits of the martingale correction is the comparison of pricing path-dependent derivatives. However, the other two classes of problem considered both contain non-martingale processes. This means that the martingale correction introduces a bias into the empirical estimator. Although the martingale correction is shown to be beneficial on these datasets, the broader implications of applying the correction to non-martingale paths and in which situations it is safe to do so are not discussed in the paper.

**Other Comments Or Suggestions:**

## Minor Comments

- On line 37, it is not clear to me what the informational content of the signature is and why this decays exponentially, when the terms of the signature decay factorially.
- On line 39, the signature was not introduced in rough path analysis. It dates back to `Iterated Integrals and Exponential Homomorphisms' by Chen (1954).
- On line 88, you swap the order of $n,\pi$ and $\pi,n$.
- The Theorems and their proofs in the Appendix are not well connected, with no references in the main body of the text to the location of the proofs.
- The proofs can be difficult to follow. The presentation would benefit from short outlines of the proof strategy at the beginning of the relevant sections and subsequent headings to section the proof. For example, this could be used to highlight the areas of the proof of Theorem 2.8 which concern conditions i), ii), and iv).
- On line 296, the abbreviation MCAR is not defined.
- On line 827, a second by is accidentally included.

**Other Strengths And Weaknesses:**

I have no other strengths or weaknesses I wish to highlight.

**Questions For Authors:**

I have three questions for the authors.

1) The structure of the proof of Theorem 2.8 under i), iii), and iv) suggests it may be possible to continue extending to $\alpha\in(1/(n+1),1/n]$ by assuming additional higher order bounds on the path increments. Is there a theoretical barrier to progressing to $\alpha\leq1/4$ or just progressively tedious algebra?
2) How should the martingale correction being empirically beneficial on non-martingale processes be interpreted? Is this a complete surprise or something that has a theoretical (or intuitive) explanation?
3)  A common setting in application is the ability to continue increasing the number of samples $N$, but being limited by the sampling rate $|\pi|$. Is it possible to comment on the asymptotic behaviour of the empirical estimator as just $N\rightarrow \infty$? Would you expect the estimator to be inconsistent in this limit?

**Relation To Broader Scientific Literature:**

This paper directly extends the work of Ni et al. (2012) and Passeggeri et al. (2020), who studied the asymptotic behaviour of the expected signature empirical estimator for Brownian motion and fractional Brownian motion respectively. Additionally, this paper gives a theoretical framework for understanding and extending existing machine learning applications of the expected signature, such as those in Lemercier (2021) and Triggiano et al. (2024).

**Theoretical Claims:**

The majority of this paper is dedicated to proving four theoretical claims.

1) Under suitable continuity conditions on $\mathbb{X}$, $S^{\mathbf{I}}(\mathbb{X}^{\pi_n})$ converging in probability to $S^{\mathbf{I}}(\mathbb{X})$ as $n\rightarrow\infty$ and $|\pi_n|\rightarrow0$ implies that it converges in $L^m$.
2) That if the sequence of paths $\\{\mathbb{X}^n, n\geq 1\\}$ is stationary and ergodic and you have a set of refining partitions as in (1), then the empirical estimator is consistent. If further the sequence $\\{\mathbb{X}^n, n\geq 1\\}$ is strongly mixing, then the empirical estimator is asymptotically normal.
3) That under suitable conditions, when $\mathbb{X}$ is a Gaussian process the empirical estimator is consistent.
4) If $\mathbb{X}$ is a martingale, then there exists a mean-zero correction term which reduces the variance of the empirical estimator, and can itself be estimated.

I have thoroughly reviewed the proof of claim (1) and found no errors. However, due to time constraints, I was only able to briefly review the proofs of claims (2), (3), and (4). No apparent issues were identified during this cursory examination. These claims are a significant theoretical contribution, providing an important advancement to the existing literature. However, given that the paper is highly theoretical and 55 pages long, I am concerned that a conference with an eight-page limit does not offer a suitable venue for adequately presenting or rigorously reviewing these theoretical results.

---

> ### Author Rebuttal · Authors · 2025-03-30
>
> We would like to thank the reviewer for the comments. We believe the reviewer has well understood the main contributions of the paper, which they summarized in "Theoretical Claims".
>
> **Minor comments**
> - Line 37: typo, the informational content (i.e. the norm) of signature terms decays factorially.
> - Line 39: we agree this was poorly worded, we shall update the references.
> - Line 88: we will fix notation to be consistent.
> - We will add references to the proof locations in the main text (e.g. Proof of Theorem 2.8. See Appendix A.1.)
> - We will add short outlines to the proofs (e.g. the main idea in the proof of Theorem 2.8 is to show the sequence of signatures along $\pi\_n$ is a Cauchy sequence in $L^m$). We shall also add a better partitioning between sections of this proof that concern conditions $(i)$, $(ii)$, and $(iv)$.
> - Line 296: we will spell out Multivariate Continuous-time Autoregressive Process (MCAR).
> - Line 827: typo.
>
> **Questions For Authors**
> 1. Yes, there is a theoretical barrier to progressing to $\alpha\leq 1/4$. This arises when applying Lemma A.1 to bound the sum of $Z^{\mathcal{I}}\_{[s,t]} $ with $i\_2=4$. More precisely, let $\alpha\leq 1/4$ and assume we follow the same proof strategy of Theorem 2.8 up to line 794 under assumption $(iv)$. Then, when $\alpha\leq 1/4$, we cannot apply (19) to bound the sum of the $Z^{\mathcal{I}}\_{[s,t]}$'s with $i\_2=4$ and so this term needs to be treated separately in the same way the $i\_2=2$ and the $i\_2=3$ had to be treated separately when $\alpha\leq 1/2$ and $\alpha\leq 1/3$ respectively. In a similar spirit to the $i\_2=2, 3$ cases we can thus attempt to apply Lemma A.1 to bound the sum of the $Z^{\mathcal{I}}\_{[s,t]}$'s. In this case though, even if we impose a condition similar to (A$\beta$) and (A$\gamma$) to bound the first term by $|\pi\_n|^{\zeta -1}$ for some $\zeta>1$, the second term, i.e. the sum of the $\\|Z^\mathcal{I}\_{s,t}\\|^2$, will be bounded by $|\pi\_n|^{8\alpha - 1/2}$ which does not necessarily vanish when $\alpha \leq 1/4$.
> 2. When the data generating process is not a martingale the corrected estimator introduces an additional bias which, depending on how "far" the process $\mathbb{X}$ is from being a martingale, might offset the benefits of the variance reduction. When $\mathbb{X}$ is "close" to being a martingale though, the corrected estimator is still theoretically better (in terms of MSE) than the classic estimator.
> 	Moreover, in practical applications, the corrected estimator might turn out to be a more informative quantity than the classic estimator. Intuitively, one might expect this to be the case when the drift of the underlying process is not relevant for the downstream task. In cases where the underlying process cannot be assumed to be a martingale we thus suggest to treat the martingale correction as a data transformation applicable in the learning pipeline (a model hyper-parameter in a similar spirit to the add-time or the lead-lag transform in the signature context) whose usefulness may be empirically ascertained via cross-validation. This is discussed in Appendix E.1 (paragraph starting on line 2436). If the reviewer feels this is an important point to include in the main text we would be happy to move the relevant discussion to Section 2.2.
> 3. Fixing the sampling rate leads to an irreducible lower bound on the precision with which we can learn the expected signature of the continuous-time process (in a similar spirit to the Nyquist–Shannon sampling theorem). If $\pi$ is fixed (e.g. because of constraints on the sampling rate) and we let $N\rightarrow \infty$ the empirical expected signature $\hat{\phi}\_{\mathbf{I}}^{\Pi(N)}(T)$ where $\Pi(N) = \pi \cup (T+ \pi) \cup \dots \cup((N-1)T+ \pi) $ is a consistent (and asymptotically normal) estimator for the expected signature $\phi^\pi\_{\mathbf{I}}(T) = \mathbb{E}[S^\mathbf{I}(\mathbb{X}^\pi)\_{[0,T]}]$ of the *piecewise linear* process $\mathbb{X}^\pi$ under the condition that $\\{\mathbb{X}^{n, \pi}, n\geq 1\\}$ – or sufficiently $\\{\mathbb{X}^{n}, n\geq 1\\}$ – is stationary and ergodic (and strongly mixing). This follows directly from the application of Birkhoff's ergodic theorem given in A.2.1 (and the dependent CLT in A.2.2) with $\mathbb{X}$ replaced by $\mathbb{X}^\pi$. The estimator $\hat{\phi}\_{\mathbf{I}}^{\Pi(N)}(T)$ is thus asymptotically (in the $N\rightarrow\infty$ limit) biased for the expected signature $\phi(T) = \mathbb{E}[S(\mathbb{X})\_{[0,T]}]$ of the underlying stochastic process $\mathbb{X}$, with bias $\phi^\pi(T) - \phi(T)$ (this is the "irreducible error" due to $\pi$-sampling). The in-fill asymptotic $|\pi|\downarrow 0$ is crucial to show such error vanishes, i.e. the empirical expected signature is a consistent estimator for the expected signature $\phi(T)$.

---

> > ### Comment · Reviewer_YgTV · 2025-04-02
> >
> > 1) Thank you for your clear response.
> >
> > 2) Thank you for this clarification. I feel that this point is not suitably addressed in the main body of the text or Appendix E.1. In particular, any modification could be considered a model configuration hyperparameter,  but the theory seems to suggest that this modification should only be beneficial when the process is a martingale. The empirical results would carry more weight if some aspects of your response to my question were included in the paper.
> >
> > 3) I was actually interested in the case where $\\{\pi_n\\}_{n \in \mathbb{N}}$ is a sequence of partitions of $[0,T]$, where each $|\pi_i|\geq\delta>0$ and the cumulative join $\bigvee_i \pi_i$ becomes dense in $[0,T]$ as $n \to \infty$, in the sense that $|\bigvee_i \pi_i|\rightarrow 0$, as opposed to the case of having $\pi$ fixed.

---

> > > ### Author Response · Authors · 2025-04-06
> > >
> > > 2. In case of acceptance, we will add part of this response to section 2.2 clarifying the trade-off between variance and bias when the process is not a martingale (and thus providing theoretical justification for using the martingale correction when the underlying process is not necessarily a martingale).
> > >
> > > 3. Thank you for clarifying the comment. If we constrain the sampling rate to be at most $\\delta$, i.e. $\\min\_{[u,v]\\in\\pi}|v-u|\\geq\\delta$, then we cannot obtain a sequence of partitions $\\pi\_i$ such that $\\bigvee\_i\\pi_i$ is dense in $[0,T]$ (the best we can do is to have $\\bigvee\_i\\pi_i$ dense in $\\{0\\}\\cup[\\delta,T-\\delta]\\cup\\{T\\}$). If instead we constrain the mesh $|\\pi|=\\max\_{[u,v]\\in\\pi}|v-u|\\geq\\delta$ then each partition might have observations arbitrarily close to each other (which we believe is not the realistic scenario the reviewer is interested in). As a middle ground between these two cases let us consider the following example, where we constrain the sampling rate of each partition to be at most $\\delta$ except for the first and last observation points, so that we can obtain $\\bigvee\_i \\pi\_i$ to be a dense subset of $[0,T]$. This example shows that an irreducible error similar to the one discussed in the initial response to this comment might still arise despite $\\bigvee\_i\\pi_i$ being dense in $[0,T]$.
> > >
> > > *Example.* Consider a two-dimensional Brownian motion $\\mathbb{W}=\\{\\mathbf{W}\_t\\in\\mathbb{R}^2,t\\in[0,T]\\}$. Assume for simplicity that the sampling rate $\\delta$ divides $T$, i.e. $\\delta K=T$. Let $\\{\\pi\_n,n\\geq 0\\}$ be a sequence of partitions such that
> > > $$\\pi\_n=\\{0,\\theta\_n,\\theta\_n+\\delta,\\theta\_n+2\\delta,\\ldots,\\theta\_n+(K-1)\\delta,T\\},$$
> > > for $\\theta\_n\\in(0,\\delta]$ with $\\{\\theta\_n,n\\geq0\\}$ dense in $(0,\\delta]$. For example one may take
> > > $$\\{\\theta\_n,n\\geq 0\\}=\\{\\delta,\\delta/2,\\delta/3,2\\delta/3,\\delta/4,3\\delta/4,\\ldots\\},$$
> > > ensuring that $\\bigvee\_n\\pi\_n$ is dense in $[0,T]$.
> > > Let us consider the task of estimating $\\phi\_I(T)=\\mathbb{E}[S^I(\\mathbb{W})\_{[0,T]}]$ where $I=(1,1,2,2)$.
> > > We can show that
> > > $$
> > > \\begin{align*}
> > > \\mathbb{E}[S^{I}(\\mathbb{W}^{\\pi\_n})\_{[0,T]}]&=\\frac{T^2}{8}-\\frac{T\\delta}{12}+\\frac{\\theta\_n(\\delta-\\theta\_n)}{6}\\\\
> > > &\\leq\\frac{T^2}{8}-\\frac{T\\delta}{12}+\\frac{\\delta^2}{24}\\\\
> > > &<\\frac{T^2}{8}=\\mathbb{E}[S^{I}(\\mathbb{W}^{\\pi\_n})\_{[0,T]}]=:\\phi\_I(T).
> > > \\end{align*}
> > > $$
> > > For the first equality see (\*) below while for the last equality see [4], Theorem 3.1. Setting
> > > $$
> > > \\hat{\\phi}^N\_I(T):=\\frac{1}{N}\\sum\_{n=1}^N S^{I}(\\mathbb{W}^{\\pi\_n})\_{[0,T]},
> > > $$
> > > for a set of independent observations $\\{\\mathbb{W}^{\\pi\_n},n\\geq1\\}$, we can hence deduce that for all $N\\geq1$,
> > > $$
> > > \\begin{align*}
> > > \\mathbb{E}[\\hat{\\phi}^N\_I(T)]\\leq\\frac{T^2}{8}-\\frac{T\\delta}{12}+\\frac{\\delta^2}{24}<\\phi\_I(T).
> > > \\end{align*}
> > > $$
> > > By applying the shuffle property of the signature and explictly computing the corresponding terms, we can show that $\\mathbb{E}[S^{I}(\\mathbb{W}^{\\pi\_n})^2\_{[0,T]}]$ is uniformly bounded and hence $\\{\\hat{\\phi}^N\_I(T),N\\geq1\\}$ is uniformly integrable. Under uniform integrability, convergence in distribution would imply convergence of means but since
> > > $$
> > > \\begin{align*}
> > > \\mathbb{E}[\\hat{\\phi}^N\_I(T)]\\not\\rightarrow\\phi\_I(T),\\quad N\\rightarrow\\infty,
> > > \\end{align*}
> > > $$
> > > we can deduce
> > > $$
> > > \\hat{\\phi}^N\_I(T)\\not\\rightarrow\\phi\_I(T),\\quad N\\rightarrow\\infty,
> > > $$
> > > in any sense (not even in distribution).
> > >
> > > ___
> > >
> > >
> > > (\*) We first apply Chen's relation and the fact that Brownian motion has independent and stationary increments to deduce that
> > > $$
> > > \\begin{align*}
> > > \\mathbb{E}[S(\\mathbb{W}^{\\pi\_n})\_{[0,T]}]&=\\mathbb{E}[S(\\mathbb{W}^{\\pi\_n})\_{[0,\\theta\_n]}\\otimes S(\\mathbb{W}^{\\pi\_n})\_{[\\theta\_n,T-\\delta+\\theta\_n]}\\otimes S(\\mathbb{W}^{\\pi\_n})\_{[T-\\delta+\\theta\_n,T]}]\\\\
> > > &=\\mathbb{E}[\\exp\_{\\otimes}(\\mathbf{W}\_{0,\\theta\_n})]\\otimes\\mathbb{E}[S(\\mathbb{W}^{\\pi})\_{[0,T-\\delta]}]\\otimes\\mathbb{E}[\\exp\_{\\otimes}(\\mathbf{W}\_{0,\\delta-\\theta\_n})],
> > > \\end{align*}
> > > $$
> > > where $\\pi$ is the partition corresponding to $\\theta=\\delta$. To obtain the $I=(1,1,2,2)$ term we expand the tensor product and plug in the analytic form of the expected signature of piecewise linear Brownian motion given in Theorem 3.2 of [4].
> > >
> > > [4] Ni, H. (2012). The expected signature of a stochastic process. PhD thesis. University of Oxford. https://ora.ox.ac.uk/objects/uuid:e0b9e045-4c09-4cb7-ace9-46c4984f16f6

---

### Decision · Program_Chairs · 2025-05-01

**Decision:**

Accept (oral)

**Comment:**

This paper establishes central limit theorems of empirical risk-based estimators of signatures for representing time-series for certain classes of generative processes that satisfy a martingale property, as well as corrections for processes that do not. They establish that this model applies to variants of Brownian motion and other stochastic differential equations that arise in options pricing, especially the rough Heston model. Numerous theoretical and experimental contributions support the key claims of this work, which the reviewers unanimously appreciated.